# VIPeR: Provably Efficient Algorithm for Offline RL with Neural Function Approximation

**Thanh Nguyen-Tang**
Department of Computer Science
Johns Hopkins University
Baltimore, MD 21218, USA
nguyent@cs.jhu.edu

**Raman Arora**
Department of Computer Science
Johns Hopkins University
Baltimore, MD 21218, USA
arora@cs.jhu.edu

## Abstract

We propose a novel algorithm for offline reinforcement learning called Value Iteration with Perturbed Rewards (VIPeR), which amalgamates the pessimism principle with random perturbations of the value function. Most current offline RL algorithms explicitly construct statistical confidence regions to obtain pessimism via lower confidence bounds (LCB), which cannot easily scale to complex problems where a neural network is used to estimate the value functions. Instead, VIPeR implicitly obtains pessimism by simply perturbing the offline data multiple times with carefully-designed i.i.d. Gaussian noises to learn an ensemble of estimated state-action value functions and acting greedily with respect to the minimum of the ensemble. The estimated state-action values are obtained by fitting a parametric model (e.g., neural networks) to the perturbed datasets using gradient descent. As a result, VIPeR only needs $\mathcal{O}(1)$ time complexity for action selection, while LCB-based algorithms require at least $\Omega(K^2)$, where $K$ is the total number of trajectories in the offline data. We also propose a novel data-splitting technique that helps remove a factor involving the log of the covering number in our bound. We prove that VIPeR yields a provable uncertainty quantifier with overparameterized neural networks and enjoys a bound on sub-optimality of $\tilde{\mathcal{O}}(\kappa H^{5/2}\tilde{d}/\sqrt{K})$, where $\tilde{d}$ is the effective dimension, $H$ is the horizon length and $\kappa$ measures the distributional shift. We corroborate the statistical and computational efficiency of VIPeR with an empirical evaluation on a wide set of synthetic and real-world datasets. To the best of our knowledge, VIPeR is the first algorithm for offline RL that is provably efficient for general Markov decision processes (MDPs) with neural network function approximation.

## 1 Introduction

Offline reinforcement learning (offline RL) (Lange et al., 2012; Levine et al., 2020) is a practical paradigm of RL for domains where active exploration is not permissible. Instead, the learner can access a fixed dataset of previous experiences available a priori. Offline RL finds applications in several critical domains where exploration is prohibitively expensive or even implausible, including healthcare (Gottesman et al., 2019; Nie et al., 2021), recommendation systems (Strehl et al., 2010; Thomas et al., 2017), and econometrics (Kitagawa & Tetenov, 2018; Athey & Wager, 2021), among others. The recent surge of interest in this area and renewed research efforts have yielded several important empirical successes (Chen et al., 2021; Wang et al., 2023; 2022; Meng et al., 2021).

A key challenge in offline RL is to efficiently exploit the given offline dataset to learn an optimal policy in the absence of any further exploration. The dominant approaches to offline RL address this challenge by incorporating uncertainty from the offline dataset into decision-making (Buckman et al., 2021; Jin et al., 2021; Xiao et al., 2021; Nguyen-Tang et al., 2022a; Ghasemipour et al., 2022; An et al., 2021; Bai et al., 2022). The main component of these uncertainty-aware approaches to offline RL is the *pessimism* principle, which constrains the learned policy to the offline data and leads to various lower confidence bound (LCB)-based algorithms. However, these methods are not easily extended or scaled to complex problems where neural function approximation is used to estimate

the value functions. In particular, it is costly to explicitly compute the statistical confidence regions of the model or value functions if the class of function approximator is given by overparameterized neural networks. For example, constructing the LCB for neural offline contextual bandits (Nguyen-Tang et al., 2022a) and RL (Xu & Liang, 2022) requires computing the inverse of a large covariance matrix whose size scales with the number of parameters in the neural network. This computational cost hinders the practical application of these provably efficient offline RL algorithms. Therefore, a largely open question is *how to design provably computationally efficient algorithms for offline RL with neural network function approximation.*

In this work, we present a solution based on a computational approach that combines the pessimism principle with randomizing the value function (Osband et al., 2016; Ishfaq et al., 2021). The algorithm is strikingly simple: we randomly perturb the offline rewards several times and act greedily with respect to the minimum of the estimated state-action values. The intuition is that taking the minimum from an ensemble of randomized state-action values can efficiently achieve pessimism with high probability while avoiding explicit computation of statistical confidence regions. We learn the state-action value function by training a neural network using gradient descent (GD). Further, we consider a novel data-splitting technique that helps remove the dependence on the potentially large log covering number in the learning bound. We show that the proposed algorithm yields a provable uncertainty quantifier with overparameterized neural network function approximation and achieves a sub-optimality bound of $\tilde{\mathcal{O}}(\kappa H^{5/2} \tilde{d} / \sqrt{K})$, where $K$ is the total number of episodes in the offline data, $\tilde{d}$ is the effective dimension, $H$ is the horizon length, and $\kappa$ measures the distributional shift. We achieve computational efficiency since the proposed algorithm only needs $\mathcal{O}(1)$ time complexity for action selection, while LCB-based algorithms require $\mathcal{O}(K^2)$ time complexity. We empirically corroborate the statistical and computational efficiency of our proposed algorithm on a wide set of synthetic and real-world datasets. The experimental results show that the proposed algorithm has a strong advantage in computational efficiency while outperforming LCB-based neural algorithms. To the best of our knowledge, ours is the first offline RL algorithm that is both provably and computationally efficient in general MDPs with neural network function approximation.

## 2 RELATED WORK

**Randomized value functions for RL.** For online RL, Osband et al. (2016; 2019) were the first to explore randomization of estimates of the value function for exploration. Their approach was inspired by posterior sampling for RL (Osband et al., 2013), which samples a value function from a posterior distribution and acts greedily with respect to the sampled function. Concretely, Osband et al. (2016; 2019) generate randomized value functions by injecting Gaussian noise into the training data and fitting a model on the perturbed data. Jia et al. (2022) extended the idea of perturbing rewards to online contextual bandits with neural function approximation. Ishfaq et al. (2021) obtained a provably efficient method for online RL with general function approximation using the perturbed rewards. While randomizing the value function is an intuitive approach to obtaining optimism in online RL, obtaining pessimism from the randomized value functions can be tricky in offline RL. Indeed, Ghasemipour et al. (2022) point out a critical flaw in several popular existing methods for offline RL that update an ensemble of randomized Q-networks toward a *shared* pessimistic temporal difference target. In this paper, we propose a simple fix to obtain pessimism properly by updating each randomized value function independently and taking the minimum over an ensemble of randomized value functions to form a pessimistic value function.

**Offline RL with function approximation.** Provably efficient offline RL has been studied extensively for linear function approximation. Jin et al. (2021) were the first to show that pessimistic value iteration is provably efficient for offline linear MDPs. Xiong et al. (2023); Yin et al. (2022) improved upon Jin et al. (2021) by leveraging variance reduction. Xie et al. (2021) proposed a Bellman-consistency assumption with general function approximation, which improves the bound of Jin et al. (2021) by a factor of $\sqrt{d}$ when realized to finite action space and linear MDPs. Wang et al. (2021); Zanette (2021) studied the statistical hardness of offline RL with linear function approximation via exponential lower bound, and Foster et al. (2021) suggested that only realizability and strong uniform data coverage are not sufficient for sample-efficient offline RL. Beyond linearity, some works study offline RL for general function approximation, both parametric and nonparametric. These approaches are either based on Fitted-Q Iteration (FQI) (Munos & Szepesvári, 2008; Le

et al., 2019; Chen & Jiang, 2019; Duan et al., 2021a;b; Hu et al., 2021; Nguyen-Tang et al., 2022b) or the pessimism principle (Uehara & Sun, 2022; Nguyen-Tang et al., 2022a; Jin et al., 2021). While pessimism-based algorithms avoid the strong assumptions of data coverage used by FQI-based algorithms, they require an explicit computation of valid confidence regions and possibly the inverse of a large covariance matrix which is computationally prohibitive and does not scale to complex function approximation setting. This limits the applicability of pessimism-based, provably efficient offline RL to practical settings. A very recent work Bai et al. (2022) estimates the uncertainty for constructing LCB via the disagreement of bootstrapped Q-functions. However, the uncertainty quantifier is only guaranteed in linear MDPs and must be computed explicitly.

We provide a more detailed discussion of our technical contribution in the context of existing literature in Section C.1.

## 3 PRELIMINARIES

In this section, we provide basic background on offline RL and overparameterized neural networks.

### 3.1 EPISODIC TIME-INHOMOGENOUS MARKOV DECISION PROCESSES (MDPs)

A finite-horizon Markov decision process (MDP) is denoted as the tuple $\mathcal{M} = (\mathcal{S}, \mathcal{A}, \mathbb{P}, r, H, d_1)$, where $\mathcal{S}$ is an arbitrary state space, $\mathcal{A}$ an arbitrary action space, $H$ the episode length, and $d_1$ the initial state distribution. We assume that $SA := |\mathcal{S}||\mathcal{A}|$ is finite but arbitrarily large, e.g., it can be as large as the total number of atoms in the observable universe $\approx 10^{82}$. Let $\mathcal{P}(\mathcal{S})$ denote the set of probability measures over $\mathcal{S}$. A time-inhomogeneous transition kernel $\mathbb{P} = \{\mathbb{P}_h\}_{h=1}^H$, where $\mathbb{P}_h : \mathcal{S} \times \mathcal{A} \to \mathcal{P}(\mathcal{S})$ maps each state-action pair $(s_h, a_h)$ to a probability distribution $\mathbb{P}_h(\cdot|s_h, a_h)$. Let $r = \{r_h\}_{h=1}^H$ where $r_h : \mathcal{S} \times \mathcal{A} \to [0, 1]$ is the mean reward function at step $h$. A policy $\pi = \{\pi_h\}_{h=1}^H$ assigns each state $s_h \in \mathcal{S}$ to a probability distribution, $\pi_h(\cdot|s_h)$, over the action space and induces a random trajectory $s_1, a_1, r_1, \ldots, s_H, a_H, r_H, s_{H+1}$ where $s_1 \sim d_1$, $a_h \sim \pi_h(\cdot|s_h)$, $s_{h+1} \sim \mathbb{P}_h(\cdot|s_h, a_h)$. We define the state value function $V_h^\pi \in \mathbb{R}^{\mathcal{S}}$ and the action-state value function $Q_h^\pi \in \mathbb{R}^{\mathcal{S} \times \mathcal{A}}$ at each timestep $h$ as $Q_h^\pi(s, a) = \mathbb{E}_\pi[\sum_{t=h}^H r_t | s_h = s, a_h = a]$, and $V_h^\pi(s) = \mathbb{E}_{a \sim \pi(\cdot|s)}[Q_h^\pi(s, a)]$, where the expectation $\mathbb{E}_\pi$ is taken with respect to the randomness of the trajectory induced by $\pi$. Let $\mathbb{P}_h$ denote the transition operator defined as $(\mathbb{P}_h V)(s, a) := \mathbb{E}_{s' \sim \mathbb{P}_h(\cdot|s,a)}[V(s')]$. For any $V : \mathcal{S} \to \mathbb{R}$, we define the Bellman operator at timestep $h$ as $(\mathbb{B}_h V)(s, a) := r_h(s, a) + (\mathbb{P}_h V)(s, a)$. The Bellman equations are given as follows. For any $(s, a, h) \in \mathcal{S} \times \mathcal{A} \times [H]$,

$$Q_h^\pi(s, a) = (\mathbb{B}_h V_{h+1}^\pi)(s, a), \quad V_h^\pi(s) = \langle Q_h^\pi(s, \cdot), \pi_h(\cdot|s) \rangle_{\mathcal{A}}, \quad V_{H+1}^\pi(s) = 0,$$

where $[H] := \{1, 2, \ldots, H\}$, and $\langle \cdot, \cdot \rangle_{\mathcal{A}}$ denotes the summation over all $a \in \mathcal{A}$. We define an optimal policy $\pi^*$ as any policy that yields the optimal value function, i.e. $V_h^{\pi^*}(s) = \sup_\pi V_h^\pi(s)$ for any $(s, h) \in \mathcal{S} \times [H]$. For simplicity, we denote $V_h^{\pi^*}$ and $Q_h^{\pi^*}$ as $V_h^*$ and $Q_h^*$, respectively. The Bellman optimality equation can be written as

$$Q_h^*(s, a) = (\mathbb{B}_h V_{h+1}^*)(s, a), \quad V_h^*(s) = \max_{a \in \mathcal{A}} Q_h^*(s, a), \quad V_{H+1}^*(s) = 0.$$

Define the occupancy density as $d_h^\pi(s, a) := \mathbb{P}((s_h, a_h) = (s, a)|\pi)$ which is the probability that we visit state $s$ and take action $a$ at timestep $h$ if we follow the policy $\pi$. We denote $d_h^{\pi^*}$ by $d_h^*$.

**Offline regime.** In the offline regime, the learner has access to a fixed dataset $\mathcal{D} = \{(s_h^t, a_h^t, r_h^t, s_{h+1}^t)\}_{h \in [H]}^{t \in [K]}$ generated a priori by some unknown behaviour policy $\mu = \{\mu_h\}_{h \in [H]}$. Here, $K$ is the total number of trajectories, and $a_h^t \sim \mu_h(\cdot|s_h^t), s_{h+1}^t \sim \mathbb{P}_h(\cdot|s_h^t, a_h^t)$ for any $(t, h) \in [K] \times [H]$. Note that we allow the trajectory at any time $t \in [K]$ to depend on the trajectories at previous times. The goal of offline RL is to learn a policy $\hat{\pi}$, based on (historical data) $\mathcal{D}$, such that $\hat{\pi}$ achieves small sub-optimality, which we define as

$$\text{SubOpt}(\hat{\pi}) := \mathbb{E}_{s_1 \sim d_1} [\text{SubOpt}(\hat{\pi}; s_1)], \quad \text{where } \text{SubOpt}(\hat{\pi}; s_1) := V_1^{\pi^*}(s_1) - V_1^{\hat{\pi}}(s_1).$$

---

**Algorithm 1** Value Iteration with Perturbed Rewards (VIPeR)

---

1: **Input**: Offline data $\mathcal{D} = \{(s_h^k, a_h^k, r_h^k)\}_{h \in [H]}^{k \in [K]}$, a parametric function family $\mathcal{F} = \{f(\cdot, \cdot; W) : W \in \mathcal{W}\} \subset \{\mathcal{X} \to \mathbb{R}\}$ (e.g. neural networks), perturbed variances $\{\sigma_h\}_{h \in [H]}$, number of bootstraps $M$, regularization parameter $\lambda$, step size $\eta$, number of gradient descent steps $J$, and cutoff margin $\psi$, split indices $\{\mathcal{I}_h\}_{h \in [H]}$ where $\mathcal{I}_h := [(H - h)K' + 1, \ldots, (H - h + 1)K']$
2: Initialize $\tilde{V}_{H+1}(\cdot) \leftarrow 0$ and initialize $f(\cdot, \cdot; W)$ with initial parameter $W_0$
3: **for** $h = H, \ldots, 1$ **do**
4:     **for** $i = 1, \ldots, M$ **do**
5:         Sample $\{\xi_h^{k,i}\}_{k \in \mathcal{I}_h} \sim \mathcal{N}(0, \sigma_h^2)$ and $\zeta_h^i = \{\zeta_h^{j,i}\}_{j \in [d]} \sim \mathcal{N}(0, \sigma_h^2 I_d)$
6:         Perturb the dataset $\tilde{\mathcal{D}}_h^i \leftarrow \{s_h^k, a_h^k, r_h^k + \tilde{V}_{h+1}(s_{h+1}^k) + \xi_h^{k,i}\}_{k \in \mathcal{I}_h}$     ▷ *Perturbation*
7:         Let $\tilde{W}_h^i \leftarrow \text{GradientDescent}(\lambda, \eta, J, \tilde{\mathcal{D}}_h^i, \zeta_h^i, W_0)$ (Algorithm 2)     ▷ *Optimization*
8:     **end for**
9:     Compute $\tilde{Q}_h(\cdot, \cdot) \leftarrow \min\{\min_{i \in [M]} f(\cdot, \cdot; \tilde{W}_h^i), (H - h + 1)(1 + \psi)\}^+$     ▷ *Pessimism*
10:     $\tilde{\pi}_h \leftarrow \arg\max_{\pi_h} \langle \tilde{Q}_h, \pi_h \rangle$ and $\tilde{V}_h \leftarrow \langle \tilde{Q}_h, \tilde{\pi}_h \rangle$     ▷ *Greedy*
11: **end for**
12: **Output**: $\tilde{\pi} = \{\tilde{\pi}_h\}_{h \in [H]}$.

---

**Notation.** For simplicity, we write $x_h^t = (s_h^t, a_h^t)$ and $x = (s, a)$. We write $\tilde{\mathcal{O}}(\cdot)$ to hide logarithmic factors of the problem parameters $(d, H, K, m, 1/\delta)$ in the standard Big-Oh notation. We use $\Omega(\cdot)$ as the standard Omega notation. We write $u \lesssim v$ if $u = \mathcal{O}(v)$ and write $u \gtrsim v$ if $v \lesssim u$. We write $A \preceq B$ iff $B - A$ is a positive definite matrix. $I_d$ denotes the $d \times d$ identity matrix.

### 3.2 Overparameterized Neural Networks

In this paper, we consider neural function approximation setting where the state-action value function is approximated by a two-layer neural network. For simplicity, we denote $\mathcal{X} := \mathcal{S} \times \mathcal{A}$ and view it as a subset of $\mathbb{R}^d$. Without loss of generality, we assume $\mathcal{X} \subset \mathbb{S}_{d-1} := \{x \in \mathbb{R}^d : \|x\|_2 = 1\}$. We consider a standard two-layer neural network: $f(x; W, b) = \frac{1}{\sqrt{m}} \sum_{i=1}^m b_i \sigma(w_i^T x)$, where $m$ is an even number, $\sigma(\cdot) = \max\{\cdot, 0\}$ is the ReLU activation function (Arora et al., 2018), and $W = (w_1^T, \ldots, w_m^T)^T \in \mathbb{R}^{md}$. During the training, we initialize $(W, b)$ via the symmetric initialization scheme (Gao et al., 2019) as follows: For any $i \leq \frac{m}{2}$, $w_i = w_{\frac{m}{2}+i} \sim \mathcal{N}(0, I_d/d)$, and $b_{\frac{m}{2}+i} = -b_i \sim \text{Unif}(\{-1, 1\})$.[1] During the training, we optimize over $W$ while the $b_i$ are kept fixed, thus we write $f(x; W, b)$ as $f(x; W)$. Denote $g(x; W) = \nabla_W f(x; W) \in \mathbb{R}^{md}$, and let $W_0$ be the initial parameters of $W$. We assume that the neural network is overparameterized, i.e, the width $m$ is sufficiently larger than the number of samples $K$. Overparameterization has been shown to be effective in studying the convergence and the interpolation behaviour of neural networks (Arora et al., 2019; Allen-Zhu et al., 2019; Hanin & Nica, 2020; Cao & Gu, 2019; Belkin, 2021). Under such an overparameterization regime, the dynamics of the training of the neural network can be captured using the framework of the neural tangent kernel (NTK) (Jacot et al., 2018).

## 4 Algorithm

In this section, we present the proposed algorithm called Value Iteration with Perturbed Rewards, or VIPeR; see Algorithm 1 for the pseudocode. The key idea underlying VIPeR is to train a parametric model (e.g., a neural network) on a perturbed-reward dataset several times and act pessimistically by picking the minimum over an ensemble of estimated state-action value functions. In particular, at each timestep $h \in [H]$, we draw $M$ independent samples of zero-mean Gaussian noise with variance $\sigma_h$. We use these samples to perturb the sum of the observed rewards, $r_h^k$, and the estimated value function with a one-step lookahead, i.e., $\tilde{V}_{h+1}(s_{h+1}^k)$ (see Line 6 of Algorithm 1). The weights $\tilde{W}_h^i$ are then updated by minimizing the perturbed regularized squared loss on $\{\tilde{D}_h^i\}_{i \in [M]}$ using gradient descent (Line 7). We pick the value function pessimistically by selecting the minimum over the finite ensemble. The chosen value function is truncated at $(H - h + 1)(1 + \psi)$ (see Line 9), where

---

[1] This symmetric initialization scheme makes $f(x; W_0) = 0$ and $\langle g(x; W_0), W_0 \rangle = 0$ for any $x$.

$\psi \geq 0$ is a small cutoff margin (more on this when we discuss the theoretical analysis). The returned policy is greedy with respect to the truncated pessimistic value function (see Line 10).

It is important to note that we split the trajectory indices $[K]$ evenly into $H$ disjoint buckets $[K] = \cup_{h \in [H]} \mathcal{I}_h$, where $\mathcal{I}_h = [(H - h)K' + 1, \ldots, (H - h + 1)K']$ for $K' := \lfloor K/H \rfloor^2$, as illustrated in Figure 1. The estimated $\tilde{Q}_h$ is thus obtained only from the offline data with (trajectory) indices from $\mathcal{I}_h$ along with $\tilde{V}_{h+1}$. This novel design removes the data dependence structure in offline RL with function approximation (Nguyen-Tang et al., 2022b) and avoids a factor involving the log of the covering number in the bound on the sub-optimality of Algorithm 1, as we show in Section D.1.

---

**Algorithm 2** GradientDescent($\lambda, \eta, J, \tilde{\mathcal{D}}_h^i, \zeta_h^i, W_0$)

1: **Input:** Regularization parameter $\lambda$, step size $\eta$, number of gradient descent steps $J$, perturbed dataset $\tilde{\mathcal{D}}_h^i = \{s_h^k, a_h^k, r_h^k + \tilde{V}_{h+1}(s_{h+1}^k) + \xi_h^{t,i}\}_{k \in \mathcal{I}_h}$, regularization perturber $\zeta_h^i$, initial parameter $W_0$
2: $L(W) := \frac{1}{2} \sum_{k \in \mathcal{I}_h} (f(s_h^k, a_h^k; W) - (r_h^k + \tilde{V}_{h+1}(s_{h+1}^k) + \xi_h^{k,i}))^2 + \frac{\lambda}{2} \|W + \zeta_h^i - W_0\|_2^2$
3: **for** $j = 0, \ldots, J - 1$ **do**
4: $\quad W_{j+1} \leftarrow W_j - \eta \nabla L(W_j)$
5: **end for**
6: **Output:** $W_J$.

---

To deal with the non-linearity of the underlying MDP, we use a two-layer fully connected neural network as the parametric function family $\mathcal{F}$ in Algorithm 1. In other words, we approximate the state-action values: $f(x; W) = \frac{1}{\sqrt{m}} \sum_{i=1}^m b_i \sigma(w_i^T x)$, as described in Section 3.2. We use two-layer neural networks to simplify the computational analysis. We utilize gradient descent to train the state-action value functions $\{f(\cdot, \cdot; \tilde{W}_h^i)\}_{i \in [M]}$, on perturbed rewards. The use of gradient descent is for the convenience of computational analysis, and our results can be extended to stochastic gradient descent by leveraging recent advances in the theory of deep learning (Allen-Zhu et al., 2019; Cao & Gu, 2019), albeit with a more involved analysis.

Existing offline RL algorithms utilize estimates of statistical confidence regions to achieve pessimism in the offline setting. Explicitly constructing these confidence bounds is computationally expensive in complex problems where a neural network is used for function approximation. For example, the lower-confidence-bound-based algorithms in neural offline contextual bandits (Nguyen-Tang et al., 2022a) and RL (Xu & Liang, 2022) require computing the inverse of a large covariance matrix with the size scaling with the number of network parameters. This is computationally prohibitive in most practical settings. Algorithm 1 (VIPeR) avoids such expensive computations while still obtaining provable pessimism and guaranteeing a rate of $\tilde{\mathcal{O}}(\frac{1}{\sqrt{K}})$ on the sub-optimality, as we show in the next section.

## 5 SUB-OPTIMALITY ANALYSIS

Next, we provide a theoretical guarantee on the sub-optimality of VIPeR for the function approximation class, $\mathcal{F}$, represented by (overparameterized) neural networks. Our analysis builds on the recent advances in generalization and optimization of deep neural networks (Arora et al., 2019; Allen-Zhu et al., 2019; Hanin & Nica, 2020; Cao & Gu, 2019; Belkin, 2021) that leverage the observation that the dynamics of the neural parameters learned by (stochastic) gradient descent can be captured by the corresponding neural tangent kernel (NTK) space (Jacot et al., 2018) when the network is overparameterized.

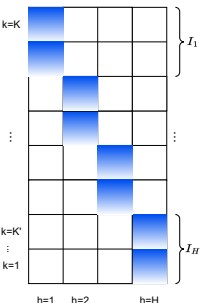

Figure 1: Data splitting.

Next, we recall some definitions and state our key assumptions, formally.

**Definition 1** (NTK (Jacot et al., 2018)). *The NTK kernel $K_{ntk} : \mathcal{X} \times \mathcal{X} \rightarrow \mathbb{R}$ is defined as*

$$K_{ntk}(x, x') = \mathbb{E}_{w \sim \mathcal{N}(0, I_d/d)} \langle x\sigma'(w^T x), x'\sigma'(w^T x') \rangle,$$

*where $\sigma'(u) = \mathbb{1}\{u \geq 0\}$.*

---

[2]Without loss of generality, we assume $K/H \in \mathbb{N}$.

Let $\mathcal{H}_{ntk}$ denote the reproducing kernel Hilbert space (RKHS) induced by the NTK, $K_{ntk}$. Since $K_{ntk}$ is a universal kernel (Ji et al., 2020), we have that $\mathcal{H}_{ntk}$ is dense in the space of continuous functions on (a compact set) $\mathcal{X} = \mathcal{S} \times \mathcal{A}$ (Rahimi & Recht, 2008).

**Definition 2** (Effective dimension). *For any $h \in [H]$, the effective dimension of the NTK matrix on data $\{x_h^k\}_{k \in \mathcal{I}_h}$ is defined as*

$$\tilde{d}_h := \frac{\log\det(I_{K'} + \mathcal{K}_h/\lambda)}{\log(1 + K'/\lambda)},$$

*where $\mathcal{K}_h := [K_{ntk}(x_h^i, x_h^j)]_{i,j \in \mathcal{I}_h}$ is the Gram matrix of $K_{ntk}$ on the data $\{x_h^k\}_{k \in \mathcal{I}_h}$. We further define $\tilde{d} := \max_{h \in [H]} \tilde{d}_h$.*

**Remark 1.** *Intuitively, the effective dimension $\tilde{d}_h$ measures the number of principal dimensions over which the projection of the data $\{x_h^k\}_{k \in \mathcal{I}_h}$ in the RKHS $\mathcal{H}_{ntk}$ is spread. It was first introduced by Valko et al. (2013) for kernelized contextual bandits and was subsequently adopted by Yang & Wang (2020) and Zhou et al. (2020) for kernelized RL and neural contextual bandits, respectively. The effective dimension is data-dependent and can be bounded by $\tilde{d} \lesssim K'^{(d+1)/(2d)}$ in the worst case (see Section B for more details).*[3]

**Definition 3** (RKHS of the infinite-width NTK). *Define $\mathcal{Q}^* := \{f(x) = \int_{\mathbb{R}^d} c(w)^T x \sigma'(w^T x) dw : \sup_w \frac{\|c(w)\|_2}{p_0(w)} < B\}$, where $c : \mathbb{R}^d \to \mathbb{R}^d$ is any function, $p_0$ is the probability density function of $\mathcal{N}(0, I_d/d)$, and $B$ is some positive constant.*

We make the following assumption about the regularity of the underlying MDP under function approximation.

**Assumption 5.1** (Completeness). *For any $V : \mathcal{S} \to [0, H+1]$ and any $h \in [H]$, $\mathbb{B}_h V \in \mathcal{Q}^*$.*[4]

Assumption 5.1 ensures that the Bellman operator $\mathbb{B}_h$ can be captured by an infinite-width neural network. This assumption is mild as $\mathcal{Q}^*$ is a dense subset of $\mathcal{H}_{ntk}$ (Gao et al., 2019, Lemma C.1) when $B = \infty$, thus $\mathcal{Q}^*$ is an expressive function class when $B$ is sufficiently large. Moreover, similar assumptions have been used in many prior works on provably efficient RL with function approximation (Cai et al., 2019; Wang et al., 2020; Yang et al., 2020; Nguyen-Tang et al., 2022b).

Next, we present a bound on the suboptimality of the policy $\tilde{\pi}$ returned by Algorithm 1. Recall that we use the initialization scheme described in Section 3.2. Fix any $\delta \in (0, 1)$.

**Theorem 1.** *Let $\sigma_h = \sigma := 1 + \lambda^{\frac{1}{2}} B + (H+1)\left[\tilde{d}\log(1 + K'/\lambda) + 2 + 2\log(3H/\delta)\right]^{\frac{1}{2}}$. Let $m = poly(K', H, d, B, \tilde{d}, \lambda, \delta)$ be some high-order polynomial of the problem parameters, $\lambda = 1 + \frac{H}{K}$, $\eta \lesssim (\lambda + K')^{-1}$, $J \gtrsim K' \log(K'(H\sqrt{\tilde{d}} + B))$, $\psi = 1$, and $M = \log\frac{HSA}{\delta}/\log\frac{1}{1-\Phi(-1)}$, where $\Phi(\cdot)$ is the cumulative distribution function of the standard normal distribution. Then, under Assumption 5.1, with probability at least $1 - MHm^{-2} - 2\delta$, for any $s_1 \in \mathcal{S}$, we have that*

$$\mathrm{SubOpt}(\tilde{\pi}; s_1) \leq \sigma(1 + \sqrt{2\log(MSAH/\delta)}) \cdot \mathbb{E}_{\pi^*}\left[\sum_{h=1}^H \|g(s_h, a_h; W_0)\|_{\Lambda_h^{-1}}\right] + \tilde{\mathcal{O}}(\frac{1}{K'})$$

*where $\Lambda_h := \lambda I_{md} + \sum_{k \in \mathcal{I}_h} g(s_h^k, a_h^k; W_0)g(s_h^k, a_h^k; W_0)^T \in \mathbb{R}^{md \times md}$.*

**Remark 2.** *Theorem 1 shows that the randomized design in our proposed algorithm yields a provable uncertainty quantifier even though we do not explicitly maintain any confidence regions in the algorithm. The implicit pessimism via perturbed rewards introduces an extra factor of $1 + \sqrt{2\log(MSAH/\delta)}$ into the confidence parameter $\beta$.*

We build upon Theorem 1 to obtain an explicit bound using the following data coverage assumption.

**Assumption 5.2** (Optimal-Policy Concentrability). *$\exists \kappa < \infty$, $\sup_{(h, s_h, a_h)} \frac{d_h^*(s_h, a_h)}{d_h^\mu(s_h, a_h)} \leq \kappa$.*

---

[3]Note that this is the worst-case bound, and the effective dimension can be significantly smaller in practice.

[4]We consider $V : \mathcal{S} \to [0, H+1]$ instead of $V : \mathcal{S} \to [0, H]$ due to the cutoff margin $\psi$ in Algorithm 1.

| work | bound | i.i.d? | explorative data? | finite spectrum? | matrix inverse? | opt |
|------|-------|--------|-------------------|------------------|-----------------|-----|
| Jin et al. (2021) | $\tilde{\mathcal{O}}\left(\frac{d_{lin}^{3/2}H^2}{\sqrt{K}}\right)$ | no | yes | yes | yes | analytical |
| Yang et al. (2020) | $\tilde{\mathcal{O}}\left(\frac{H^2\sqrt{\tilde{d}^2+\tilde{d}\tilde{n}}}{\sqrt{K}}\right)$ | no | – | no | yes | oracle |
| Xu & Liang (2022) | $\tilde{\mathcal{O}}\left(\frac{\tilde{d}H^2}{\sqrt{K}}\right)$ | yes | yes | yes | yes | oracle |
| **This work** | $\tilde{\mathcal{O}}\left(\frac{\kappa H^{5/2}\tilde{d}}{\sqrt{K}}\right)$ | **no** | **no** | **no** | **no** | **GD** |

Table 1: State-of-the-art results for offline RL with function approximation. The third and the fourth columns ask if the corresponding result needs the data to be i.i.d, and well-explored, respectively; the fifth column asks if the induced RKHS needs to have a finite spectrum; the sixth column asks if the algorithm needs to invert a covariance matrix and the last column presents the optimizer being used. Here $\tilde{n}$ is the log of the covering number.

Assumption 5.2 requires any positive-probability trajectory induced by the optimal policy to be covered by the behavior policy. This data coverage assumption is significantly milder than the uniform coverage assumptions in many FQI-based offline RL algorithms (Munos & Szepesvári, 2008; Chen & Jiang, 2019; Nguyen-Tang et al., 2022b) and is common in pessimism-based algorithms (Rashidinejad et al., 2021; Nguyen-Tang et al., 2022a; Chen & Jiang, 2022; Zhan et al., 2022).

**Theorem 2.** *For the same parameter settings and the same assumption as in Theorem 1, we have that with probability at least* $1 - MHm^{-2} - 5\delta$,

$$\text{SubOpt}(\tilde{\pi}) \leq \frac{2\tilde{\sigma}\kappa H}{\sqrt{K'}}\left(\sqrt{2\tilde{d}\log(1+K'/\lambda)+1}+\sqrt{\frac{\log\frac{H}{\delta}}{\lambda}}\right) + \frac{16H}{3K'}\log\frac{\log_2(K'H)}{\delta} + \tilde{\mathcal{O}}(\frac{1}{K'}),$$

*where* $\tilde{\sigma} := \sigma(1+\sqrt{2\log(SAH/\delta)})$.

**Remark 3.** *Theorem 2 shows that with appropriate parameter choice, VIPeR achieves a sub-optimality of* $\tilde{\mathcal{O}}\left(\frac{\kappa H^{3/2}\sqrt{\tilde{d}}\cdot\max\{B, H\sqrt{\tilde{d}}\}}{\sqrt{K}}\right)$. *Compared to Yang et al. (2020), we improve by a factor of* $K^{\frac{2}{d\gamma-1}}$ *for some* $\gamma \in (0,1)$ *at the expense of* $\sqrt{H}$. *When realized to a linear MDP in* $\mathbb{R}^{d_{lin}}$, $\tilde{d} = d_{lin}$ *and our bound reduces into* $\tilde{\mathcal{O}}\left(\frac{\kappa H^{5/2}d_{lin}}{\sqrt{K}}\right)$ *which improves the bound* $\tilde{\mathcal{O}}(d_{lin}^{3/2}H^2/\sqrt{K})$ *of PEVI (Jin et al., 2021, Corollary 4.6) by a factor of* $\sqrt{d_{lin}}$. *We provide the result summary and comparison in Table 1 and give a more detailed discussion in Subsection B.1.*

## 6 EXPERIMENTS

In this section, we empirically evaluate the proposed algorithm VIPeR against several state-of-the-art baselines, including (a) PEVI (Jin et al., 2021), which explicitly constructs lower confidence bound (LCB) for pessimism in a linear model (thus, we rename this algorithm as LinLCB for convenience in our experiments); (b) NeuraLCB (Nguyen-Tang et al., 2022a) which explicitly constructs an LCB using neural network gradients; (c) NeuraLCB (Diag), which is NeuraLCB with a diagonal approximation for estimating the confidence set as suggested in NeuraLCB (Nguyen-Tang et al., 2022a); (d) Lin-VIPeR which is VIPeR realized to the linear function approximation instead of neural network function approximation; (e) NeuralGreedy (LinGreedy, respectively) which uses neural networks (linear models, respectively) to fit the offline data and act greedily with respect to the estimated state-action value functions without any pessimism. Note that when the parametric class, $\mathcal{F}$, in Algorithm 1 is that of neural networks, we refer to VIPeR as Neural-VIPeR. We do not utilize data splitting in the experiments. We provide further algorithmic details of the baselines in Section H.

We evaluate all algorithms in two problem settings: (1) the underlying MDP is a linear MDP whose reward functions and transition kernels are linear in some known feature map (Jin et al., 2020), and (2) the underlying MDP is non-linear with horizon length $H = 1$ (i.e., non-linear contextual bandits) (Zhou et al., 2020), where the reward function is either synthetic or constructed from MNIST

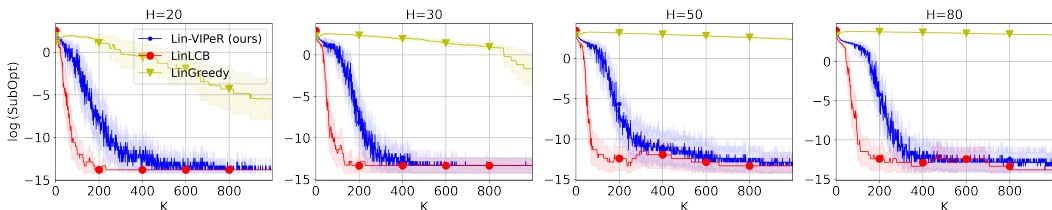

Figure 2: Empirical results of sub-optimality (in log scale) on linear MDPs.

dataset (LeCun et al., 1998). We also evaluate (a variant of) our algorithm and show its strong performance advantage in the D4RL benchmark (Fu et al., 2020) in Section A.3. We implemented all algorithms in Pytorch (Paszke et al., 2019) on a server with Intel(R) Xeon(R) Gold 6248 CPU @ 2.50GHz, 755G RAM, and one NVIDIA Tesla V100 Volta GPU Accelerator 32GB Graphics Card.[5]

## 6.1 LINEAR MDPs

We first test the effectiveness of pessimism implicit in VIPeR (Algorithm 1). To that end, we construct a hard instance of linear MDPs (Yin et al., 2022; Min et al., 2021); due to page limitation, we defer the details of our construction to Section A.1. We test for different values of $H \in \{20, 30, 50, 80\}$ and report the sub-optimality of LinLCB, Lin-VIPeR, and LinGreedy, averaged over 30 runs, in Figure 2. We find that LinGreedy, which is uncertainty-agnostic, fails to learn from offline data and has poor performance in terms of sub-optimality when compared to pessimism-based algorithms LinLCB and Lin-VIPeR. Further, LinLCB outperforms Lin-VIPeR when $K$ is smaller than 400, but the performance of the two algorithms matches for larger sample sizes. Unlike LinLCB, Lin-VIPeR does not construct any confidence regions or require computing and inverting large (covariance) matrices. The Y-axis is in log scale; thus, Lin-VIPeR already has small sub-optimality in the first $K \approx 400$ samples. These show the effectiveness of the randomized design for pessimism implicit in Algorithm 1.

## 6.2 NEURAL CONTEXTUAL BANDITS

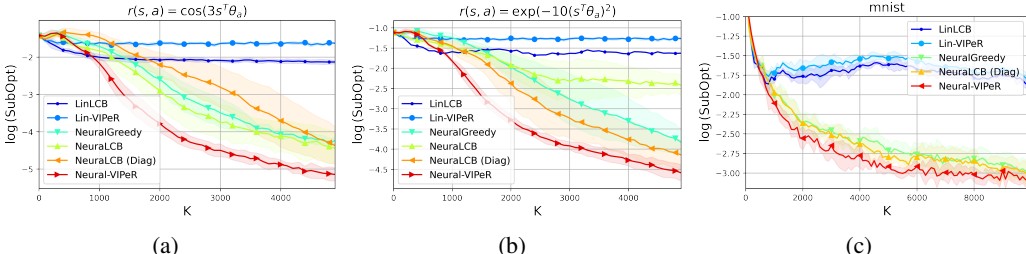

| (a) | (b) | (c) |

Figure 3: Sub-optimality (on log-scale) vs. sample size (K) for neural contextual bandits with following reward functions: (a) $r(s, a) = \cos(3s^T\theta_a)$, (b) $r(s, a) = \exp(-10(s^T\theta_a)^2)$, and (c) MNIST.

Next, we compare the performance and computational efficiency of various algorithms against VIPeR when neural networks are employed. For simplicity, we consider contextual bandits, a special case of MDPs with horizon $H = 1$. Following Zhou et al. (2020); Nguyen-Tang et al. (2022a), we use the bandit problems specified by the following reward functions: (a) $r(s, a) = \cos(3s^T\theta_a)$; (b) $r(s, a) = \exp(-10(s^T\theta_a)^2)$, where $s$ and $\theta_a$ are generated uniformly at random from the unit sphere $\mathbb{S}_{d-1}$ with $d = 16$ and $A = 10$; (c) MNIST, where $r(s, a) = 1$ if $a$ is the true label of the input image $s$ and $r(s, a) = 0$, otherwise. To predict the value of different actions from the same state $s$ using neural networks, we transform a state $s \in \mathbb{R}^d$ into $dA$-dimensional vectors $s^{(1)} = (s, 0, \ldots, 0), s^{(2)} = (0, s, 0, \ldots, 0), \ldots, s^{(A)} = (0, \ldots, 0, s)$ and train the network to map $s^{(a)}$ to $r(s, a)$ given a pair of data $(s, a)$. For Neural-VIPeR, NeuralGreedy, NeuraLCB, and NeuraLCB (Diag), we use the same neural network architecture with two hidden layers of width $m = 64$ and train the network with Adam optimizer (Kingma & Ba, 2015). Due to page limitations, we defer other experimental details and hyperparameter setting to Section A.2. We report the

---

[5]Our code is available here: `https://github.com/thanhnguyentang/neural-offline-rl`.

sub-optimality averaged over 5 runs in Figure 3. We see that algorithms that use a linear model, i.e., LinLCB and Lin-VIPeR significantly underperform neural-based algorithms, i.e., NeuralGreedy, NeuraLCB, NeuraLCB (Diag) and Neural-VIPeR, attesting to the crucial role neural representations play in RL for non-linear problems. It is also interesting to observe from the experimental results that NeuraLCB does not always outperform its diagonal approximation, NeuraLCB (Diag) (e.g., in Figure 3(b)), putting a question mark on the empirical effectiveness of NTK-based uncertainty for offline RL. Finally, Neural-VIPeR outperforms all algorithms in the tested benchmarks, suggesting the effectiveness of our randomized design with neural function approximation.

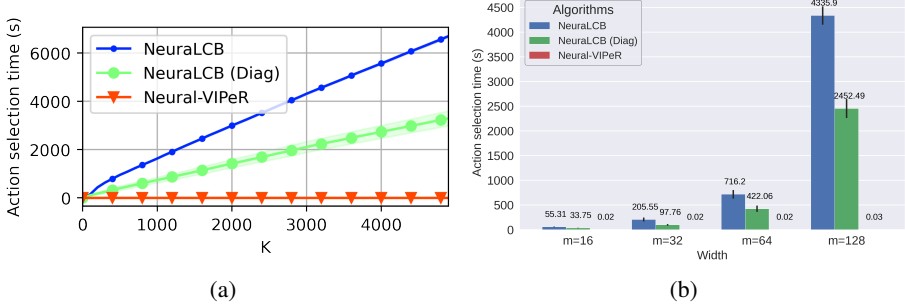

(a)          (b)

Figure 4: Elapsed time (in seconds) for action selection in the contextual bandits problem with $r(s,a) = 10(s^T\theta_a)^2$: (a) Runtime of action selection versus the number of (offline) data points $K$, and (b) runtime of action selection versus the network width $m$ (for $K = 500$).

Figure 4 shows the average runtime for action selection of neural-based algorithms NeuraLCB, NeuraLCB (Diag), and Neural-VIPeR. We observe that algorithms that use explicit confidence regions, i.e., NeuraLCB and NeuraLCB (Diag), take significant time selecting an action when either the number of offline samples $K$ or the network width $m$ increases. This is perhaps not surprising because NeuraLCB and NeuraLCB (Diag) need to compute the inverse of a large covariance matrix to sample an action and maintain the confidence region for each action per state. The diagonal approximation significantly reduces the runtime of NeuraLCB, but the runtime still scales with the number of samples and the network width. In comparison, the runtime for action selection for

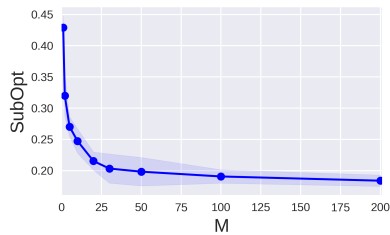

Figure 5: Sub-optimality of Neural-VIPeR versus different values of $M$.

Neural-VIPeR is constant. Since NeuraLCB, NeuraLCB (Diag), and Neural-VIPeR use the same neural network architecture, the runtime spent training one model is similar. The only difference is that Neural-VIPeR trains $M$ models while NeuraLCB and NeuraLCB (Diag) train a single model. However, as the perturbed data in Algorithm 1 are independent, training $M$ models in Neural-VIPeR is embarrassingly parallelizable.

Finally, in Figure 5, we study the effect of the ensemble size on the performance of Neural-VIPeR. We use different values of $M \in \{1, 2, 5, 10, 20, 30, 50, 100, 200\}$ for sample size $K = 1000$. We find that the sub-optimality of Neural-VIPeR decreases graciously as $M$ increases. Indeed, the grid search from the previous experiment in Figure 3 also yields $M = 10$ and $M = 20$ from the search space $M \in \{1, 10, 20\}$ as the best result. This suggests that the ensemble size can also play an important role as a hyperparameter that can determine the amount of pessimism needed in a practical setting.

## 7 CONCLUSION

We propose a novel algorithmic approach for offline RL that involves randomly perturbing value functions and pessimism. Our algorithm eliminates the computational overhead of explicitly maintaining a valid confidence region and computing the inverse of a large covariance matrix for pessimism. We bound the suboptimality of the proposed algorithm as $\tilde{\mathcal{O}}\big(\kappa H^{5/2}\tilde{d}/\sqrt{K}\big)$. We support our theoretical claims of computational efficiency and the effectiveness of our algorithm with extensive experiments.

ACKNOWLEDGEMENTS

This research was supported, in part, by DARPA GARD award HR00112020004, NSF CAREER award IIS-1943251, an award from the Institute of Assured Autonomy, and Spring 2022 workshop on "Learning and Games" at the Simons Institute for the Theory of Computing.

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

# A EXPERIMENT DETAILS

## A.1 LINEAR MDPs

In this subsection, we provide further details to the experiment setup used in Subsection 6.1. We describe in detail a variant of the hard instance of linear MDPs (Yin et al., 2022) used in our experiment. The linear MDP has $\mathcal{S} = \{0, 1\}$, $\mathcal{A} = \{0, 1, \cdots, 99\}$, and the feature dimension $d = 10$. Each action $a \in [99] = \{1, \ldots, 99\}$ is represented by its binary encoding vector $u_a \in \mathbb{R}^8$ with entry being either $-1$ or $1$. The feature mapping $\phi(s, a)$ is given by $\phi(s, a) = [u_a^T, \delta(s, a), 1 - \delta(s, a)]^T \in \mathbb{R}^{10}$, where $\delta(s, a) = 1$ if $(s, a) = (0, 0)$ and $\delta(s, a) = 0$ otherwise. The true measure $\nu_h(s)$ is given by $\nu_h(s) = [0, \cdots, 0, (1 - s) \oplus \alpha_h, s \oplus \alpha_h]$ where $\{\alpha_h\}_{h \in [H]} \in \{0, 1\}^H$ are generated uniformly at random and $\oplus$ is the XOR operator. We define $\theta_h = [0, \cdots, 0, r, 1 - r]^T \in \mathbb{R}^{10}$ where $r = 0.99$. Recall that the transition follows $\mathbb{P}_h(s'|s, a) = \langle \phi(s, a), \nu_h(s') \rangle$ and the mean reward $r_h(s, a) = \langle \phi(s, a), \theta_h \rangle$. We generated a priori $K \in \{1, \ldots, 1000\}$ trajectories using the behavior policy $\mu$, where for any $h \in [H]$ we set $\mu_h(0|0) = p, \mu_h(1|0) = 1 - p, \mu_h(a|0) = 0, \forall a > 1; \mu_h(0|1) = p, \mu_h(a|1) = (1 - p)/99, \forall a > 0$, where we set $p = 0.6$.

We run over $K \in \{1, \ldots, 1000\}$ and $H \in \{20, 30, 50, 80\}$. We set $\lambda = 0.01$ for all algorithms. For Lin-VIPeR, we grid searched $\sigma_h = \sigma \in \{0.0, 0.1, 0.5, 1.0, 2.0\}$ and $M \in \{1, 2, 10, 20\}$. For LinLCB, we grid searched its uncertainty multiplier $\beta \in \{0.1, 0.5, 1, 2\}$. The sub-optimality metric is used to compare algorithms. For each $H \in \{20, 30, 50, 80\}$, each algorithm was executed for 30 times and the averaged results (with std) are reported in Figure 2.

## A.2 NEURAL CONTEXTUAL BANDITS

In this subsection, we provide in detail the experimental and hyperparameter setup in our experiment in Subsection 6.2. For Neural-VIPeR, NeuralGreedy, NeuraLCB and NeuraLCB (Diag), we use the same neural network architecture with two hidden layers whose width $m = 64$, train the network with Adam optimizer (Kingma & Ba, 2015) with learning rate being grid-searched over $\{0.0001, 0.001, 0.01\}$ and batch size of 64. For NeuraLCB, NeuraLCB (Diag), and LinLCB, we grid-searched $\beta$ over $\{0.001, 0.01, 0.1, 1, 5, 10\}$. For Neural-VIPeR and Lin-VIPeR, we grid-searched $\sigma_h = \sigma$ over $\{0.001, 0.01, 0.1, 1, 5, 10\}$ and $M$ over $\{1, 10, 20\}$. We did not run NeuraLCB in MNIST as the inverse of a full covariance matrix in this case is extremely expensive. We fixed the regularization parameter $\lambda = 0.01$ for all algorithms. Offline data is generated by the $(1 - \epsilon)$-optimal policy which generates non-optimal actions with probability $\epsilon$ and optimal actions with probability $1 - \epsilon$. We set $\epsilon = 0.5$ in our experiments. To estimate the expected sub-optimality, we randomly obtain $1,000$ novel samples (i.e. not used in training) to compute the average sub-optimality and keep these same samples for all algorithms.

## A.3 EXPERIMENT IN D4RL BENCHMARK

In this subsection, we evaluate the effectiveness of the reward perturbing design of VIPeR in the Gym domain in the D4RL benchmark (Fu et al., 2020). The Gym domain has three environments (HalfCheetah, Hopper, and Walker2d) with five datasets (random, medium, medium-replay, medium-expert, and expert), making up 15 different settings.

**Design.** To adapt the design of VIPeR to continuous control, we use the actor-critic framework. Specifically, we have $M$ critics $\{Q_{\theta^i}\}_{i \in [M]}$ and one actor $\pi_\phi$, where $\{\theta^i\}_{i \in [M]}$ and $\phi$ are the learnable parameters for the critics and actor, respectively. Note that in the continuous domain, we consider discounted MDP with discount factor $\gamma$, instead of finite-time episode MDP as we initially considered in our setting in the main paper. In the presence of the actor $\pi_\phi$, there are two modifications to Algorithm 1. The first modification is that when training the critics $\{Q_\theta^i\}_{i \in [M]}$, we augment the training loss in Algorithm 2 with a new penalization term. Specifically, the critic loss for $Q_{\theta^i}$ on a training sample $\tau := (s, a, r, s')$ (sampled from the offline data $\mathcal{D}$) is

$$\mathcal{L}(\theta^i; \tau) = (Q_{\theta^i}(s, a) - (r + \gamma Q_{\bar{\theta}^i}(s') + \xi))^2 + \beta \underbrace{\mathbb{E}_{a' \sim \pi_\phi(\cdot|s)} \left[ (Q_{\theta^i}(s, a') - \bar{Q}(s, a'))^2 \right]}_{\text{penalization term } R(\theta^i; s, \phi)}, \quad (1)$$

where $\bar{\theta}^i$ has the same value of the current $\theta^i$ but is kept fixed, $\bar{Q} = \frac{1}{M} \sum_{i=1}^{M} Q_{\theta^i}$ and $\xi \sim \mathcal{N}(0, \sigma^2)$ is Gaussian noise, and $\beta$ is a penalization parameter (note that $\beta$ here is totally different from the $\beta$ in Theorem 1). The penalization term $R(\theta^i; s, \phi)$ discourages overestimation in the value function estimate $Q_{\theta^i}$ for out-of-distribution (OOD) actions $a' \sim \pi_\phi(\cdot|s)$. Our design of $R(\theta^i; s, \phi)$ is initially inspired by the OOD penalization in Bai et al. (2022) that creates a pessimistic pseudo target for the values at OOD actions. Note that we do not need any penalization for OOD actions in our experiment for contextual bandits in Section 6.2. This is because in the contextual bandit setting in Section 6.2 the action space is finite and not large, thus the offline data often sufficiently cover all good actions. In the continuous domain such as the Gym domain of D4RL, however, it is almost certain that there are actions that are not covered by the offline data since the action space is continuous. We also note that the inclusion of the OOD action penalization term $R(\theta^i; s, \phi)$ in this experiment does not contradict our guarantee in Theorem 1 since in the theorem we consider finite action space while in this experiment we consider continuous action space. We argue that the inclusion of some regularization for OOD actions (e.g., $R(\theta^i; s, \phi)$) is necessary for the continuous domain. [6]

The second modification to Algorithm 1 for the continuous domain is the actor training, which is the implementation of policy extraction in line 10 of Algorithm 1. Specifically, to train the actor $\pi_\phi$ given the ensemble of critics $\{Q_\theta^i\}_{i \in [M]}$, we use soft actor update in Haarnoja et al. (2018) via

$$\max_\phi \left\{ \mathbb{E}_{s \sim \mathcal{D}, a' \sim \pi_\phi(\cdot|s)} \left[ \min_{i \in [M]} Q_{\theta^i}(s, a') - \log \pi_\phi(a'|s) \right] \right\}, \tag{2}$$

which is trained using gradient ascent in practice. Note that in the discrete action domain, we do not need such actor training as we can efficiently extract the greedy policy with respect to the estimated action-value functions when the action space is finite. Also note that we do not use data splitting and value truncation as in the original design of Algorithm 1.

**Hyperparameters.** For the hyper-parameters of our training, we set $M = 10$ and the noise variance $\sigma = 0.01$. For $\beta$, we decrease it from $0.5$ to $0.2$ by linear decay for the first 50K steps and exponential decay for the remaining steps. For the other hyperparameters of actor-critic training, we fix them the same as in Bai et al. (2022). Specifically, the Q-network is the fully connected neural network with three hidden layers all of which has 256 neurons. The learning rate for the actor and the critic are $10^{-4}$ and $3 \times 10^{-4}$, respectively. The optimizer is Adam.

**Results.** We compare VIPeR with several state-of-the-art algorithms, including (i) BEAR (Kumar et al., 2019) that use MMD distance to constraint policy to the offline data, (ii) UWAC (Wu et al., 2021) that improves BEAR using dropout uncertainty, (iii) CQL (Kumar et al., 2020) that minimizes Q-values of OOD actions, (iv) MOPO (Yu et al., 2020) that uses model-based uncertainty via ensemble dynamics, (v) TD3-BC (Fujimoto & Gu, 2021) that uses adaptive behavior cloning, and (vi) PBRL (Bai et al., 2022) that use uncertainty quantification via disagreement of bootstrapped Q-functions. We follow the evaluation protocol in Bai et al. (2022). We run our algorithm for five seeds and report the average final evaluation scores with standard deviation. We report the scores of our method and the baselines in Table 2. We can see that our method has a strong advantage of good performance (highest scores) in 11 out of 15 settings, and has good stability (small std) in all settings. Overall, we also have the strongest average scores aggregated over all settings.

## B  EXTENDED DISCUSSION

Here we provide extended discussion of our result.

### B.1  COMPARISON WITH OTHER WORKS AND DISCUSSION

We provide further discussion regarding comparison with other works in the literature.

---

[6] In our experiment, we also observe that without this penalization term, the method struggles to learn any good policy. However, using only the penalization term without the first term in Eq. (1), we observe that the method cannot learn either.

| | | BEAR | UWAC | CQL | MOPO | TD3-BC | PBRL | VIPeR |
|---|---|---|---|---|---|---|---|---|
| Random | HalfCheetah | 2.3 ±0.0 | 2.3 ±0.0 | 17.5 ±1.5 | **35.9** ±2.9 | 11.0 ±1.1 | 11.0 ±5.8 | 14.5 ±2.1 |
| | Hopper | 3.9 ±2.3 | 2.7 ±0.3 | 7.9 ±0.4 | 16.7 ±12.2 | 8.5 ±0.6 | 26.8 ±9.3 | **31.4** ±0.0 |
| | Walker2d | 12.8 ±10.2 | 2.0 ±0.4 | 5.1 ±1.3 | 4.2 ±5.7 | 1.6 ±1.7 | 8.1 ±4.4 | **20.5** ±0.5 |
| Medium | HalfCheetah | 43.0 ±0.2 | 42.2 ±0.4 | 47.0 ±0.5 | **73.1** ±2.4 | 48.3 ±0.3 | 57.9 ±1.5 | 58.5 ±1.1 |
| | Hopper | 51.8 ±4.0 | 50.9 ±4.4 | 53.0 ±28.5 | 38.3 ±34.9 | 59.3 ±4.2 | 75.3 ±31.2 | **99.4** ±6.2 |
| | Walker2d | -0.2 ±0.1 | 75.4 ±3.0 | 73.3 ±17.7 | 41.2 ±30.8 | 83.7 ±2.1 | **89.6** ±0.7 | 89.6 ±1.2 |
| Medium Replay | HalfCheetah | 36.3 ±3.1 | 35.9 ±3.7 | 45.5 ±0.7 | **69.2** ±1.1 | 44.6 ±0.5 | 45.1 ±8.0 | 45.0 ±8.6 |
| | Hopper | 52.2 ±19.3 | 25.3 ±1.7 | 88.7 ±12.9 | 32.7 ±9.4 | 60.9 ±18.8 | **100.6** ±1.0 | 100.2 ±1.0 |
| | Walker2d | 7.0 ±7.8 | 23.6 ±6.9 | 81.8 ±2.7 | 73.7 ±9.4 | 81.8 ±5.5 | 77.7 ±14.5 | **83.1** ±4.2 |
| Medium Expert | HalfCheetah | 46.0 ±4.7 | 42.7 ±0.3 | 75.6 ±25.7 | 70.3 ±21.9 | 90.7 ±4.3 | 92.3 ±1.1 | **94.2** ±1.2 |
| | Hopper | 50.6 ±25.3 | 44.9 ±8.1 | 105.6 ±12.9 | 60.6 ±32.5 | 98.0 ±9.4 | **110.8** ±0.8 | 110.6 ±1.0 |
| | Walker2d | 22.1 ±44.9 | 96.5 ±9.1 | 107.9 ±1.6 | 77.4 ±27.9 | **110.1** ±0.5 | 110.1 ±0.3 | 109.8 ±0.5 |
| Expert | HalfCheetah | 92.7 ±0.6 | 92.9 ±0.6 | 96.3 ±1.3 | 81.3 ±21.8 | 96.7 ±1.1 | 92.4 ±1.7 | **97.4** ±0.9 |
| | Hopper | 54.6 ±21.0 | 110.5 ±0.5 | 96.5 ±28.0 | 62.5 ±29.0 | 107.8 ±7 | 110.5 ±0.4 | **110.8** ±0.4 |
| | Walker2d | 106.6 ±6.8 | 108.4 ±0.4 | 108.5 ±0.5 | 62.4 ±3.2 | **110.2** ±0.3 | 108.3 ±0.3 | 108.3 ±0.2 |
| | Average | 38.78 ±10.0 | 50.41 ±2.7 | 67.35 ±9.1 | 53.3 ±16.3 | 67.55 ±3.8 | 74.37 ±5.3 | **78.2** ±1.9 |

Table 2: Average normalized score and standard deviation of all algorithms over five seeds in the Gym domain in the "v2" dataset of D4RL (Fu et al., 2020). The scores for all the baselines are from Table 1 of Bai et al. (2022). The highest scores are highlighted.

**Comparing to Jin et al. (2021).** When the underlying MDP reduces into a linear MDP, if we use the linear model as the plug-in parametric model in Algorithm 1, our bound reduces into $\tilde{\mathcal{O}}\left(\frac{\kappa H^{5/2} d_{lin}}{\sqrt{K}}\right)$ which improves the bound $\tilde{\mathcal{O}}(d_{lin}^{3/2} H^2/\sqrt{K})$ of PEVI (Jin et al., 2021, Corollary 4.6) by a factor of $\sqrt{d_{lin}}$ and worsen by a factor of $\sqrt{H}$ due to the data splitting. Thus, our bound is more favorable in the linear MDPs with high-dimensional features. Moreover, our bound is guaranteed in more practical scenarios where the offline data can have been adaptively generated and is not required to uniformly cover the state-action space. The explicit bound $\tilde{\mathcal{O}}(d_{lin}^{3/2} H^2/\sqrt{K})$ of PEVI (Jin et al., 2021, Corollary 4.6) is obtained under the assumption that the offline data have uniform coverage and are generated independently on the episode basis.

**Comparing to Yang et al. (2020).** Though Yang et al. (2020) work in the online regime, it shares some part of the literature with our work in function approximation for RL. Besides different learning regimes (offline versus online), we offer three key distinctions which can potentially be used in the online regime as well: (i) perturbed rewards, (ii) optimization, and (iii) data split. Regarding (i), our perturbed reward design can be applied to online RL with function approximation to obtain a provably efficient online RL that is computationally efficient and thus remove the need of maintaining explicit confidence regions and performing the inverse of a large covariance matrix. Regarding (ii), we incorporate the optimization analysis into our algorithm which makes our algorithm and analysis more practical. We also note that unlike (Yang et al., 2020), we do not make any assumption on the eigenvalue decay rate of the empirical NTK kernel as the empirical NTK kernel is data-dependent. Regarding (iii), our data split technique completely removes the factor $\sqrt{\log \mathcal{N}_\infty(\mathcal{H}, 1/K, B)}$ in the bound at the expense of increasing the bound by a factor of $\sqrt{H}$. In complex models, such log covering number can be excessively larger than the horizon $H$, making the algorithm too optimistic in the online regime (optimistic in the offline regime, respectively). For example, the target function class is RKHS with a $\gamma$-polynomial decay, the log covering number scales as (Yang et al., 2020, Lemma D1),

$$\sqrt{\log \mathcal{N}_\infty(\mathcal{H}, 1/K, B)} \lesssim K^{\frac{2}{\alpha\gamma-1}},$$

for some $\alpha \in (0, 1)$. In the case of two-layer ReLU NTK, $\gamma = d$ (Bietti & Mairal, 2019), thus $\sqrt{\log \mathcal{N}_\infty(\mathcal{H}, 1/K, B)} \lesssim K^{\frac{2}{\alpha d-1}}$ which is much larger than $\sqrt{H}$ when the size of dataset is large. Note that our data-splitting technique is general that can be used in the online regime as well.

**Comparing to Xu & Liang (2022).** Xu & Liang (2022) consider a different setting where per-timestep rewards are not available and only the total reward of the whole trajectory is given. Used

with neural function approximation, they obtain $\tilde{\mathcal{O}}(D_{\text{eff}}H^2/\sqrt{K})$ where $D_{\text{eff}}$ is their effective dimension. Note that Xu & Liang (2022) do not use data splitting and still achieve the same order of $D_{\text{eff}}$ as our result with data splitting. It at first might appear that our bound is inferior to their bound as we pay the cost of $\sqrt{H}$ due to data splitting. However, to obtain that bound, they make three critical assumptions: (i) the offline data trajectories are independently and identically distributed (i.i.d.) (see their Assumption 3), (ii) the offline data is uniformly explorative over all dimensions of the feature space (also see their Assumption 3), and (iii) the eigenfunctions of the induced NTK RKHS has finite spectrum (see their Assumption 4). The i.i.d. assumption under the RKHS space with finite dimensions (due to the finite spectrum assumption) and the well-explored dataset is critical in their proof to use a matrix concentration that does not incur an extra factor of $\sqrt{D_{\text{eff}}}$ as it would normally do without these assumptions (see Section E, the proof of their Lemma 2). Note that the celebrated ReLU NTK does not satisfy the finite spectrum assumption (Bietti & Mairal, 2019). Moreover, we do not make any of these three assumptions above for our bound to hold. That suggests that our bound is much more general. In addition, we do not need to compute any confidence regions nor perform the inverse of a large covariance matrix.

**Comparing to Yin et al. (2023).**    During the submission of our work, a concurrent work of Yin et al. (2023) appeared online. Yin et al. (2023) study provably efficient offline RL with a general parametric function approximation that unifies the guarantees of offline RL in linear and generalized linear MDPs, and beyond with potential applications to other classes of functions in practice. We remark that the result in Yin et al. (2023) is orthogonal/complementary to our paper since they consider the parametric class with third-time differentiability which cannot apply to neural networks (not necessarily overparameterized) with non-smooth activation such as ReLU. In addition, they do not consider reward perturbing in their algorithmic design or optimization errors in their analysis.

## B.2   Worse-Case Rate of Effective Dimension

In the main paper, we prove an $\tilde{\mathcal{O}}\left(\frac{\kappa H^{5/2}\tilde{d}}{\sqrt{K}}\right)$ sub-optimality bound which depends on the notion of effective dimension defined in Definition 2. Here we give a worst-case rate of the effective dimension $\tilde{d}$ for the two-layer ReLU NTK. We first briefly review the background of RKHS.

Let $\mathcal{H}$ be an RKHS defined on $\mathcal{X} \subseteq \mathbb{R}^d$ with kernel function $\rho : \mathcal{X} \times \mathcal{X} \to \mathbb{R}$. Let $\langle \cdot, \cdot \rangle_{\mathcal{H}} : \mathcal{H} \times \mathcal{H} \to \mathbb{R}$ and $\|\cdot\|_{\mathcal{H}} : \mathcal{H} \to \mathbb{R}$ be the inner product and the RKSH norm on $\mathcal{H}$. By the reproducing kernel property of $\mathcal{H}$, there exists a feature mapping $\phi : \mathcal{X} \to \mathcal{H}$ such that $f(x) = \langle f, \phi(x) \rangle_{\mathcal{H}}$ and $\rho(x, x') = \langle \phi(x), \phi(x') \rangle_{\mathcal{H}}$. We assume that the kernel function $\rho$ is uniformly bounded, i.e. $\sup_{x \in \mathcal{X}} \rho(x, x) < \infty$. Let $\mathcal{L}^2(\mathcal{X})$ be the space of square-integral functions on $\mathcal{X}$ with respect to the Lebesgue measure and let $\langle \cdot, \cdot \rangle_{\mathcal{L}^2}$ be the inner product on $\mathcal{L}^2(\mathcal{X})$. The kernel function $\rho$ induces an integral operator $T_\rho : \mathcal{L}^2(\mathcal{X}) \to \mathcal{L}^2(\mathcal{X})$ defined as

$$T_\rho f(x) = \int_{\mathcal{X}} \rho(x, x')f(x')dx'.$$

By Mercer's theorem (Steinwart & Christmann, 2008), $T_\rho$ has countable and positive eigenvalues $\{\lambda_i\}_{i \geq 1}$ and eigenfunctions $\{\nu_i\}_{i \geq 1}$. The kernel function and $\mathcal{H}$ can be expressed as

$$\rho(x, x') = \sum_{i=1}^{\infty} \lambda_i \nu_i(x)\nu_i(x'),$$

$$\mathcal{H} = \{f \in \mathcal{L}^2(\mathcal{X}) : \sum_{i=1}^{\infty} \frac{\langle f, \nu_i \rangle_{\mathcal{L}^2}}{\lambda_i} < \infty\}.$$

Now consider the NTK defined in Definition 1:

$$K_{ntk}(x, x') = \mathbb{E}_{w \sim \mathcal{N}(0, I_d/d)} \langle x\sigma'(w^T x), x'\sigma'(w^T x') \rangle.$$

It follows from (Bietti & Mairal, 2019, Proposition 1) that $\lambda_i \asymp i^{-d}$. Thus, by (Srinivas et al., 2010, Theorem 5), the data-dependent effective dimension of $\mathcal{H}_{ntk}$ can be bounded in the worst case by

$$\tilde{d} \lesssim K'^{(d+1)/(2d)}.$$

We remark that this is the worst-case bound that considers uniformly over all possible realizable of training data. The effective dimension $\tilde{d}$ is on the other hand data-dependent, i.e. its value depends on the specific training data at hand thus $\tilde{d}$ can be actually much smaller than the worst-case rate.

## C   PROOF OF THEOREM 1 AND THEOREM 2

In this section, we provide both the outline and detailed proofs of Theorem 1 and Theorem 2.

### C.1   TECHNICAL REVIEW AND PROOF OVERVIEW

**Technical Review.**   In what follows, we provide more detailed discussion when placing our technical contribution in the context of the related literature. Our technical result starts with the value difference lemma in Jin et al. (2021) to connect bounding the suboptimality of an offline algorithm to controlling the uncertainty quantification in the value estimates. Thus, our key technical contribution is to provably quantify the uncertainty of the perturbed value function estimates which were obtained via reward perturbing and gradient descent. This problem setting is largely different from the current analysis of overparameterized neural networks for supervised learning which does not require uncertainty quantification.

Our work is not the first to consider uncertainty quantification with overparameterized neural networks, since it has been studied in Zhou et al. (2020); Nguyen-Tang et al. (2022a); Jia et al. (2022). However, there are significant technical differences between our work and these works. The work in Zhou et al. (2020); Nguyen-Tang et al. (2022a) considers contextual bandits with overparameterized neural networks trained by (S)GD and quantifies the uncertainty of the value function with explicit empirical covariance matrices. We consider general MDP and use reward perturbing to implicitly obtain uncertainty, thus requiring different proof techniques.

Jia et al. (2022) is more related to our work since they consider reward perturbing with overparameterized neural networks (but they consider contextual bandits). However, our reward perturbing strategy is largely different from that in Jia et al. (2022). Specifically, Jia et al. (2022) perturbs each reward only once while we perturb each reward multiple times, where the number of perturbing times is crucial in our work and needs to be controlled carefully. We show in Theorem 1 that our reward perturbing strategy is effective in enforcing sufficient pessimism for offline learning in general MDP and the empirical results in Figure 2, Figure 3, Figure 5, and Table 2 are strongly consistent with our theoretical suggestion. Thus, our technical proofs are largely different from those of Jia et al. (2022).

Finally, the idea of perturbing rewards multiple times in our algorithm is inspired by Ishfaq et al. (2021). However, Ishfaq et al. (2021) consider reward perturbing for obtaining optimism in online RL. While perturbing rewards are intuitive to obtain optimism for online RL, for offline RL, under distributional shift, it can be paradoxically difficult to properly obtain pessimism with randomization and ensemble (Ghasemipour et al., 2022), especially with neural function approximation. We show affirmatively in our work that simply taking the minimum of the randomized value functions after perturbing rewards multiple times is sufficient to obtain provable pessimism for offline RL. In addition, Ishfaq et al. (2021) do not consider neural network function approximation and optimization. Controlling the uncertainty of randomization (via reward perturbing) under neural networks with extra optimization errors induced by gradient descent sets our technical proof significantly apart from that of Ishfaq et al. (2021).

Besides all these differences, in this work, we propose an intricately-designed data splitting technique that avoids the uniform convergence argument and could be of independent interest for studying sample-efficient RL with complex function approximation.

**Proof Overview.**   The key steps for proving Theorem 1 and Theorem 2 are highlighted in Subsection C.2 and Subsection C.3, respectively. Here, we discuss an overview of our proof strategy. The key technical challenge in our proof is to quantify the uncertainty of the perturbed value function estimates. To deal with this, we carefully control both the near-linearity of neural networks in the NTK regime and the estimation error induced by reward perturbing. A key result that we use to control the linear approximation to the value function estimates is Lemma D.3. The technical challenge

| Parameters | Meaning/Expression |
|---|---|
| $m$ | Network width |
| $\lambda$ | Regularization parameter |
| $\eta$ | Learning rate |
| $M$ | Number of bootstraps |
| $\{\sigma_h\}_{h\in[H]}$ | Noise variances |
| $J$ | Number of GD steps |
| $\psi$ | Cutoff margin |
| $K$ | Number of offline episodes |
| $R$ | Radius parameter |
| $\delta$ | Failure level |
| $K'$ | bucket size, $K/H$ |
| $\mathcal{I}_h$ | index buckets, $[(H-h)K'+1, (H-h)K'+2, \ldots, (H-h+1)K']$ |
| $B$ | Parameter radius of the Bellman operator |
| $\gamma_{h,1}$ | $c_1\sigma_h\sqrt{\log(KM/\delta)}$ |
| $\gamma_{h,2}$ | $c_2\sigma_h\sqrt{d\log(dKM/\delta)}$ |
| $B_1$ | $\lambda^{-1}\sqrt{2K(H+\psi)^2 + 8C_gR^{4/3}m^{-1/6}\sqrt{\log m}}\sqrt{K}C_gR^{1/3}m^{-1/6}\sqrt{\log m}$ |
| $\tilde{B}_1$ | $\lambda^{-1}\sqrt{2K'(H+\psi+\gamma_{h,1})^2 + \lambda\gamma_{h,2}^2 + 8C_gR^{4/3}m^{-1/6}\sqrt{\log m}}\sqrt{K'}C_gR^{1/3}m^{-1/6}\sqrt{\log m}$ |
| $\tilde{B}_2$ | $\lambda^{-1}K'C_gR^{4/3}m^{-1/6}\sqrt{\log m}$ |
| $\iota_0$ | $Bm^{-1/2}(2\sqrt{d} + \sqrt{2\log(3H/\delta)})$ |
| $\iota_1$ | $C_gR^{4/3}m^{-1/6}\sqrt{\log m} + C_g\left(\tilde{B}_1 + \tilde{B}_2 + \lambda^{-1}(1-\eta\lambda)^J\left(K'(H+\psi+\gamma_{h,1})^2 + \lambda\gamma_{h,2}^2\right)\right)$ |
| $\iota_2$ | $C_gR^{4/3}m^{-1/6}\sqrt{\log m} + C_g\left(B_1 + \tilde{B}_2 + \lambda^{-1}(1-\eta\lambda)^JK'(H+\psi)^2\right)$ |
| $\iota$ | $\iota_0 + \iota_1 + 2\iota_2$ |
| $\beta$ | $\frac{BK'}{\sqrt{m}}(2\sqrt{d} + \sqrt{2\log(3H/\delta)})\lambda^{-1/2}C_g + \lambda^{1/2}B$ $+ (H+\psi)\left[\sqrt{\tilde{d}_h\log(1+\frac{K'}{\lambda})} + K'\log\lambda + 2\log(3H/\delta)\right]$ |

Table 3: The problem parameters and the additional parameters that we introduce for our proofs. Here $c_1$, $c_2$, and $C_g$ are some absolute constants independent of the problem parameters.

in establishing Lemma D.3 is how to carefully control and propagate the optimization error incurred by gradient descent. The complete proof of Lemma D.3 is provided in Section E.3.

The implicit uncertainty quantifier induced by the reward perturbing is established in Lemma D.1 and Lemma D.2, where we carefully design a series of intricate auxiliary loss functions and establish the anti-concentrability of the perturbed value function estimates. This requires a careful design of the variance of the noises injected into the rewards.

To deal with removing a potentially large covering number when we quantify the implicit uncertainty, we propose our data splitting technique which is validated in the proof of Lemma D.1 in Section E.1. Moreover, establishing Lemma D.1 in the overparameterization regime induces an additional challenge since a standard analysis would result in a vacuous bound that scales with the overparameterization. We avoid this issue by carefully incorporating the use of the effective dimension in Lemma D.1.

### C.2  PROOF OF THEOREM 1

In this subsection, we present the proof of Theorem 1. We first decompose the suboptimality $\text{SubOpt}(\tilde{\pi}; s)$ and present the main lemmas to bound the evaluation error and the summation of the implicit confidence terms, respectively. The detailed proof of these lemmas are deferred to Section D. For proof convenience, we first provide the key parameters that we use consistently throughout our proofs in Table 3.

We define the model evaluation error at any $(x, h) \in \mathcal{X} \times [H]$ as

$$\text{err}_h(x) = (\mathbb{B}_h \tilde{V}_{h+1} - \tilde{Q}_h)(x), \tag{3}$$

where $\mathbb{B}_h$ is the Bellman operator defined in Section 3, and $\tilde{V}_h$ and $\tilde{Q}_h$ are the estimated (action-) state value functions returned by Algorithm 1. Using the standard suboptimality decomposition (Jin et al., 2021, Lemma 3.1), for any $s_1 \in \mathcal{S}$,

$$\text{SubOpt}(\tilde{\pi}; s_1) = -\sum_{h=1}^{H} \mathbb{E}_{\tilde{\pi}} \left[ \text{err}_h(s_h, a_h) \right] + \sum_{h=1}^{H} \mathbb{E}_{\pi^*} \left[ \text{err}_h(s_h, a_h) \right]$$

$$+ \sum_{h=1}^{H} \mathbb{E}_{\pi^*} \underbrace{\left[ \langle \tilde{Q}_h(s_h, \cdot), \pi_h^*(\cdot|s_h) - \tilde{\pi}_h(\cdot|s_h) \rangle_{\mathcal{A}} \right]}_{\leq 0},$$

where the third term is non-positive as $\tilde{\pi}_h$ is greedy with respect to $\tilde{Q}_h$. Thus, for any $s_1 \in \mathcal{S}$, we have

$$\text{SubOpt}(\tilde{\pi}; s_1) \leq -\sum_{h=1}^{H} \mathbb{E}_{\tilde{\pi}} \left[ \text{err}_h(s_h, a_h) \right] + \sum_{h=1}^{H} \mathbb{E}_{\pi^*} \left[ \text{err}_h(s_h, a_h) \right]. \tag{4}$$

In the following main lemma, we bound the evaluation error $err_h(s, a)$. In the rest of the proof, we consider an additional parameter $R$ and fix any $\delta \in (0, 1)$.

**Lemma C.1.** *Let*

$$\begin{cases} m = \Omega\left(d^{3/2}R^{-1}\log^{3/2}(\sqrt{m}/R)\right) \\ R = \mathcal{O}\left(m^{1/2}\log^{-3} m\right), \\ m = \Omega\left(K'^{10}(H+\psi)^2\log(3K'H/\delta)\right) \\ \lambda > 1 \\ K'C_g^2 \geq \lambda R \geq \max\{4\tilde{B}_1, 4\tilde{B}_2, 2\sqrt{2\lambda^{-1}K'(H+\psi+\gamma_{h,1})^2 + 4\gamma_{h,2}^2}\}, \\ \eta \leq (\lambda + K'C_g^2)^{-1}, \\ \psi > \iota, \\ \sigma_h \geq \beta, \forall h \in [H], \end{cases} \tag{5}$$

where $\tilde{B}_1$, $\tilde{B}_2$, $\gamma_{h,1}$, $\gamma_{h,2}$, and $\iota$ are defined in Table 3, $C_g$ is a absolute constant given in Lemma G.1, and $R$ is an additional parameter. Let $M = \log\frac{HSA}{\delta} / \log\frac{1}{1-\Phi(-1)}$ where $\Phi(\cdot)$ is the cumulative distribution function of the standard normal distribution. With probability at least $1 - MHm^{-2} - 2\delta$, for any $(x, h) \in \mathcal{X} \times [H]$, we have

$$-\iota \leq \mathrm{err}_h(x) \leq \sigma_h(1 + \sqrt{2\log(MSAH/\delta)}) \cdot \|g(x; W_0)\|_{\Lambda_h^{-1}} + \iota$$

where $\Lambda_h := \lambda I_{md} + \sum_{k \in \mathcal{I}_h} g(x_h^k; W_0)g(x_h^k; W_0)^T \in \mathbb{R}^{md \times md}$.

Now we can prove Theorem 1.

*Proof of Theorem 1.* Theorem 1 can directly follow from substituting Lemma C.1 into Equation (4). We now only need to simplify the conditions in Equation (5). To satisfy Equation (5), it suffices to set

$$\begin{cases} \lambda = 1 + \frac{H}{K} \\ \psi = 1 > \iota \\ \sigma_h = \beta \\ 8C_g R^{4/3} m^{-1/6}\sqrt{\log m} \leq 1 \\ \lambda^{-1}K'H^2 \geq 2 \\ \tilde{B}_1 \leq \sqrt{2K'(H + \psi + \gamma_{h,1})^2 + \lambda\gamma_{h,2}^2 + 1}\sqrt{K'}C_g R^{1/3}m^{-1/6}\sqrt{\log m} \leq 1 \\ \tilde{B}_2 \leq K'C_g R^{4/3}m^{-1/6}\sqrt{\log m} \leq 1. \end{cases}$$

Combining with Equation 5, we have

$$\begin{cases} \lambda = 1 + \frac{H}{K} \\ \psi = 1 > \iota \\ \sigma_h = \beta \\ \eta \lesssim (\lambda + K')^{-1} \\ m \gtrsim \max\left\{ R^8\log^3 m, K'^{10}(H+1)^2\log(3K'H/\delta), d^{3/2}R^{-1}\log^{3/2}(\sqrt{m}/R), K'^6 R^8\log^3 m \right\} \\ m \gtrsim [2K'(H + 1 + \beta\sqrt{\log(K'M/\delta)})^2 + \lambda\beta^2 d\log(dK'M/\delta) + 1]^3 K'^3 R\log^3 m \\ 4\sqrt{K'}(H + 1 + \beta\sqrt{\log(K'M/\delta)}) + 4\beta\sqrt{d\log(dK'M/\delta)} \leq R \lesssim K'. \end{cases}$$
(6)

Note that with the above choice of $\lambda = 1 + \frac{H}{K}$, we have

$$K'\log\lambda = \log(1 + \frac{1}{K'})^{K'} \leq \log 3 < 2.$$

We further set that $m \gtrsim B^2 K'^2 d\log(3H/\delta)$, we have

$$\beta = \frac{BK'}{\sqrt{m}}(2\sqrt{d} + \sqrt{2\log(3H/\delta)})\lambda^{-1/2}C_g + \lambda^{1/2}B$$

$$+ (H + \psi)\left[\sqrt{\tilde{d}_h\log(1 + \frac{K'}{\lambda})} + K'\log\lambda + 2\log(3H/\delta)\right]$$

$$\leq 1 + \lambda^{1/2}B + (H + 1)\left[\sqrt{\tilde{d}_h\log(1 + \frac{K'}{\lambda})} + 2 + 2\log(3H/\delta)\right] = o(\sqrt{K'}).$$

Thus,

$$4\sqrt{K'}(H + 1 + \beta\sqrt{\log(K'M/\delta)}) + 4\beta\sqrt{d\log(dK'M/\delta)} << K'$$

for $K'$ large enough. Therefore, there exists $R$ that satisfies Equation (6). We now only need to verify $\iota < 1$. We have

$$\iota_0 = Bm^{-1/2}(2\sqrt{d} + \sqrt{2\log(3H/\delta)}) \leq 1/3,$$

$$\iota_1 = C_g R^{4/3} m^{-1/6}\sqrt{\log m} + C_g\left(\tilde{B}_1 + \tilde{B}_2 + \lambda^{-1}(1-\eta\lambda)^J\left(K'(H+1+\gamma_{h,1})^2 + \lambda\gamma_{h,2}^2\right)\right) \lesssim 1/3$$

if

$$(1-\eta\lambda)^J\left[K'(H+1+\beta\sqrt{\log(K'M/\delta)})^2 + \lambda\beta^2 d\log(dK'M/\delta)\right] \lesssim 1. \tag{7}$$

Note that

$$(1-\eta\lambda)^J \le e^{-\eta\lambda J},$$
$$K'(H+1+\beta\sqrt{\log(K'M/\delta)})^2 + \lambda\beta^2 d\log(dK'M/\delta) \lesssim K'H^2\lambda\beta^2 d\log(dK'M/\delta).$$

Thus, Equation (7) is satisfied if

$$J \gtrsim \eta\lambda\log\left(K'H^2\lambda\beta^2 d\log(dK'M/\delta)\right).$$

Finally note that $\iota_2 \le \iota_1$. Rearranging the derived conditions here gives the complete parameter conditions in Theorem 1. Specifically, the polynomial form of $m$ is $m \gtrsim \max\{R^8\log^3 m, K'^{10}(H+1)^2\log(3K'H/\delta), d^{3/2}R^{-1}\log^{3/2}(\sqrt{m}/R), K'^6 R^8\log^3 m, B^2 K'^2 d\log(3H/\delta)\}, m \gtrsim [2K'(H+1+\beta\sqrt{\log(K'M/\delta)})^2 + \lambda\beta^2 d\log(dK'M/\delta) + 1]^3 K'^3 R\log^3 m.$

$\square$

## C.3 PROOF OF THEOREM 2

In this subsection, we give a detailed proof of Theorem 2. We first present intermediate lemmas whose proofs are deferred to Section D. For any $h \in [H]$ and $k \in \mathcal{I}_h = [(H-h)K'+1, \ldots, (H-h+1)K']$, we define the filtration

$$\mathcal{F}_h^k = \sigma\left(\{(s_{h'}^t, a_{h'}^t, r_{h'}^t)\}_{h'\in[H]}^{t\le k} \cup \{(s_{h'}^{k+1}, a_{h'}^{k+1}, r_{h'}^{k+1})\}_{h'\le h-1} \cup \{(s_h^{k+1}, a_h^{k+1})\}\right).$$

Let

$$\Lambda_h^k := \lambda I + \sum_{t\in\mathcal{I}_k, t\le k} g(x_h^t; W_0)g(x_h^t; W_0)^T,$$
$$\tilde{\beta} := \beta(1 + 2\sqrt{\log(SAH/\delta)}).$$

In the following lemma, we connect the expected sub-optimality of $\tilde{\pi}$ to the summation of the uncertainty quantifier at empirical data.

**Lemma C.2.** *Suppose that the conditions in Theorem 1 all hold. With probability at least $1 - MHm^{-2} - 3\delta$,*

$$\text{SubOpt}(\tilde{\pi}) \le \frac{2\tilde{\beta}}{K'}\sum_{h=1}^{H}\sum_{k\in\mathcal{I}_h}\mathbb{E}_{\pi^*}\left[\|g(x_h; W_0)\|_{(\Lambda_h^k)^{-1}}\Big|\mathcal{F}_h^{k-1}, s_1^k\right] + \frac{16}{3K'}H\log(\log_2(K'H)/\delta)$$
$$+ \frac{2}{K'} + 2\iota,$$

**Lemma C.3.** *Under Assumption 5.2, for any $h \in [H]$ and fixed $W_0$, with probability at least $1-\delta$,*

$$\sum_{k\in\mathcal{I}_h}\mathbb{E}_{\pi^*}\left[\|g(x_h; W_0)\|_{(\Lambda_h^k)^{-1}}\Big|\mathcal{F}_{k-1}, s_1^k\right] \le \sum_{k\in\mathcal{I}_h}\kappa\|g(x_h; W_0)\|_{(\Lambda_h^k)^{-1}} + \kappa\sqrt{\frac{K'\log(1/\delta)}{\lambda}}.$$

**Lemma C.4.** *If $\lambda \ge C_g^2$ and $m = \Omega(K'^4\log(K'H/\delta))$, then with probability at least $1-\delta$, for any $h \in [H]$, we have*

$$\sum_{k\in\mathcal{I}_h}\|g(x_h; W_0)\|_{(\Lambda_h^k)^{-1}}^2 \le 2\tilde{d}_h\log(1 + K'/\lambda) + 1.$$

*where $\tilde{d}_h$ is the effective dimension defined in Definition 2.*

*Proof of Theorem 2.* Theorem 2 directly follows from Lemma C.2-C.3-C.4 using the union bound.

$\square$

## D  PROOF OF LEMMA C.1

In this section, we provide the proof for Lemma C.1. We set up preparation for all the results in the rest of the paper and provide intermediate lemmas that we use to prove Lemma C.1. The detailed proofs of these intermediate lemmas are deferred to Section E.

### D.1  PREPARATION

To prepare for the lemmas and proofs in the rest of the paper, we define the following quantities. Recall that we use abbreviation $x = (s, a) \in \mathcal{X} \subset \mathbb{S}_{d-1}$ and $x_h^k = (s_h^k, a_h^k) \in \mathcal{X} \subset \mathbb{S}_{d-1}$. For any $h \in [H]$ and $i \in [M]$, we define the perturbed loss function

$$\tilde{L}_h^i(W) := \frac{1}{2} \sum_{k \in \mathcal{I}_h} \left( f(x_h^k; W) - \tilde{y}_h^{i,k} \right)^2 + \frac{\lambda}{2} \|W + \zeta_h^i - W_0\|_2^2, \tag{8}$$

where

$$\tilde{y}_h^{i,k} := r_h^k + \tilde{V}_{h+1}(s_{h+1}^k) + \xi_h^{i,k},$$

$\tilde{V}_{h+1}$ is computed by Algorithm 1 at Line 10 for timestep $h + 1$, and $\{\xi_h^{i,k}\}$ and $\zeta_h^i$ are the Gaussian noises obtained at Line 5 of Algorithm 1.

Here the subscript $h$ and the superscript $i$ in $\tilde{L}_h^i(W)$ emphasize the dependence on the ensemble sample $i$ and timestep $h$. The gradient descent update rule of $\tilde{L}_h^i(W)$ is

$$\tilde{W}_h^{i,(j+1)} = \tilde{W}_h^{i,(j)} - \eta \nabla \tilde{L}_h^i(W), \tag{9}$$

where $\tilde{W}_h^{i,(0)} = W_0$ is the initialization parameters. Note that

$$\tilde{W}_h^i = \text{GradientDescent}(\lambda, \eta, J, \tilde{\mathcal{D}}_h^i, \zeta_h^i, W_0) = \tilde{W}_h^{i,(J)},$$

where $\tilde{W}_h^i$ is returned by Line 7 of Algorithm 1. We consider a non-perturbed auxiliary loss function

$$L_h(W) := \frac{1}{2} \sum_{k \in \mathcal{I}_h} \left( f(x_h^k; W) - y_h^k \right)^2 + \frac{\lambda}{2} \|W - W_0\|_2^2, \tag{10}$$

where

$$y_h^k := r_h^k + \tilde{V}_{h+1}(s_{h+1}^k).$$

Note that $L_h(W)$ is simply a non-perturbed version of $\tilde{L}_h^i(W)$ where we drop all the noises $\{\xi_h^{i,k}\}$ and $\{\zeta_h^i\}$. We consider the gradient update rule for $L_h(W)$ as follows

$$\hat{W}_h^{(j+1)} = \hat{W}_h^{(j)} - \eta \nabla L_h(W), \tag{11}$$

where $\hat{W}_h^{(0)} = W_0$ is the initialization parameters. To correspond with $\tilde{W}_h^i$, we denote

$$\hat{W}_h := \hat{W}_h^{(J)}. \tag{12}$$

We also define the auxiliary loss functions for both non-perturbed and perturbed data in the linear model with feature $g(\cdot; W_0)$ as follows

$$\tilde{L}_h^{i,lin}(W) := \frac{1}{2} \sum_{k \in \mathcal{I}_h} \left( \langle g(x_h^k; W_0), W \rangle - \tilde{y}_h^{i,k} \right)^2 + \frac{\lambda}{2} \|W + \zeta_h^i - W_0\|_2^2, \tag{13}$$

$$L_h^{lin}(W) := \frac{1}{2} \sum_{k \in \mathcal{I}_h} \left( \langle g(x_h^k; W_0), W \rangle - y_h^k \right)^2 + \frac{\lambda}{2} \|W - W_0\|_2^2. \tag{14}$$

We consider the auxiliary gradient updates for $\tilde{L}_h^{i,lin}(W)$ as

$$\tilde{W}_h^{i,lin,(j+1)} = \tilde{W}_h^{i,lin,(j)} - \eta \nabla \tilde{L}_h^{i,lin}(W), \tag{15}$$

$$\hat{W}_h^{lin,(j+1)} = \hat{W}_h^{lin,(j)} - \eta\nabla\tilde{L}_h^{lin}(W), \tag{16}$$

where $\tilde{W}_h^{i,lin,(0)} = \hat{W}_h^{i,lin,(0)} = W_0$ for all $i, h$. Finally, we define the least-square solutions to the auxiliary perturbed and non-perturbed loss functions for the linear model as follows

$$\tilde{W}_h^{i,lin} = \underset{W\in\mathbb{R}^{md}}{\arg\min}\ \tilde{L}_h^{i,lin}(W), \tag{17}$$

$$\hat{W}_h^{lin} = \underset{W\in\mathbb{R}^{md}}{\arg\min}\ L_h^{lin}(W). \tag{18}$$

For any $h \in [H]$, we define the auxiliary covariance matrix $\Lambda_h$ as follows

$$\Lambda_h := \lambda I_{md} + \sum_{k\in\mathcal{I}_h} g(x_h^k; W_0)g(x_h^k; W_0)^T. \tag{19}$$

It is worth remarking that Algorithm 1 only uses Equation (8) and (9) thus it does not actually require any of the auxiliary quantities defined in this subsection during its run time. The auxiliary quantities here are only for our theoretical analysis.

## D.2 PROOF OF LEMMA C.1

In this subsection, we give detailed proof of Lemma C.1. To prepare for proving Lemma C.1, we first provide the following intermediate lemmas. The detailed proofs of these intermediate lemmas are deferred to Section E.

In the following lemma, we bound the uncertainty $f(x; \hat{W}_h)$ in estimating the Bellman operator at the estimated state-value function $\mathbb{B}_h\tilde{V}_{h+1}$.

**Lemma D.1.** *Let*

$$\begin{cases} m = \Omega\left(K'^{10}(H+\psi)^2\log(3K'H/\delta)\right) \\ \lambda > 1 \\ K'C_g^2 \geq \lambda \end{cases}$$

*With probability at least $1 - Hm^{-2} - 2\delta$, for any $x \in \mathbb{S}_{d-1}$, and any $h \in [H]$,*

$$|f(x; \hat{W}_h) - (\mathbb{B}_h\tilde{V}_{h+1})(x)| \leq \beta\cdot\|g(x; W_0)\|_{\Lambda_h^{-1}} + \iota_2 + \iota_0,$$

*where $\tilde{V}_{h+1}$ is computed by Algorithm 1 for timestep $h + 1$, $\hat{W}_h$ is defined in Equation (12), and $\beta$, $\iota_2$ and $\iota_0$ are defined in Table 3.*

In the following lemma, we establish the anti-concentration of $\tilde{Q}_h$.

**Lemma D.2.** *Let*

$$\begin{cases} m = \Omega\left(d^{3/2}R^{-1}\log^{3/2}(\sqrt{m}/R)\right) \\ R = \mathcal{O}\left(m^{1/2}\log^{-3}m\right), \\ \eta \leq (\lambda + K'C_g^2)^{-1}, \\ R \geq \max\{4\tilde{B}_1, 4\tilde{B}_2, 2\sqrt{2\lambda^{-1}K'(H+\psi+\gamma_{h,1})^2 + 4\gamma_{h,2}^2}\}, \end{cases} \tag{20}$$

*where $\tilde{B}_1$, $\tilde{B}_2$, $\gamma_{h,1}$ and $\gamma_{h,2}$ are defined in Table 3, and $C_g$ is a constant given in Lemma G.1. Let $M = \log\frac{HSA}{\delta}/\log\frac{1}{1-\Phi(-1)}$ where $\Phi(\cdot)$ is the cumulative distribution function of the standard normal distribution and $M$ is the number of bootstrapped samples in Algorithm 1. Then with probability $1 - MHm^{-2} - \delta$, for any $x \in \mathbb{S}_{d-1}$ and $h \in [H]$,*

$$\tilde{Q}_h(x) \leq \max\{\langle g(x; W_0), \hat{W}_h^{lin} - W_0\rangle - \sigma_h\|g(x; W_0)\|_{\Lambda_h^{-1}} + \iota_1 + \iota_2, 0\},$$

*where $\hat{W}_h^{lin}$ is defined in Equation (18), $\tilde{Q}_h$ is computed by Line 9 of Algorithm 1, and $\iota_1$ and $\iota_2$ are defined in Table 3.*

We prove the following linear approximation error lemma.

**Lemma D.3.** *Let*

$$\begin{cases} m = \Omega\left(d^{3/2}R^{-1}\log^{3/2}(\sqrt{m}/R)\right) \\ R = \mathcal{O}\left(m^{1/2}\log^{-3}m\right), \\ \eta \leq (\lambda + K'C_g^2)^{-1}, \\ R \geq \max\{4\tilde{B}_1, 4\tilde{B}_2, 2\sqrt{2\lambda^{-1}K'(H+\psi+\gamma_{h,1})^2 + 4\gamma_{h,2}^2}\}, \end{cases} \tag{21}$$

*where $\tilde{B}_1$, $\tilde{B}_2$, $\gamma_{h,1}$ and $\gamma_{h,2}$ are defined in Table 3, and $C_g$ is a constant given in Lemma G.1. With probability at least $1 - MHm^{-2} - \delta$, for any $(x, i, j, h) \in \mathbb{S}_{d-1} \times [M] \times [J] \times [H]$,*

$$|f(x; \tilde{W}_h^{i,(j)}) - \langle g(x; W_0), \tilde{W}_h^{i,lin} - W_0 \rangle| \leq \iota_1,$$

*where $\tilde{W}_h^{i,(j)}$, $\tilde{W}_h^{i,lin}$, and $\iota_1$ are defined in Equation (9), Equation (17), and Table 3, respectively.*

*In addition, with probability at least $1 - Hm^{-2}$, for any for any $(x, j, h) \in \mathbb{S}_{d-1} \times [J] \times [H]$,*

$$|f(x; \hat{W}_h^{(j)}) - \langle g(x; W_0), \hat{W}_h^{lin} - W_0 \rangle| \leq \iota_2,$$

*where $\hat{W}_h^{(j)}$, $\hat{W}_h^{lin}$, and $\iota_2$ are defined in Equation (11), Equation (18), and Table 3, respectively.*

We now can prove Lemma C.1.

*Proof of Lemma C.1.* Note that the first fourth conditions in Equation (5) of Lemma C.1 satisfy Equation (21). Moreover, the event in which the inequality in Lemma D.3 holds already implies the event in which the inequality in Lemma D.1 holds (see the proofs of Lemma D.3 and Lemma D.1 in Section D). Now in the rest of the proof, we consider the joint event in which both the inequality of Lemma D.3 and that of Lemma D.1 hold. Then, we also have the inequality in Lemma D.1. Consider any $x \in \mathcal{X}, h \in [H]$.

It follows from Lemma D.1 that

$$(\mathbb{B}_h \tilde{V}_{h+1})(x) \geq f(x; \hat{W}_h) - \beta \cdot \|g(x; W_0)\|_{\Lambda_h^{-1}} - \iota_0 - \iota_2. \tag{22}$$

It follows from Lemma D.2 that

$$\tilde{Q}_h(x) \leq \max\{\langle g(x; W_0), \hat{W}_h^{lin} - W_0 \rangle - \sigma_h\|g(x; W_0)\|_{\Lambda_h^{-1}} + \iota_1 + \iota_2, 0\}. \tag{23}$$

Note that $\tilde{Q}_h(x) \geq 0$. If $\langle g(x; W_0), \hat{W}_h^{lin} - W_0 \rangle - \sigma_h\|g(x; W_0)\|_{\Lambda_h^{-1}} + \iota_1 + \iota_2 \leq 0$, Equation (23) implies that $\tilde{Q}_h(x) = 0$ and thus

$$\begin{aligned} err_h(x) &= (\mathbb{B}_h \tilde{V}_{h+1})(x) - \tilde{Q}_h(x) \\ &= (\mathbb{B}_h \tilde{V}_{h+1})(x) \geq 0. \end{aligned}$$

Otherwise, if $\langle g(x; W_0), \hat{W}_h^{lin} - W_0 \rangle - \sigma_h\|g(x; W_0)\|_{\Lambda_h^{-1}} + \iota_1 + \iota_2 > 0$, Equation (23) implies that

$$\tilde{Q}_h(x) \leq \langle g(x; W_0), \hat{W}_h^{lin} - W_0 \rangle - \sigma_h\|g(x; W_0)\|_{\Lambda_h^{-1}} + \iota_1 + \iota_2. \tag{24}$$

Thus, combining Equation (22), (24) and Lemma D.3, with the choice $\sigma_h \geq \beta$, we have

$$err_h(x) := (\mathbb{B}_h \tilde{V}_{h+1})(x) - \tilde{Q}_h(x) \geq -(\iota_0 + \iota_1 + 2\iota_2) = -\iota.$$

As $\iota \geq 0$, in either case, we have

$$err_h(x) := (\mathbb{B}_h \tilde{V}_{h+1})(x) - \tilde{Q}_h(x) \geq -\iota. \tag{25}$$

Note that due to Equation (25), we have

$$\tilde{Q}_h(x) \leq (\mathbb{B}_h \tilde{V}_{h+1})(x) + \iota \leq H - h + 1 + \iota < H - h + 1 + \psi,$$

where the last inequality holds due to the choice $\psi > \iota$. Thus, we have

$$\tilde{Q}_h(x) = \min\{\min_{i\in[M]} f(x; \tilde{W}_h^i), H - h + 1 + \psi\}^+ = \max\{\min_{i\in[M]} f(x; \tilde{W}_h^i), 0\}. \qquad (26)$$

Substituting Equation (26) into the definition of $err_h(x)$, we have

$$\begin{aligned}
err_h(x) &= (\mathbb{B}_h \tilde{V}_{h+1})(x) - \tilde{Q}_h(x) \\
&\leq (\mathbb{B}_h \tilde{V}_{h+1})(x) - \min_{i\in[M]} f(x; \tilde{W}_h^i) \\
&= (\mathbb{B}_h \tilde{V}_{h+1})(x) - f(x; \hat{W}_h) + f(x; \hat{W}_h) - \min_{i\in[M]} f(x; \tilde{W}_h^i) \\
&\leq \beta \cdot \|g(x; W_0)\|_{\Lambda_h^{-1}} + \iota_0 + \iota_2 + f(x; \hat{W}_h) - \min_{i\in[M]} f(x; \tilde{W}_h^i) \\
&\leq \beta \cdot \|g(x; W_0)\|_{\Lambda_h^{-1}} + \iota_0 + \iota_2 + \langle g(x; W_0), \hat{W}_h^{lin} - W_0 \rangle + \iota_2 \\
&\quad - \min_{i\in[M]} \langle g(x; W_0), \tilde{W}_h^{i,lin} - W_0 \rangle + \iota_1 \\
&= \beta \cdot \|g(x; W_0)\|_{\Lambda_h^{-1}} + \iota_0 + \iota_2 + \max_{i\in[M]} \langle g(x; W_0), \hat{W}_h^{lin} - \tilde{W}_h^{i,lin} \rangle + \iota_1 + \iota_2 \\
&\leq \beta \cdot \|g(x; W_0)\|_{\Lambda_h^{-1}} + \iota_0 + \iota_2 + \sqrt{2\log(MSAH/\delta)}\sigma_h\|g(x; W_0)\|_{\Lambda_h^{-1}} + \iota_1 + \iota_2
\end{aligned}$$

where the first inequality holds due to Equation (26), the second inequality holds due to Lemma D.1, the third inequality holds due to Lemma D.3, and the last inequality holds due to Lemma E.2 and Lemma G.3 via the union bound.

$\square$

## D.3 PROOF OF LEMMA C.2

*Proof of Lemma C.2.* Let $Z_k := \tilde{\beta} \sum_{h=1}^{H} \mathbb{E}_{\pi^*}\left[\mathbb{1}\{k \in \mathcal{I}_h\}\|g(x_h; W_0)\|_{(\Lambda_h^k)^{-1}}|s_1^k, \mathcal{F}_h^{k-1}\right]$ where $\mathbb{1}\{\}$ is the indicator function. Under the event in which the inequality in Theorem 1 holds, we have

$$\begin{aligned}
\text{SubOpt}(\tilde{\pi}) &\leq \min\left\{H, \tilde{\beta} \cdot \mathbb{E}_{\pi^*}\left[\sum_{h=1}^{H}\|g(x_h; W_0)\|_{\Lambda_h^{-1}}\right] + 2\iota\right\} \\
&\leq \min\left\{H, \tilde{\beta}\mathbb{E}_{\pi^*}\left[\sum_{h=1}^{H}\|g(x_h; W_0)\|_{\Lambda_h^{-1}}\right]\right\} + 2\iota \\
&= \frac{1}{K'}\sum_{k=1}^{K}\min\left\{H, \tilde{\beta}\mathbb{E}_{\pi^*}\left[\sum_{h=1}^{H}\mathbb{1}\{k \in \mathcal{I}_h\}\|g(x_h; W_0)\|_{\Lambda_h^{-1}}\right]\right\} + 2\iota \\
&\leq \frac{1}{K'}\sum_{k=1}^{K}\min\left\{H, \tilde{\beta}\mathbb{E}_{\pi^*}\left[\sum_{h=1}^{H}\mathbb{1}\{k \in \mathcal{I}_h\}\|g(x_h; W_0)\|_{(\Lambda_h^k)^{-1}}|\mathcal{F}_h^{k-1}\right]\right\} + 2\iota \\
&= \frac{1}{K'}\sum_{k=1}^{K}\min\left\{H, \mathbb{E}[Z_k|\mathcal{F}_h^{k-1}]\right\} + 2\iota \\
&\leq \frac{1}{K'}\sum_{k=1}^{K}\mathbb{E}\left[\min\{H, Z_k\}|\mathcal{F}_h^{k-1}\right] + 2\iota, \qquad (27)
\end{aligned}$$

where the first inequality holds due to Theorem 1 and that $\text{SubOpt}(\tilde{\pi}; s_1) \leq H, \forall s_1 \in \mathcal{S}$, the second inequality holds due to $\min\{a, b + c\} \leq \min\{a, b\} + c$, the third inequality holds due to that $\Lambda_h^{-1} \preceq (\Lambda_h^k)^{-1}$, the fourth inequality holds due to Jensen's inequality for the convex function $f(x) = \min\{H, x\}$. It follows from Lemma G.9 that with probability at least $1 - \delta$,

$$\sum_{k=1}^{K}\mathbb{E}\left[\min\{H, Z_k\}|\mathcal{F}_{k-1}\right] \leq 2\sum_{k=1}^{K} Z_k + \frac{16}{3}H\log(\log_2(KH)/\delta) + 2. \qquad (28)$$

Substituting Equation (28) into Equation (27) and using the union bound complete the proof.

$\square$

## D.4 PROOF OF LEMMA C.3

*Proof of Lemma C.3.* Let $Z_h^k := \mathbb{1}\{k \in \mathcal{I}_h\}\frac{d_h^*(x_h^k)}{d_h^\mu(x_h^k)}\|g(x_h^k; W_0)\|_{(\Lambda_h^k)^{-1}}$. We have $Z_h^k$ is $\mathcal{F}_h^k$-measurable, and by Assumption 5.2, we have,

$$|Z_h^k| \leq \frac{d_h^*(x_h^k)}{d_h^\mu(x_h^k)}\|g(x_h^k; W_0))\|_2\sqrt{\|(\Lambda_h^k)^{-1}\|} \leq 1/\sqrt{\lambda}\frac{d_h^*(x_h^k)}{d_h^\mu(x_h^k)} < \infty,$$

$$\mathbb{E}\left[Z_h^k|\mathcal{F}_h^{k-1}, s_1^k\right] = \mathbb{E}_{x_h \sim d_h^\mu}\left[\mathbb{1}\{k \in \mathcal{I}_h\}\frac{d_h^*(x_h)}{d_h^\mu(x_h)}\|g(x_h; W_0)\|_{(\Lambda_h^k)^{-1}}\bigg|\mathcal{F}_h^{k-1}, s_1^k\right].$$

Thus, by Lemma G.4, for any $h \in [H]$, with probability at least $1 - \delta$, we have:

$$\sum_{k=1}^K \mathbb{E}_{x \sim d_h^*}\left[\mathbb{1}\{k \in \mathcal{I}_h\}\|g(x_h; W_0)\|_{(\Lambda_h^k)^{-1}}\bigg|\mathcal{F}_h^{k-1}, s_1^k\right]$$

$$= \sum_{k=1}^K \mathbb{E}_{x_h \sim d_h^\mu)}\left[\mathbb{1}\{k \in \mathcal{I}_h\}\frac{d_h^*(x_h)}{d_h^\mu(x_h)}\|\phi_h(x_h)\|_{(\Lambda_h^k)^{-1}}\bigg|\mathcal{F}_h^{k-1}, s_1^k\right]$$

$$\leq \sum_{k=1}^K \mathbb{1}\{k \in \mathcal{I}_h\}\frac{d_h^*(x_h^k)}{d_h^\mu(x_h^k)}\|g(x_h^k; W_0)\|_{(\Lambda_h^k)^{-1}} + \sqrt{\frac{1}{\lambda}\log(1/\delta)}\sqrt{\sum_{k=1}^K \mathbb{1}\{k \in \mathcal{I}_h\}\left(\frac{d_h^*(x_h^k)}{d_h^\mu(x_h^k)}\right)^2}$$

$$\leq \kappa \sum_{k=1}^K \mathbb{1}\{k \in \mathcal{I}_h\}\|g(x_h; W_0)\|_{(\Lambda_h^k)^{-1}} + \kappa\sqrt{\frac{K'\log(1/\delta)}{\lambda}}$$

$$= \kappa \sum_{k \in \mathcal{I}_h}\|g(x_h; W_0)\|_{(\Lambda_h^k)^{-1}} + \kappa\sqrt{\frac{K'\log(1/\delta)}{\lambda}}$$

$\square$

## D.5 PROOF OF LEMMA C.4

*Proof of Lemma C.4.* For any fixed $h \in [H]$, let

$$U = [g(x_h^k; W_0)]_{k \in \mathcal{I}_h} \in \mathbb{R}^{md \times K'}.$$

By the union bound, with probability at least $1 - \delta$, for any $h \in [H]$, we have

$$\sum_{k \in \mathcal{I}_h}\|g(x_h; W_0)\|_{(\Lambda_h^k)^{-1}}^2 \leq 2\log\frac{\det \Lambda_h}{\det(\lambda I)}$$

$$= 2\log\det\left(I + \sum_{k \in \mathcal{I}_h}g(x_h^k; W_0)g(x_h^k; W_0)^T/\lambda\right)$$

$$= 2\log\det(I + UU^T/\lambda)$$

$$= 2\log\det(I + U^TU/\lambda)$$

$$= 2\log\det(I + \mathcal{K}_h/\lambda + (U^TU - \mathcal{K}_h)/\lambda)$$

$$\leq 2\log\det(I + \mathcal{K}_h/\lambda) + 2\operatorname{tr}\left((I + \mathcal{K}_h/\lambda)^{-1}(U^TU - \mathcal{K}_h)/\lambda\right)$$

$$\leq 2\log\det(I + \mathcal{K}_h/\lambda) + 2\|(I + \mathcal{K}_h/\lambda)^{-1}\|_F\|U^TU - \mathcal{K}_h\|_F$$

$$\leq 2\log\det(I + \mathcal{K}_h/\lambda) + 2\sqrt{K'}\|U^TU - \mathcal{K}_h\|_F$$

$$\leq 2\log\det(I + \mathcal{K}_h/\lambda) + 1$$

$$= 2\tilde{d}_h\log(1 + K'/\lambda) + 1$$

where the first inequality holds due to $\lambda \geq C_g^2$ and (Abbasi-yadkori et al., 2011, Lemma 11), the third equality holds due to that $\log\det(I + AA^T) = \log\det(I + A^T A)$, the second inequality holds due to that $\log\det(A + B) \leq \log\det(A) + \text{tr}(A^{-1}B)$ as the result of the convexity of logdet, the third inequality holds due to that $\text{tr}(A) \leq \|A\|_F$, the fourth inequality holds due to $2\sqrt{K'}\|U^T U - \mathcal{K}_h\|_F \leq 1$ by the choice of $m = \Omega(K'^4 \log(K'H/\delta))$, Lemma G.2 and the union bound, and the last equality holds due to the definition of $\tilde{d}_h$. □

# E   PROOFS OF LEMMAS IN SECTION D

## E.1   PROOF OF LEMMA D.1

In this subsection, we give detailed proof of Lemma D.1. For this, we first provide a lemma about the linear approximation of the Bellman operator. In the following lemma, we show that $\mathbb{B}_h \tilde{V}_{h+1}$ can be well approximated by the class of linear functions with features $g(\cdot; W_0)$ with respect to $l_\infty$-norm.

**Lemma E.1.** *Under Assumption 5.1, with probability at least $1 - \delta$ over $w_1, \ldots, w_m$ drawn i.i.d. from $\mathcal{N}(0, I_d)$, for any $h \in [H]$, there exist $c_1, \ldots, c_m$ where $c_i \in \mathbb{R}^d$ and $\|c_i\|_2 \leq \frac{B}{m}$ such that*

$$\bar{Q}_h(x) := \sum_{i=1}^m c_i^T x \sigma'(w_i^T x),$$

$$\|\mathbb{B}_h \tilde{V}_{h+1} - \bar{Q}_h\|_\infty \leq \frac{B}{\sqrt{m}}(2\sqrt{d} + \sqrt{2\log(H/\delta)})$$

*Moreover, $\bar{Q}_h(x)$ can be re-written as*

$$\bar{Q}_h(x) = \langle g(x; W_0), \bar{W}_h \rangle,$$
$$\bar{W}_h := \sqrt{m}[a_1 c_1^T, \ldots, a_m c_m^T]^T \in \mathbb{R}^{md}, \text{ and } \|\bar{W}_h\|_2 \leq B. \tag{29}$$

We now can prove Lemma D.1.

*Proof of Lemma D.1.* We first bound the difference $\langle g(x; W_0), \bar{W}_h \rangle - \langle g(x; W_0), \hat{W}_h^{lin} - W_0 \rangle$:

$$\langle g(x; W_0), \bar{W}_h \rangle - \langle g(x; W_0), \hat{W}_h^{lin} - W_0 \rangle = g(x; W_0)^T \bar{W}_h - g(x; W_0)^T \Lambda_h^{-1} \sum_{k \in \mathcal{I}_h} g(x_h^k; W_0) y_h^k$$

$$= \underbrace{g(x; W_0)^T \bar{W}_h - g(x; W_0)^T \Lambda_h^{-1} \sum_{k \in \mathcal{I}_h} g(x_h^k; W_0) \cdot (\mathbb{B}_h \tilde{V}_{h+1})(x_h^k)}_{I_1}$$

$$+ \underbrace{g(x; W_0)^T \Lambda_h^{-1} \sum_{k \in \mathcal{I}_h} g(x_h^k; W_0) \cdot \left[ (\mathbb{B}_h \tilde{V}_{h+1})(x_h^k) - (r_h^k + \tilde{V}_{h+1}(s_{h+1}^k)) \right]}_{I_2}.$$

For bounding $I_1$, it follows from Lemma E.1 that with probability at least $1 - \delta/3$, for any for any $x \in \mathbb{S}_{d-1}$ and any $h \in [H]$,

$$|(\mathbb{B}_h \tilde{V}_{h+1})(x) - \langle g(x; W_0), \bar{W}_h \rangle| \leq \iota_0,$$

where $\iota_0$ is defined in Table 3. where $\bar{W}_h$ is defined in Lemma E.1. Thus, with probability at least $1 - \delta/3$, for any for any $x \in \mathbb{S}_{d-1}$ and any $h \in [H]$,

$$I_1 = g(x; W_0)^T \bar{W}_h - g(x; W_0)^T \Lambda_h^{-1} \sum_{k \in \mathcal{I}_h} g(x_h^k; W_0) \cdot \left[ (\mathbb{B}_h \tilde{V}_{h+1})(x_h^k) - g(x_h^k; W_0)^T \bar{W}_h \right]$$

$$- g(x; W_0)^T \bar{W}_h + \lambda g(x; W_0)^T \Lambda_h^{-1} \bar{W}_h$$

$$\leq \|g(x;W_0)^T\|_{\Lambda_h^{-1}} \sum_{k\in\mathcal{I}_h} \iota_0\|g(x_h^k;W_0)^T\|_{\Lambda_h^{-1}} + \lambda\|g(x;W_0)\|_{\Lambda_h^{-1}}\|\bar{W}_h\|_{\Lambda_h^{-1}}$$

$$\leq \|g(x;W_0)\|_{\Lambda_h^{-1}} \left[ K'\iota_0\lambda^{-1/2}C_g + \lambda^{1/2}B \right], \tag{30}$$

where the first equation holds due to the definition of $\Lambda_h$, and the last inequality holds due to Step I with $\|\bar{W}_h\|_{\Lambda_h^{-1}} \leq \sqrt{\|\Lambda_h^{-1}\|_2} \cdot \|\bar{W}_h\|_2 \leq \lambda^{-1/2}B$.

For bounding $I_2$, we have

$$I_2 \leq \underbrace{\left\| \sum_{k\in\mathcal{I}_h} g(x_h^k;W_0)\left[(\mathbb{B}_h\tilde{V}_{h+1})(x_h^k) - r_h^k - \tilde{V}_{h+1}(s_{h+1}^k)\right] \right\|_{\Lambda_h^{-1}}}_{I_3} \|g(x;W_0)\|_{\Lambda_h^{-1}}. \tag{31}$$

If we directly apply the result of Jin et al. (2021) in linear MDP, we would get

$$I_2 \lesssim dmH\sqrt{\log(2dmK'H/\delta)} \cdot \|g(x;W_0)\|_{\Lambda_h^{-1}},$$

which gives a vacuous bound as $m$ is sufficiently larger than $K$ in our problem. Instead, in the following, we present an alternate proof that avoids such vacuous bound.

For notational simplicity, we write

$$\epsilon_h^k := (\mathbb{B}_h\tilde{V}_{h+1})(x_h^k) - r_h^k - \tilde{V}_{h+1}(s_{h+1}^k),$$
$$E_h := [(\epsilon_h^k)_{k\in\mathcal{I}_h}]^T \in \mathbb{R}^{K'}.$$

We denote $\mathcal{K}_h^{init} := [\langle g(x_h^i;W_0), g(x_h^j;W_0)\rangle]_{i,j\in\mathcal{I}_h}$ as the Gram matrix of the empirical NTK kernel on the data $\{x_h^k\}_{k\in[K]}$. We denote

$$G_0 := \left(g(x_h^k;W_0)\right)_{k\in\mathcal{I}_h} \in \mathbb{R}^{md\times K'},$$
$$\mathcal{K}_h^{int} := G_0^T G_0 \in \mathcal{R}^{K'\times K'}.$$

Recall the definition of the Gram matrix $\mathcal{K}_h$ of the NTK kernel on the data $\{x_h^k\}_{k\in\mathcal{I}_h}$. It follows from Lemma G.2 and the union bound that if $m = \Omega(\epsilon^{-4}\log(3K'H/\delta))$ with probability at least $1 - \delta/3$, for any $h\in[H]$,

$$\|\mathcal{K}_h - \mathcal{K}_h^{init}\|_F \leq \sqrt{K'}\epsilon. \tag{32}$$

We now can bound $I_3$. We have

$$I_3^2 = \left\| \sum_{k=\mathcal{I}_h} g(x_h^k;W_0)\epsilon_h^k \right\|_{\Lambda_h^{-1}}^2$$
$$= E_h^T G_0^T(\lambda I_{md} + G_0 G_0^T)^{-1} G_0 E_h$$
$$= E_h^T G_0^T G_0(\lambda I_{K'} + G_0^T G_0)^{-1} E_h$$
$$= E_h^T \mathcal{K}_h^{init}(\mathcal{K}_h^{init} + \lambda I_K)^{-1} E_h$$
$$= \underbrace{E_h^T \mathcal{K}_h(\mathcal{K}_h + \lambda I_{K'})^{-1} E_h}_{I_5} + \underbrace{E_h^T \left(\mathcal{K}_h(\mathcal{K}_h + \lambda I_{K'})^{-1} - \mathcal{K}_h^{init}(\mathcal{K}_h^{int} + \lambda I_{K'})^{-1} E_h\right)}_{I_4}. \tag{33}$$

We bound each $I_4$ and $I_5$ separately. For bounding $I_4$, applying Lemma G.1, with $1 - Hm^{-2}$, for any $h\in[H]$,

$$I_4 \leq \left\|\mathcal{K}_h(\mathcal{K}_h + \lambda I_{K'})^{-1} - \mathcal{K}_h^{init}(\mathcal{K}_h^{int} + \lambda I_{K'})^{-1}\right\|_2 \|E_h\|_2^2$$
$$= \left\|(\mathcal{K}_h - \mathcal{K}_h^{init})(\mathcal{K}_h + \lambda I_{K'})^{-1} + \mathcal{K}_h^{init}\left((\mathcal{K}_h + \lambda I_{K'})^{-1} - (\mathcal{K}_h^{int} + \lambda I_{K'})^{-1}\right)\right\|_2 \|E_h\|_2^2$$
$$\leq \|\mathcal{K}_h - \mathcal{K}_h^{init}\|_2/\lambda + \|\mathcal{K}_h^{init}\|_2 \cdot \|\mathcal{K}_h - \mathcal{K}_h^{init}\|_2/\lambda^2\|E_h\|_2^2$$

$$\leq \frac{\lambda + K'C_g^2}{\lambda^2} \|\mathcal{K}_h - \mathcal{K}_h^{init}\|_2 \|E_h\|_2^2$$
$$\leq 2K'C_g^2 K'(H+\psi)^2 \|\mathcal{K}_h - \mathcal{K}_h^{init}\|_2, \tag{34}$$

where the first inequality holds due to the triangle inequality, the second inequality holds due to the triangle inequality, Lemma G.7, and $\|(\mathcal{K}_h + \lambda I_{K'})^{-1}\|_2 \leq \lambda^{-1}$, the third inequality holds due to $\|\mathcal{K}_h^{init}\|_2 \leq \|G_0\|_2^2 \leq \|G_0\|_F^2 \leq K'C_g^2$ due to Lemma G.1, the fourth inequality holds due to $\|E_h\|_2 \leq \sqrt{K'}(H+\psi)$, $\lambda \geq 1$, and $K'C_g^2 \geq \lambda$.

Substituting Equation (32) in Equation (34) using the union bound, with probability $1 - Hm^{-2} - \delta/3$, for any $h \in [H]$,

$$I_4 \leq 2K'C_g^2 K'(H+\psi)^2 \sqrt{K'}\epsilon \leq 1, \tag{35}$$

where the last inequality holds due to the choice of $\epsilon = 1/2K'^{-5/2}(H+\psi)^{-2}C_g^{-2}$ and thus

$$m = \Omega(\epsilon^{-4}\log(3K'H/\delta)) = \Omega\left(K'^{10}(H+\psi)^2 \log(3K'H/\delta)\right).$$

For bounding $I_5$, as $\lambda > 1$, we have

$$I_5 = E_h^T \mathcal{K}_h (\mathcal{K}_h + \lambda I_{K'})^{-1} E_h$$
$$\leq E_h^T (\mathcal{K}_h + (\lambda-1)I_K)(\mathcal{K}_h + \lambda I_{K'})^{-1} E_h$$
$$= E_h^T \left[(\mathcal{K}_h + (\lambda-1)I_{K'})^{-1} + I_{K'}\right]^{-1} E_h. \tag{36}$$

Let $\sigma(\cdot)$ be the $\sigma$-algebra induced by the set of random variables. For any $h \in [H]$ and $k \in \mathcal{I}_h = [(H-h)K'+1, \dots, (H-h+1)K']$, we define the filtration

$$\mathcal{F}_h^k = \sigma\left(\{(s_{h'}^t, a_{h'}^t, r_{h'}^t)\}_{h' \in [H]}^{t \leq k} \cup \{(s_{h'}^{k+1}, a_{h'}^{k+1}, r_{h'}^{k+1})\}_{h' \leq h-1} \cup \{(s_h^{k+1}, a_h^{k+1})\}\right)$$

which is simply all the data up to episode $k+1$ and timestep $h$ but right before $r_h^{k+1}$ and $s_{h+1}^{k+1}$ are generated (in the offline data). [7] Note that for any $k \in \mathcal{I}_h$, we have $(s_h^k, a_h^k, r_h^k, s_{h+1}^k) \in \mathcal{F}_h^k$, and

$$\tilde{V}_{h+1} \in \sigma\left(\{(s_{h'}^k, a_{h'}^k, r_{h'}^k)\}_{h' \in [h+1, \dots, H]}^{k \in \mathcal{I}_{h'}}\right) \subseteq \mathcal{F}_h^{k-1} \subseteq \mathcal{F}_h^k.$$

Thus, for any $k \in \mathcal{I}_h$, we have

$$\epsilon_h^k = (\mathbb{B}_h \tilde{V}_{h+1})(x_h^k) - r_h^k - \tilde{V}_{h+1}(s_{h+1}^k) \in \mathcal{F}_h^k.$$

The key property in our data split design is that we nicely have that

$$\tilde{V}_{h+1} \in \sigma\left(\{(s_{h'}^k, a_{h'}^k, r_{h'}^k)\}_{h' \in [h+1, \dots, H]}^{k \in \mathcal{I}_{h'}}\right) \subseteq \mathcal{F}_h^{k-1}.$$

Thus, conditioned on $\mathcal{F}_h^{k-1}$, $\tilde{V}_{h+1}$ becomes deterministic. This implies that

$$\mathbb{E}\left[\epsilon_h^k | \mathcal{F}_h^{k-1}\right] = \left[(\mathbb{B}_h \tilde{V}_{h+1})(s_h^k, a_h^k) - r_h^k - \tilde{V}_{h+1}(s_{h+1}^k) | \mathcal{F}_h^{k-1}\right] = 0.$$

Note that this is only possible with our data splitting technique. Otherwise, $\epsilon_h^k$ is not zero-mean due to the data dependence structure induced in offline RL with function approximation (Nguyen-Tang et al., 2022b). Our data split technique is a key to avoid the uniform convergence argument with the log covering number that is often used to bound this term in Jin et al. (2021), which is often large for complex models. For example, in a two-layer ReLU NTK, the eigenvalues of the induced RKHS has $d$-polynomial decay (Bietti & Mairal, 2019), thus its log covering number roughly follows, by (Yang et al., 2020, Lemma D1),

$$\log \mathcal{N}_\infty(\mathcal{H}_{ntk}, \epsilon, B) \lesssim \left(\frac{1}{\epsilon}\right)^{\frac{4}{\alpha d - 1}},$$

---

[7]To be more precise, we need to include into the filtration the randomness from the generated noises $\{\xi_h^{k,i}\}$ and $\{\zeta_h^i\}$ but since these noises are independent of any other randomness, they do not affect any derivations here but only complicate the notations and representations.

for some $\alpha \in (0, 1)$.

Therefore, for any $h \in [H]$, $\{\epsilon_h^k\}_{k \in \mathcal{I}_h}$ is adapted to the filtration $\{\mathcal{F}_h^k\}_{k \in \mathcal{I}_h}$. Applying Lemma G.5 with $Z_t = \epsilon_t^h \in [-(H + \psi), H + \psi]$, $\sigma^2 = (H + \psi)^2$, $\rho = \lambda - 1$, for any $\delta > 0$, with probability at least $1 - \delta/3$, for any $h \in [H]$,

$$E_h^T \left[ (\mathcal{K}_h + (\lambda - 1)I_{K'})^{-1} + I \right]^{-1} E_h \leq (H + \psi)^2 \log\det (\lambda I_{K'} + \mathcal{K}_h) + 2(H + \psi)^2 \log(3H/\delta) \tag{37}$$

Substituting Equation (37) into Equation (36), we have

$$\begin{aligned}
I_5 &\leq (H + \psi)^2 \log\det(\lambda I_{K'} + \mathcal{K}_h) + 2(H + \psi)^2 \log(H/\delta) \\
&= (H + \psi)^2 \log\det(I_{K'} + \mathcal{K}_h/\lambda) + (H + \psi)^2 K' \log \lambda + 2(H + \psi)^2 \log(H/\delta) \\
&= (H + \psi)^2 \tilde{d}_h \log(1 + K'/\lambda) + (H + \psi)^2 K' \log \lambda + 2(H + \psi)^2 \log(H/\delta), \tag{38}
\end{aligned}$$

where the last equation holds due to the definition of the effective dimension.

Combining Equations (38), (35), (33), (31), and (30) via the union bound, with probability at least $1 - Hm^{-2} - \delta$, for any $x \in \mathbb{S}_{d-1}$ and any $h \in [H]$,

$$|\langle g(x; W_0), \bar{W}_h \rangle - \langle g(x; W_0), \hat{W}_h^{lin} - W_0 \rangle| \leq \beta \cdot \|g(x; W_0)\|_{\Lambda_h^{-1}},$$

where

$$\beta := K' \iota_0 \lambda^{-1/2} C_g + \lambda^{1/2} B + (H + \psi) \left[ \sqrt{\tilde{d}_h \log(1 + K'/\lambda) + K' \log \lambda + 2 \log(3H/\delta)} \right]. \tag{39}$$

Combing with Lemma D.3 using the union bound, with probability at least $1 - Hm^{-2} - 2\delta$, for any $x \in \mathbb{S}_{d-1}$, and any $h \in [H]$,

$$\begin{aligned}
f(x; \hat{W}_h) - (\mathbb{B}_h \tilde{V}_{h+1})(x) &\leq \langle g(x; W_0), \hat{W}_h^{lin} - W_0 \rangle + \iota_2 - \langle g(x; W_0), \bar{W}_h \rangle + \iota_0 \\
&\leq \beta \cdot \|g(x; W_0)\|_{\Lambda_h^{-1}} + \iota_2 + \iota_0, \\
&= \beta \cdot \|g(x; W_0)\|_{\Lambda_h^{-1}} + \iota_2 + \iota_0
\end{aligned}$$

where $\iota_2$, and $\beta$ are defined in Table 3.

Similarly, it is easy to show that

$$(\mathbb{B}_h \tilde{V}_{h+1})(x) - f(x; \hat{W}_h) \leq \beta \cdot \|g(x; W_0)\|_{\Lambda_h^{-1}} + \iota_2 + \iota_0.$$

$\square$

## E.2 PROOF OF LEMMA D.2

Before proving Lemma D.2, we prove the following intermediate lemmas. The detailed proofs of these intermediate lemmas are deferred to Section F.

**Lemma E.2.** *Conditioned on all the randomness except $\{\xi_h^{k,i}\}$ and $\{\zeta_h^i\}$, for any $i \in [M]$,*

$$\tilde{W}_h^{i,lin} - \hat{W}_h^{lin} \sim \mathcal{N}(0, \sigma_h^2 \Lambda_h^{-1}).$$

**Lemma E.3.** *If we set $M = \log \frac{HSA}{\delta} / \log \frac{1}{1 - \Phi(-1)}$ where $\Phi(\cdot)$ is the cumulative distribution function of the standard normal distribution, then with probability at least $1 - \delta$, for any $(x, h) \in \mathcal{X} \times [H]$,*

$$\min_{i \in [M]} \langle g(x; W_0), \tilde{W}_h^{i,lin} \rangle \leq \langle g(x; W_0), \hat{W}_h^{lin} \rangle - \sigma_h \|g(x; W_0)\|_{\Lambda_h^{-1}}.$$

We are now ready to prove Lemma D.2.

*Proof of Lemma D.2.* Note that the parameter condition in Equation (20) of Lemma D.2 satisfies Equation (21) of Lemma D.3, thus given the parameter condition Lemma D.2, Lemma D.3 holds. For the rest of the proof, we consider under the joint event in which both the inequality of Lemma D.3 and that of Lemma E.3 hold. By the union bound, probability that this joint event holds is at least $1 - MHm^{-2} - \delta$. Thus, for any $x \in \mathbb{S}_{d-1}$, $h \in [H]$, and $i \in [M]$,

$$\min_{i \in [M]} f(x; \tilde{W}_h^i) - f(x; \hat{W}_h) \leq \min_{i \in [M]} \langle g(x; W_0), \tilde{W}_h^{i,lin} - W_0 \rangle - \langle g(x; W_0), \hat{W}_h^{lin} - W_0 \rangle + \iota_1 + \iota_2$$

$$\leq -\sigma_h \|g(x; W_0)\|_{\Lambda_h^{-1}} + \iota_1 + \iota_2$$

where the first inequality holds due to Lemma D.3, and the second inequality holds due to Lemma E.3. Thus, we have

$$\tilde{Q}_h(x) = \min\{\min_{i \in [M]} f(x; \tilde{W}_h^i), H - h + 1 + \psi\}^+ \leq \max\{\min_{i \in [M]} f(x; \tilde{W}_h^i), 0\}$$

$$\leq \max\{\langle g(x; W_0), \hat{W}_h^{lin} - W_0 \rangle - \sigma_h \|g(x; W_0)\|_{\Lambda_h^{-1}} + \iota_1 + \iota_2, 0\}.$$

$\square$

### E.3 PROOF OF LEMMA D.3

In this subsection, we provide a detailed proof of Lemma D.3. We first provide intermediate lemmas that we use for proving Lemma D.3. The detailed proofs of these intermediate lemmas are deferred to Section F.

The following lemma bounds the the gradient descent weight of the perturbed loss function around the linear weight counterpart.

**Lemma E.4.** *Let*

$$\begin{cases} m = \Omega\left(d^{3/2} R^{-1} \log^{3/2}(\sqrt{m}/R)\right) \\ R = \mathcal{O}\left(m^{1/2} \log^{-3} m\right), \\ \eta \leq (\lambda + K'C_g^2)^{-1}, \\ R \geq \max\{4\tilde{B}_1, 4\tilde{B}_2, 2\sqrt{2\lambda^{-1}K'(H + \psi + \gamma_{h,1})^2 + 4\gamma_{h,2}^2}\}, \end{cases} \tag{40}$$

*where $\tilde{B}_1$, $\tilde{B}_2$, $\gamma_{h,1}$ and $\gamma_{h,2}$ are defined in Table 3 and $C_g$ is a constant given in Lemma G.1. With probability at least $1 - MHm^{-2} - \delta$, for any $(i, j, h) \in [M] \times [J] \times [H]$, we have*

- $\tilde{W}_h^{i,(j)} \in \mathcal{B}(W_0; R)$,

- $\|\tilde{W}_h^{i,(j)} - \tilde{W}_h^{i,lin}\|_2 \leq \tilde{B}_1 + \tilde{B}_2 + \lambda^{-1}(1 - \eta\lambda)^j \left(K'(H + \psi + \gamma_{h,1})^2 + \lambda\gamma_{h,2}^2\right)$

Similar to Lemma E.4, we obtain the following lemma for the gradient descent weights of the non-perturbed loss function.

**Lemma E.5.** *Let*

$$\begin{cases} m = \Omega\left(d^{3/2} R^{-1} \log^{3/2}(\sqrt{m}/R)\right) \\ R = \mathcal{O}\left(m^{1/2} \log^{-3} m\right), \\ \eta \leq (\lambda + K'C_g^2)^{-1}, \\ R \geq \max\{4B_1, 4\tilde{B}_2, 2\sqrt{2\lambda^{-1}K'}(H + \psi)\}, \end{cases} \tag{41}$$

*where $B_1$, $\tilde{B}_2$, $\gamma_{h,1}$ and $\gamma_{h,2}$ are defined in Table 3 and $C_g$ is a constant given in Lemma G.1. With probability at least $1 - MHm^{-2} - \delta$, for any $(i, j, h) \in [M] \times [J] \times [H]$, we have*

- $\hat{W}_h^{(j)} \in \mathcal{B}(W_0; R)$,

- $\|\hat{W}_h^{(j)} - \hat{W}_h^{lin}\|_2 \leq B_1 + \tilde{B}_2 + \lambda^{-1}(1 - \eta\lambda)^j K'(H + \psi)^2$

We now can prove Lemma D.3.

*Proof of Lemma D.3.* Note that Equation (21) implies both Equation (40) of Lemma E.4 and Equation (41) of Lemma E.5, thus both Lemma E.4 and Lemma E.5 holds under Equation (21). Thus, by the union bound, with probability at least $1 - MHm^{-2} - \delta$, for any $(i, j, h) \in [M] \times [J] \times [H]$, and $x \in \mathbb{S}_{d-1}$,

$$|f(x; \tilde{W}_h^{i,(j)}) - \langle g(x; W_0), \tilde{W}_h^{i,lin} - W_0 \rangle|$$
$$\leq |f(x; \tilde{W}_h^{i,(j)}) - \langle g(x; W_0), \tilde{W}_h^{i,(j)} - W_0 \rangle| + |\langle g(x; W_0), \tilde{W}_h^{i,(j)} - \tilde{W}_h^{i,lin} \rangle|$$
$$\leq C_g R^{4/3} m^{-1/6} \sqrt{\log m} + C_g \left( \tilde{B}_1 + \tilde{B}_2 + \lambda^{-1}(1 - \eta\lambda)^j \left( K'(H + \psi + \gamma_{h,1})^2 + \lambda\gamma_{h,2}^2 \right) \right) = \iota_1,$$

where the first inequality holds due to the triangle inequality, the second inequality holds due to Cauchy-Schwarz inequality, Lemma G.1, and Lemma E.4.

Similarly, by the union bound, with probability at least $1 - Hm^{-2}$, for any $(i, j, h) \in [M] \times [J] \times [H]$, and $x \in \mathbb{S}_{d-1}$,

$$|f(x; \hat{W}_h^{(j)}) - \langle g(x; W_0), \hat{W}_h^{lin} - W_0 \rangle| \leq |f(x; \hat{W}_h^{(j)}) - \langle g(x; W_0), \hat{W}_h^{(j)} - W_0 \rangle|$$
$$+ |\langle g(x; W_0), \hat{W}_h^{(j)} - \hat{W}_h^{lin} \rangle|$$
$$\leq C_g R^{4/3} m^{-1/6} \sqrt{\log m} + C_g \left( B_1 + \tilde{B}_2 + \lambda^{-1}(1 - \eta\lambda)^j K'(H + \psi)^2 \right) = \iota_2,$$

where the first inequality holds due to the triangle inequality, the second inequality holds due to Cauchy-Schwarz inequality, Lemma E.5, and Lemma G.1. $\qquad\square$

# F  PROOFS OF LEMMAS IN SECTION E

In this section, we provide the detailed proofs of Lemmas in Section E.

## F.1  PROOF OF LEMMA E.1

*Proof of Lemma E.1.* As $\mathbb{B}_h \tilde{V}_{h+1} \in \mathcal{Q}^*$ by Assumption 5.1, where $\mathcal{Q}^*$ is defined in Section 5, we have

$$\mathbb{B}_h \tilde{V}_{h+1} = \int_{\mathbb{R}^d} c(w)^T x \sigma'(w^T x) dw,$$

for some $c : \mathbb{R}^d \to \mathbb{R}^d$ such that $\sup_w \frac{\|c(w)\|_2}{p_0(w)} \leq B$. The lemma then directly follows from approximation by finite sum (Gao et al., 2019). $\qquad\square$

## F.2  PROOF OF LEMMA E.2

*Proof of Lemma E.2.* Let $\bar{W} := W + \zeta_h^i$ and

$$\bar{L}_h^i(\bar{W}) := \sum_{k \in \mathcal{I}_h} \left( \langle g(x_h^k; W_0), \bar{W} \rangle - \bar{y}_h^{i,k} \right)^2 + \lambda \|\bar{W}\|_2^2,$$

where $\bar{y}_h^k = r_h^k + \tilde{V}_{h+1}(s_{h+1}^k) + \xi^{k,i} + \langle g(x_h^k; W_0), \zeta_h^i \rangle$. We have $\tilde{L}_h^{i,lin}(W) = \bar{L}_h^i(\bar{W})$ and $\arg\max_W \tilde{L}_h^{i,lin}(W) = \arg\max_{\bar{W}} \bar{L}_h^i(\bar{W}) - \zeta_h^i$ as both $\tilde{L}_h^{i,lin}(W)$ and $\bar{L}_h^i$ are convex. Using the regularized least-squares solution,

$$\arg\max_{\bar{W}} \bar{L}_h^i(\bar{W}) = \Lambda_h^{-1} \sum_{k \in \mathcal{I}_h} g(x_h^k; W_0) \bar{y}_h^k$$

$$= \Lambda_h^{-1} \left[ \sum_{k \in \mathcal{I}_h} g(x_h^k; W_0)(r_h^k + \tilde{V}_{h+1}(s_{h+1}^k) + \xi^{k,i}) + \sum_{k \in \mathcal{I}_h} g(x_h^k; W_0)\langle g(x_h^k; W_0), \zeta_h^i \rangle \right]$$

$$= \Lambda_h^{-1} \left[ \sum_{k \in \mathcal{I}_h} g(x_h^k; W_0)(r_h^k + \tilde{V}_{h+1}(s_{h+1}^k) + \xi_h^{k,i}) + \sum_{k \in \mathcal{I}_h} g(x_h^k; W_0)g(x_h^k; W_0)^T \zeta_h^i \right]$$

$$= \Lambda_h^{-1} \left[ \sum_{k \in \mathcal{I}_h} g(x_h^k; W_0)(r_h^k + \tilde{V}_{h+1}(s_{h+1}^k) + \xi_h^{k,i}) + (\Lambda_h - \lambda I_{md})\zeta_h^i \right].$$

Thus, we have

$$\tilde{W}_h^i = \arg\max_W \tilde{L}_h^{i,lin}(W) = \arg\max_{\bar{W}} \bar{L}_h^i(\bar{W}) - \zeta_h^i$$

$$= \Lambda_h^{-1} \left[ \sum_{k \in \mathcal{I}_h} g(x_h^k; W_0)(r_h^k + \tilde{V}_{h+1}(s_{h+1}^k) + \xi_h^{k,i}) + (\Lambda_h - \lambda I_{md})\zeta_h^i \right] - \zeta_h^i$$

$$= \Lambda_h^{-1} \left[ \sum_{k \in \mathcal{I}_h} g(x_h^k; W_0)(r_h^k + \tilde{V}_{h+1}(s_{h+1}^k) + \xi_h^{k,i}) - \lambda\zeta_h^i \right]$$

$$= \hat{W}_h + \Lambda_h^{-1} \left[ \sum_{k \in \mathcal{I}_h} g(x_h^k; W_0)\xi_h^{k,i} - \lambda\zeta_h^i \right]$$

By direct computation, it is easy to see that

$$\tilde{W}_h^i - \hat{W}_h = \Lambda_h^{-1} \left[ \sum_{k \in \mathcal{I}_h} g(x_h^k; W_0)\xi_h^{k,i} - \lambda\zeta_h^i \right] \sim \mathcal{N}(0, \sigma_h^2 \Lambda_h^{-1}).$$

$\square$

### F.3 PROOF OF LEMMA E.3

In this subsection, we provide a proof for E.3. We first provide a bound for the perturbed noises used in Algorithm 1 in the following lemma.

**Lemma F.1.** *There exist absolute constants $c_1, c_2 > 0$ such that for any $\delta > 0$, event $\mathcal{E}(\delta)$ holds with probability at least $1 - \delta$, for any $(k, h, i) \in [K] \times [H] \times [M]$,*

$$|\xi_h^{k,i}| \leq c_1\sigma_h\sqrt{\log(K'HM/\delta)} =: \gamma_{h,1},$$
$$\|\zeta_h^i\|_2 \leq c_2\sigma_h\sqrt{d\log(dK'HM/\delta)} =: \gamma_{h,2}.$$

*Proof of Lemma F.1.* It directly follows from the Gaussian concentration inequality in Lemma G.3 and the union bound. $\square$

We now can prove Lemma E.3.

*Proof of Lemma E.3.* By Lemma E.2,

$$\tilde{W}_h^{i,lin} - \hat{W}_h^{lin} \sim \mathcal{N}(0, \sigma_h^2 \Lambda_h^{-1}).$$

Using the anti-concentration of Gaussian distribution, for any $x = (s, a) \in \mathcal{S} \times \mathcal{A}$ and any $i \in [M]$,

$$\mathbb{P}\left( \langle g(x; W_0), \tilde{W}_h^{i,lin} \rangle \leq \langle g(x; W_0), \hat{W}_h^{lin} \rangle - \sigma_h\|g(x; W_0)\|_{\Lambda_h^{-1}} \right) = \Phi(-1) \in (0, 1).$$

As $\{\tilde{W}_h^{i,lin}\}_{i \in [M]}$ are independent, using the union bound, with probability at least $1 - SAH(1 - \Phi(-1))^M$, for any $x = (s, a) \in \mathcal{S} \times \mathcal{A}$, and $h \in [H]$,

$$\min_{i \in [M]} \langle g(x; W_0), \tilde{W}_h^{i,lin} \rangle \leq \langle g(x; W_0), \hat{W}_h^{lin} \rangle - \sigma_h\|g(x; W_0)\|_{\Lambda_h^{-1}}.$$

Setting $\delta = SAH(1 - \Phi(-1))^M$ completes the proof. $\square$

### F.4 PROOF OF LEMMA E.4

In this subsection, we provide a detailed proof of Lemma E.4. We first prove the following intermediate lemma whose proof is deferred to Subsection F.5.

**Lemma F.2.** *Let*

$$\begin{cases} m = \Omega\left(d^{3/2}R^{-1}\log^{3/2}(\sqrt{m}/R)\right) \\ R = \mathcal{O}\left(m^{1/2}\log^{-3}m\right). \end{cases}$$

*and additionally let*

$$\begin{cases} \eta \le (K'C_g^2 + \lambda/2)^{-1}, \\ \eta \le \frac{1}{2\lambda}. \end{cases}$$

*Then with probability at least $1 - MHm^{-2} - \delta$, for any $(i,j,h) \in [M] \times [J] \times [H]$, if $\tilde{W}_h^{i,(j)} \in \mathcal{B}(W_0; R)$ for any $j' \in [j]$, then*

$$\|f_{j'} - \tilde{y}\|_2 \lesssim \sqrt{K'(H + \psi + \gamma_{h,1})^2 + \lambda\gamma_{h,2}^2 + (\lambda\eta)^{-2}R^{4/3}m^{-1/6}\sqrt{\log m}}.$$

We now can prove Lemma E.4.

*Proof of Lemma E.4.* To simplify the notations, we define

$$\begin{aligned}
\Delta_j &:= \tilde{W}_h^{i,(j)} - \tilde{W}_h^{i,lin,(j)} \in \mathbb{R}^{md}, \\
G_j &:= \left(g(x_h^k; \tilde{W}_h^{i,(j)})\right)_{k \in \mathcal{I}_h} \in \mathbb{R}^{md \times K'}, \\
H_j &:= G_j G_j^T \in \mathbb{R}^{md \times md}, \\
f_j &:= \left(f(x_h^k; \tilde{W}_h^{i,(j)})\right)_{k \in \mathcal{I}_h} \in \mathbb{R}^{K'} \\
\tilde{y} &:= \left(\tilde{y}_h^{i,k}\right)_{k \in \mathcal{I}_h} \in \mathbb{R}^{K'}.
\end{aligned}$$

The gradient descent update rule for $\tilde{W}_h^{i,(j)}$ in Equation (9) can be written as:

$$\tilde{W}_h^{i,(j+1)} = \tilde{W}_h^{i,(j)} - \eta\left[G_j(f_j - \tilde{y}) + \lambda(\tilde{W}_h^{i,(j)} + \zeta_h^i - W_0)\right].$$

The auxiliary updates in Equation (15) can be written as:

$$\tilde{W}_h^{i,lin,(j+1)} = \tilde{W}_h^{i,lin,(j)} - \eta\left[G_0\left(G_0^T(\tilde{W}_h^{i,lin,(j)} - W_0) - \tilde{y}\right) + \lambda(\tilde{W}_h^{i,lin,(j)} + \zeta_h^i - W_0)\right].$$

**Step 1: Proving $\tilde{W}_h^{i,(j)} \in \mathcal{B}(W_0; R)$ for all $j$.** In the first step, we prove by induction that with probability at least $1 - MHm^{-2} - \delta$, for any $(i,j,h) \in [M] \times [J] \times [H]$, we have

$$\tilde{W}_h^{i,(j)} \in \mathcal{B}(W_0; R).$$

In the rest of the proof, we consider under the event that Lemma F.2 holds. Note that the condition in Lemma E.4 satisfies that of Lemma F.2 and under the above event of Lemma F.2, Lemma G.1 and Lemma F.1 both hold. It is trivial that

$$\tilde{W}_h^{i,(0)} = W_0 \in \mathcal{B}(W_0; R).$$

For any fix $j \ge 0$, we assume that

$$\tilde{W}_h^{i,(j')} \in \mathcal{B}(W_0; R), \forall j' \in [j]. \tag{42}$$

We will prove that $\tilde{W}_h^{i,(j+1)} \in \mathcal{B}(W_0; R)$. We have

$$\|\Delta_{j+1}\|_2$$

$$= \left\| (1 - \eta\lambda)\Delta_j - \eta \left[ G_0(f_j - G_0^T(\tilde{W}_h^{i,(j)} - W_0)) + G_0 G_0^T(\tilde{W}_h^{i,(j)} - \tilde{W}_h^{i,lin,(j)}) + (f_j - \tilde{y})(G_j - G_0) \right] \right\|_2$$

$$\leq \underbrace{\|(I - \eta(\lambda I + H_0))\Delta_j\|_2}_{I_1} + \underbrace{\eta\|f_j - \tilde{y}\|_2\|G_j - G_0\|_2}_{I_2} + \underbrace{\eta\|G_0\|_2\|f_j - G_0^T(\tilde{W}_h^{i,(j)} - W_0)\|_2}_{I_3}.$$

We bound $I_1$, $I_2$ and $I_3$ separately.

**Bounding $I_1$.** For bounding $I_1$,

$$\begin{aligned}
I_1 &= \|(I - \eta(\lambda I + H_0))\Delta_j\|_2 \\
&\leq \|I - \eta(\lambda I + H_0)\|_2\|\Delta_j\|_2 \\
&\leq (1 - \eta(\lambda + K'C_g^2))\|\Delta_j\|_2 \\
&\leq (1 - \eta\lambda)\|\Delta_j\|_2
\end{aligned}$$

where the first inequality holds due to the spectral norm inequality, the second inequality holds due to

$$\eta(\lambda I + H_0) \preceq \eta(\lambda + \|G_0\|^2)I \preceq \eta(\lambda + K'C_g^2)I \preceq I,$$

where the first inequality holds due to that $H_0 \preceq \|H_0\|_2 I \preceq \|G_0\|_2^2 I$, the second inequality holds due to that $\|G_0\|_2 \leq \sqrt{K}C_g$ due to Lemma G.1, and the last inequality holds due to the choice of $\eta$ in Equation (40).

**Bounding $I_2$.** For bounding $I_2$,

$$\begin{aligned}
I_2 &= \eta\|f_j - \tilde{y}\|_2\|G_j - G_0\|_2 \\
&\leq \eta\|f_j - \tilde{y}\|_2 \max_{k\in\mathcal{I}_h} \sqrt{K'}\|g(x_h^k; \tilde{W}_h^{i,(j)}) - g(x_h^k; W_0)\|_2 \\
&\leq \eta\|f_j - \tilde{y}\|_2\sqrt{K'}C_g R^{1/3}m^{-1/6}\sqrt{\log m} \\
&\leq \eta\sqrt{2K'(H + \psi + \gamma_{h,1})^2 + \lambda\gamma_{h,2}^2 + 8C_g R^{4/3}m^{-1/6}\sqrt{\log m}}\sqrt{K'}C_g R^{1/3}m^{-1/6}\sqrt{\log m}
\end{aligned}$$

where the first inequality holds due to Cauchy-Schwarz inequality, the second inequality holds due to the induction assumption in Equation (42) and Lemma G.1, and the third inequality holds due to Lemma F.2 and the induction assumption in Equation (42).

**Bounding $I_3$.** For bounding $I_3$,

$$\begin{aligned}
I_3 &= \eta\|G_0\|_2\|f_j - G_0^T(\tilde{W}_h^{i,(j)} - W_0)\|_2 \\
&\leq \eta\sqrt{K'}C_g\sqrt{K'} \max_{k\in\mathcal{I}_h} |f(x_h^k; \tilde{W}_h^{i,(j)}) - g(x_h^k; W_0)^T(\tilde{W}_h^{i,(j)} - W_0)| \\
&\leq \eta K'C_g R^{4/3}m^{-1/6}\sqrt{\log m},
\end{aligned}$$

where the first inequality holds due to Cauchy-Schwarz inequality and due to that $\|G_0\|_2 \leq \sqrt{K'}C_g$ and the second inequality holds due to the induction assumption in Equation (42) and Lemma G.1.

Combining the bounds of $I_1, I_2, I_3$ above, we have

$$\|\Delta_{j+1}\|_2 \leq (1 - \eta\lambda)\|\Delta_j\|_2 + I_2 + I_3.$$

Recursively applying the inequality above for all $j$, we have

$$\|\Delta_j\|_2 \leq \frac{I_2 + I_3}{\eta\lambda} \leq \frac{R}{4} + \frac{R}{4} = \frac{R}{2}, \tag{43}$$

where the second inequality holds due the choice specified in Equation (40). We also have

$$\begin{aligned}
\lambda\|\tilde{W}_h^{i,lin,(j+1)} + \zeta_h^i - W_0\|_2^2 &\leq 2\tilde{L}_h^{i,lin}(\tilde{W}_h^{i,lin,(j+1)}) \\
&\leq 2\tilde{L}_h^{i,lin}(\tilde{W}_h^{i,lin,(0)})
\end{aligned}$$

$$
\begin{aligned}
&= 2\tilde{L}_h^{i,lin}(W_0) \\
&= \sum_{k\in\mathcal{I}_h} \Big( \underbrace{\langle g(x_h^k;W_0),W_0\rangle}_{=0} - \tilde{y}_h^{i,k} \Big)^2 + \lambda\|\zeta_h^i\|_2^2 \\
&= \sum_{k\in\mathcal{I}_h} (\tilde{y}_h^{i,k})^2 + \lambda\|\zeta_h^i\|_2^2 \\
&\le K'(H+\psi+\gamma_{h,1})^2 + \lambda\gamma_{h,2}^2,
\end{aligned}
\tag{44}
$$

where the first inequality holds due the the definition of $\tilde{L}_h^{i,lin}(\tilde{W}_h^{i,lin,(j+1)})$, the second inequality holds due to the monotonicity of $\tilde{L}_h^{i,lin}(W)$ on the gradient descent updates $\{\tilde{W}_h^{i,lin,(j')}\}_{j'}$ for the squared loss on a linear model, the third equality holds due to $\langle g(x_h^k;W_0),W_0\rangle = 0$ from the symmetric initialization scheme, and the last inequality holds due to Lemma F.1. Thus, we have

$$
\begin{aligned}
\|\tilde{W}_h^{i,lin,(j+1)} - W_0\|_2 &\le \sqrt{2\|\tilde{W}_h^{i,lin,(j+1)} + \zeta_h^i - W_0\|_2^2 + \|\zeta_h^i\|_2^2} \\
&\le \sqrt{2\lambda^{-1}K'(H+\psi+\gamma_{h,1})^2 + 4\gamma_{h,2}^2} \\
&\le \frac{R}{2},
\end{aligned}
\tag{45}
$$

where the first inequality holds due to Cauchy-Schwarz inequality, the second inequality holds due to Equation (44) and Lemma F.1, and the last inequality holds due to the choice specified in Equation (40).

Combining Equation (43) and Equation (45), we have

$$
\begin{aligned}
\|\tilde{W}_h^{i,(j+1)} - W_0\|_2 &\le \|\tilde{W}_h^{i,(j+1)} - \tilde{W}_h^{i,lin,(j+1)}\|_2 + \|\tilde{W}_h^{i,lin,(j+1)} - W_0\|_2 \\
&\le \frac{R}{2} + \frac{R}{2} = R,
\end{aligned}
$$

where the first inequality holds due to the triangle inequality.

**Step 2: Bounding** $\|\tilde{W}_h^{i,(j)} - \tilde{W}_h^{i,lin}\|_2$. By the standard result of gradient descent on ridge linear regression, $\tilde{W}_h^{i,(j)}$ converges to $\tilde{W}_h^{i,lin}$ with the convergence rate,

$$
\begin{aligned}
\|\tilde{W}_h^{i,lin,(j)} - \tilde{W}_h^{i,lin}\|_2^2 &\le (1-\eta\lambda)^j \frac{2}{\lambda}(\tilde{L}(W_0) - \tilde{L}(\tilde{W}_h^{i,lin})) \\
&\le (1-\eta\lambda)^j \frac{2}{\lambda}\tilde{L}(W_0) \\
&\le \lambda^{-1}(1-\eta\lambda)^j \left( K'(H+\psi+\gamma_{h,1})^2 + \lambda\gamma_{h,2}^2 \right).
\end{aligned}
$$

Thus, for any $j$, we have

$$
\begin{aligned}
\|\tilde{W}_h^{i,(j)} - \tilde{W}_h^{i,lin}\|_2 &\le \|\tilde{W}_h^{i,(j)} - \tilde{W}_h^{i,lin,(j)}\|_2 + \|\tilde{W}_h^{i,lin,(j)} - \tilde{W}_h^{i,lin}\|_2 \\
&\le (\eta\lambda)^{-1}(I_2+I_3) + \lambda^{-1}(1-\eta\lambda)^j \left( K'(H+\psi+\gamma_{h,1})^2 + \lambda\gamma_{h,2}^2 \right),
\end{aligned}
\tag{46}
$$

where the first inequality holds due to the triangle inequality, the second inequality holds due to Equation (43) and Equation (46). $\qquad\square$

### F.5 Proof of Lemma F.2

*Proof of Lemma F.2.* We bound this term following the proof flow of (Zhou et al., 2020, Lemma C.3) with modifications for different neural parameterization and noisy targets. Suppose that for some fixed $j$,

$$
\tilde{W}_h^{i,(j')} \in \mathcal{B}(W_0;R), \forall j' \in [j].
\tag{47}
$$

Let us define

$$
G(W) := \left(g(x_h^k;W)\right)_{k\in\mathcal{I}_h} \in \mathbb{R}^{md\times K'},
$$

$$f(W) := \left(f(x_h^k; W)\right)_{k \in \mathcal{I}_h} \in \mathbb{R}^{K'},$$
$$e(W', W) := f(W') - f(W) - G(W)^T(W' - W) \in \mathbb{R}.$$

To further simplify the notations in this proof, we drop $i, h$ in $\tilde{L}_h^i(W)$ defined in Equation (8) to write $\tilde{L}_h^i(W)$ as $\tilde{L}(W)$ and write $W_j = W_h^{i,(j)}$, where

$$\tilde{L}(W) = \frac{1}{2} \sum_{k \in \mathcal{I}_h} \left(f(x_h^k; W) - \tilde{y}_h^{i,k}\right)^2 + \frac{\lambda}{2} \|W + \zeta_h^i - W_0\|_2^2$$

$$= \frac{1}{2} \|f(W) - \tilde{y}\|_2^2 + \frac{\lambda}{2} \|W + \zeta_h^i - W_0\|_2^2.$$

Suppose that $W \in \mathbb{B}(W_0; R)$. By that $\|\cdot\|_2^2$ is 1-smooth,

$$\tilde{L}(W') - \tilde{L}(W) \leq \langle f(W) - \tilde{y}, f(W') - f(W)\rangle + \frac{1}{2} \|f(W') - f(W)\|_2^2$$

$$+ \lambda \langle W + \zeta_h^i - W_0, W' - W\rangle + \frac{\lambda}{2} \|W' - W\|_2^2$$

$$= \langle f(W) - \tilde{y}, G(W)^T(W' - W) + e(W', W)\rangle + \frac{1}{2} \|G(W)^T(W' - W) + e(W', W)\|_2^2$$

$$+ \lambda \langle W + \zeta_h^i - W_0, W' - W\rangle + \frac{\lambda}{2} \|W' - W\|_2^2$$

$$= \langle \nabla \tilde{L}(W), W' - W\rangle$$

$$+ \underbrace{\langle f(W) - \tilde{y}, e(W', W)\rangle + \frac{1}{2} \|G(W)^T(W' - W) + e(W', W)\|_2^2 + \frac{\lambda}{2} \|W' - W\|_2^2}_{I_1}. \quad (48)$$

For bounding $I_1$,

$$I_1 \leq \|f(W) - \tilde{y}\|_2 \|e(W', W)\|_2 + K' C_g^2 \|W' - W\|_2^2 + \|e(W', W)\|_2^2 + \frac{\lambda}{2} \|W' - W\|_2^2$$

$$= \|f(W) - \tilde{y}\|_2 \|e(W', W)\|_2 + (K' C_g^2 + \lambda/2)\|W' - W\|_2^2 + \|e(W', W)\|_2^2, \quad (49)$$

where the first inequality holds due to Cauchy-Schwarz inequality, $W \in \mathcal{B}(W_0; R)$ and Lemma G.1. Substituting Equation (49) into Equation (48) with $W' = W - \eta \nabla \tilde{L}(W)$,

$$\tilde{L}(W') - \tilde{L}(W) \leq -\eta(1 - (KC_g^2 + \lambda/2)\eta)\|\nabla \tilde{L}(W)\|_2^2 + \|f(W) - \tilde{y}\|_2 \|e(W', W)\|_2$$

$$+ \|e(W', W)\|_2^2. \quad (50)$$

By the 1-strong convexity of $\|\cdot\|_2^2$, for any $W'$,

$$\tilde{L}(W') - \tilde{L}(W) \geq \langle f(W) - \tilde{y}, f(W') - f(W)\rangle + \lambda \langle W + \zeta_h^i - W_0, W' - W\rangle + \frac{\lambda}{2} \|W' - W\|_2^2$$

$$= \langle f(W) - \tilde{y}, G(W)^T(W' - W) + e(W', W)\rangle + \lambda \langle W + \zeta_h^i - W_0, W' - W\rangle + \frac{\lambda}{2} \|W' - W\|_2^2$$

$$= \langle \nabla \tilde{L}(W), W' - W\rangle + \langle f(W) - \tilde{y}, e(W', W)\rangle + \frac{\lambda}{2} \|W' - W\|_2^2$$

$$\geq -\frac{\|\nabla \tilde{L}(W)\|_2^2}{2\lambda} - \|f(W) - \tilde{y}\|_2 \|e(W', W)\|_2, \quad (51)$$

where the last inequality holds due to Cauchy-Schwarz inequality.

Substituting Equation (51) into Equation (50), for any $W'$,

$$\tilde{L}(W - \eta \nabla \tilde{L}(W)) - \tilde{L}(W)$$

$$\leq \underbrace{2\lambda \eta(1 - (KC_g^2 + \lambda/2)\eta)}_{\alpha} \left(\tilde{L}(W') - \tilde{L}(W) + \|f(W) - \tilde{y}\|_2 \|e(W', W)\|_2\right)$$

$$+ \|f(W) - \tilde{y}\|_2 \|e(W - \eta \nabla \tilde{L}(W), W)\|_2 + \|e(W - \eta \nabla \tilde{L}(W), W)\|_2^2$$

$$\leq \alpha \left( \tilde{L}(W') - \tilde{L}(W) + \frac{\gamma_1}{2} \|f(W) - \tilde{y}\|_2^2 + \frac{1}{2\gamma_1} \|e(W', W)\|_2^2 \right)$$

$$+ \frac{\gamma_2}{2} \|f(W) - \tilde{y}\|_2^2 + \frac{1}{2\gamma_2} \|e(W - \eta \nabla \tilde{L}(W), W)\|_2^2 + \|e(W - \eta \nabla \tilde{L}(W), W)\|_2^2$$

$$\leq \alpha \left( \tilde{L}(W') - \tilde{L}(W) + \gamma_1 \tilde{L}(W) + \frac{1}{2\gamma_1} \|e(W', W)\|_2^2 \right)$$

$$+ \gamma_2 \tilde{L}(W) + \frac{1}{2\gamma_2} \|e(W - \eta \nabla \tilde{L}(W), W)\|_2^2 + \|e(W - \eta \nabla \tilde{L}(W), W)\|_2^2, \tag{52}$$

where the second inequality holds due to Cauchy-Schwarz inequality for any $\gamma_1, \gamma_2 > 0$, and the third inequality holds due to $\|f(W) - \tilde{y}\|_2^2 \leq 2\tilde{L}(W)$.

Rearranging terms in Equation (52) and setting $W = W_j$, $W' = W_0$, $\gamma_1 = \frac{1}{4}$, $\gamma_2 = \frac{\alpha}{4}$,

$$\tilde{L}(W_{j+1}) - \tilde{L}(W_0) \leq (1 - \alpha + \alpha\gamma_1 + \gamma_2)\tilde{L}(W_j) - (1 - \frac{\alpha}{2})\tilde{L}(W_0) + \frac{\alpha}{2}\tilde{L}(W_0)$$

$$+ \frac{\alpha}{2\gamma_1} \|e(W_0, W_j)\|_2^2 + \frac{1}{2\gamma_2} \|e(W_{j+1}, W_j)\|_2^2 + \|e(W_{j+1}, W_j)\|_2^2$$

$$= (1 - \frac{\alpha}{2}) \left( \tilde{L}(W_j) - \tilde{L}(W_0) \right) + \frac{\alpha}{2}\tilde{L}(W_0) + 2\alpha\|e(W_0, W_j)\|_2^2$$

$$+ (1 + \frac{2}{\alpha})\|e(W_{j+1}, W_j)\|_2^2$$

$$\leq (1 - \frac{\alpha}{2}) \left( \tilde{L}(W_j) - \tilde{L}(W_0) \right) + \frac{\alpha}{2}\tilde{L}(W_0) + (1 + \frac{2}{\alpha} + 2\alpha)e, \tag{53}$$

where $e := C_g R^{4/3} m^{-1/6} \sqrt{\log m}$, the last inequality holds due to Equation (47) and Lemma G.1. Applying Equation (53), we have

$$\tilde{L}(W_j) - \tilde{L}(W_0) \leq \frac{2}{\alpha} \left( \frac{\alpha}{2}\tilde{L}(W_0) + (1 + \frac{2}{\alpha} + 2\alpha)e \right).$$

Rearranging the above inequality,

$$\tilde{L}(W_j) \leq 2\tilde{L}(W_0) + (\frac{2}{\alpha} + \frac{4}{\alpha^2} + 4)e$$

where the last inequality holds due to the choice of $\eta$. Finally, we have

$$\|f_j - \tilde{y}\|_2^2 \leq 2\tilde{L}(W_j)$$

and $\tilde{L}(W_0) = \frac{1}{2}\|\tilde{y}\|_2^2 + \frac{\lambda}{2}\|\zeta_h^i\|_2^2 \leq \frac{K'}{2}(H + \psi + \gamma_{h,1})^2 + \frac{\lambda}{2}\gamma_{h,2}^2$ due to Lemma F.1. □

# G   SUPPORT LEMMAS

**Lemma G.1.** *Let* $m = \Omega \left( d^{3/2} R^{-1} \log^{3/2}(\sqrt{m}/R) \right)$ *and* $R = \mathcal{O} \left( m^{1/2} \log^{-3} m \right)$. *With probability at least* $1 - e^{-\Omega(\log^2 m)} \geq 1 - m^{-2}$ *with respect to the random initialization, it holds for any* $W, W' \in \mathcal{B}(W_0; R)$ *and* $x \in \mathbb{S}_{d-1}$ *that*

$$\|g(x; W)\|_2 \leq C_g,$$

$$\|g(x; W) - g(x; W_0)\|_2 \leq \mathcal{O} \left( C_g R^{1/3} m^{-1/6} \sqrt{\log m} \right),$$

$$|f(x; W) - f(x; W') - \langle g(x; W'), W - W' \rangle| \leq \mathcal{O} \left( C_g R^{4/3} m^{-1/6} \sqrt{\log m} \right),$$

*where* $C_g = \mathcal{O}(1)$ *is a constant independent of* $d$ *and* $m$. *Moreover, without loss of generality, we assume* $C_g \leq 1$.

*Proof of Lemma G.1.* Due to (Yang et al., 2020, Lemma C.2) and (Cai et al., 2019, Lemma F.1, F.2), we have the first two inequalities and the following:

$$|f(x; W) - \langle g(x; W_0), W - W_0 \rangle| \leq \mathcal{O} \left( C_g R^{4/3} m^{-1/6} \sqrt{\log m} \right).$$

For any $W, W' \in \mathcal{B}(W_0; R)$,

$$f(x; W) - f(x; W') - \langle g(x; W'), W - W' \rangle$$
$$= f(x; W) - \langle g(x; W_0), W - W_0 \rangle - (f(x; W') - \langle g(x; W_0), W' - W_0 \rangle)$$
$$+ \langle g(x; W_0) - g(x; W'), W_0 - W' \rangle.$$

Thus,

$$|f(x; W) - f(x; W') - \langle g(x; W'), W - W' \rangle|$$
$$\leq |f(x; W) - \langle g(x; W_0), W - W_0 \rangle| + |f(x; W') - \langle g(x; W_0), W' - W_0 \rangle|$$
$$+ \|g(x; W_0) - g(x; W')\|_2 \|W_0 - W'\|_2 \leq \mathcal{O}\left(C_g R^{4/3} m^{-1/6} \sqrt{\log m}\right).$$

$\square$

**Lemma G.2** ((Arora et al., 2019, Theorem 3)). *If $m = \Omega(\epsilon^{-4} \log(1/\delta))$, then for any $x, x' \in \mathcal{X} \subset \mathbb{S}_{d-1}$, with probability at least $1 - \delta$,*

$$|\langle g(x; W_0), g(x', W_0) \rangle - K_{ntk}(x, x')| \leq 2\epsilon.$$

**Lemma G.3.** *Let $X \sim \mathcal{N}(0, a\Lambda^{-1})$ be a d-dimensional normal variable where $a$ is a scalar. There exists an absolute constant $c > 0$ such that for any $\delta > 0$, with probability at least $1 - \delta$,*

$$\|X\|_\Lambda \leq c\sqrt{da \log(d/\delta)}.$$

*For $d = 1$, $c = \sqrt{2}$.*

**Lemma G.4** (A variant of Hoeffding-Azuma inequality). *Suppose $\{Z_k\}_{k=0}^\infty$ is a real-valued stochastic process with corresponding filtration $\{\mathcal{F}_k\}_{k=0}^\infty$, i.e. $\forall k$, $Z_k$ is $\mathcal{F}_k$-measurable. Suppose that for any $k$, $\mathbb{E}[|Z_k|] < \infty$ and $|Z_k - \mathbb{E}[Z_k|\mathcal{F}_{k-1}]| \leq c_k$ almost surely. Then for all positive $n$ and $t$, we have:*

$$\mathbb{P}\left(\left|\sum_{k=1}^n Z_k - \sum_{k=1}^n \mathbb{E}[Z_k|\mathcal{F}_{k-1}]\right| \geq t\right) \leq 2\exp\left(\frac{-t^2}{\sum_{i=1}^n c_i^2}\right).$$

**Lemma G.5** ((Chowdhury & Gopalan, 2017, Theorem 1)). *Let $\mathcal{H}$ be an RKHS defined over $\mathcal{X} \subseteq \mathbb{R}^d$. Let $\{x_t\}_{t=1}^\infty$ be a discrete time stochastic process adapted to filtration $\{\mathcal{F}_t\}_{t=0}^\infty$. Let $\{Z_k\}_{k=1}^\infty$ be a real-valued stochastic process such that $Z_k \in \mathcal{F}_k$, and $Z_k$ is zero-mean and $\sigma$-sub Gaussian conditioned on $\mathcal{F}_{k-1}$. Let $E_k = (Z_1, \ldots, Z_{k-1})^T \in \mathbb{R}^{k-1}$ and $\mathcal{K}_k$ be the Gram matrix of $\mathcal{H}$ defined on $\{x_t\}_{t \leq k-1}$. For any $\rho > 0$ and $\delta \in (0, 1)$, with probability at least $1 - \delta$,*

$$E_k^T\left[(\mathcal{K}_k + \rho I)^{-1} + I\right]^{-1} E_k \leq \sigma^2 \text{logdet}\left[(1 + \rho)I + \mathcal{K}_k\right] + 2\sigma^2 \log(1/\delta).$$

**Lemma G.6.** *For any matrices $A$ and $B$ where $A$ is invertible,*

$$\text{logdet}(A + B) \leq \text{logdet}(A) + tr(A^{-1}B).$$

**Lemma G.7.** *For any invertible matrices $A, B$,*

$$\|A^{-1} - B^{-1}\|_2 \leq \frac{\|A - B\|_2}{\lambda_{\min}(A)\lambda_{\min}(B)}.$$

*Proof of Lemma G.7.* We have:

$$\|A^{-1} - B^{-1}\|_2 = \|(AB)^{-1}(AB)(A^{-1} - B^{-1})\|_2$$
$$= \|(AB)^{-1}(ABA^{-1} - A)\|_2$$
$$\leq \|(AB)^{-1}\|_2 \|ABA^{-1} - A\|_2$$
$$= \|(AB)^{-1}\|_2 \|ABA^{-1} - AAA^{-1}\|_2$$
$$= \|(AB)^{-1}\|_2 \|A(B - A)A^{-1}\|_2$$
$$= \|(AB)^{-1}\|_2 \|B - A\|_2$$
$$\leq \lambda_{\max}(A^{-1})\lambda_{\max}(B^{-1})\|_2 \|B - A\|_2.$$

$\square$

**Lemma G.8** (Freedman's inequality (Tropp, 2011)). *Let $\{X_k\}_{k=1}^n$ be a real-valued martingale difference sequence with the corresponding filtration $\{\mathcal{F}_k\}_{k=1}^n$, i.e. $X_k$ is $\mathcal{F}_k$-measurable and $\mathbb{E}[X_k|\mathcal{F}_{k-1}] = 0$. Suppose for any $k$, $|X_k| \leq M$ almost surely and define $V := \sum_{k=1}^n \mathbb{E}\left[X_k^2|\mathcal{F}_{k-1}\right]$. For any $a, b > 0$, we have:*

$$\mathbb{P}\left(\sum_{k=1}^n X_k \geq a, V \leq b\right) \leq \exp\left(\frac{-a^2}{2b + 2aM/3}\right).$$

*In an alternative form, for any $t > 0$, we have:*

$$\mathbb{P}\left(\sum_{k=1}^n X_k \geq \frac{2Mt}{3} + \sqrt{2bt}, V \leq b\right) \leq e^{-t}.$$

**Lemma G.9** (Improved online-to-batch argument Nguyen-Tang et al. (2023)). *Let $\{X_k\}$ be any real-valued stochastic process adapted to the filtration $\{\mathcal{F}_k\}$, i.e. $X_k$ is $\mathcal{F}_k$-measurable. Suppose that for any $k$, $X_k \in [0, H]$ almost surely for some $H > 0$. For any $K > 0$, with probability at least $1 - \delta$, we have:*

$$\sum_{k=1}^K \mathbb{E}[X_k|\mathcal{F}_{k-1}] \leq 2\sum_{k=1}^K X_k + \frac{16}{3}H\log(\log_2(KH)/\delta) + 2.$$

*Proof of Lemma G.9.* Let $Z_k = X_k - \mathbb{E}[X_k|\mathcal{F}_{k-1}]$ and $f(K) = \sum_{k=1}^K \mathbb{E}[X_k|\mathcal{F}_{k-1}]$. We have $Z_k$ is a real-valued difference martingale with the corresponding filtration $\{\mathcal{F}_k\}$ and that

$$V := \sum_{k=1}^K \mathbb{E}\left[Z_k^2|\mathcal{F}_{k-1}\right] \leq \sum_{k=1}^K \mathbb{E}\left[X_k^2|\mathcal{F}_{k-1}\right] \leq H\sum_{k=1}^K \mathbb{E}[X_k|\mathcal{F}_{k-1}] = Hf(K).$$

Note that $|Z_k| \leq H$ and $f(K) \in [0, KH]$ and let $m = \log_2(KH)$. Also note that $f(K) = \sum_{k=1}^K X_k - \sum_{k=1}^K Z_k \geq -\sum_{k=1}^K Z_k$. Thus if $\sum_{k=1}^K Z_k \leq -1$, we have $f(K) \geq 1$. For any $t > 0$, leveraging the peeling technique (Bartlett et al., 2005), we have:

$$\mathbb{P}\left(\sum_{k=1}^K Z_k \leq -\frac{2Ht}{3} - \sqrt{4Hf(K)t} - 1\right)$$

$$= \mathbb{P}\left(\sum_{k=1}^K Z_k \leq -\frac{2Ht}{3} - \sqrt{4Hf(K)t} - 1, f(K) \in [1, KH]\right)$$

$$\leq \sum_{i=1}^m \mathbb{P}\left(\sum_{k=1}^K Z_k \leq -\frac{2Ht}{3} - \sqrt{4Hf(K)t} - 1, f(K) \in [2^{i-1}, 2^i)\right)$$

$$\leq \sum_{i=1}^m \mathbb{P}\left(\sum_{k=1}^K Z_k \leq -\frac{2Ht}{3} - \sqrt{4H2^{i-1}t} - 1, V \leq H2^i, f(K) \in [2^{i-1}, 2^i)\right)$$

$$\leq \sum_{i=1}^m \mathbb{P}\left(\sum_{k=1}^K Z_k \leq -\frac{2Ht}{3} - \sqrt{2H2^it}, V \leq H2^i\right)$$

$$\leq \sum_{i=1}^m e^{-t} = me^{-t},$$

where the first equation is by that $\sum_{k=1}^K Z_k \leq -\frac{2Ht}{3} - \sqrt{4Hf(K)t} - 1 \leq -1$ thus $f(K) \geq 1$, the second inequality is by that $V \leq Hf(K)$, and the last inequality is by Lemma G.8. Thus, with probability at least $1 - me^{-t}$, we have:

$$\sum_{k=1}^K X_k - f(K) = \sum_{k=1}^K Z_k \geq -\frac{2Ht}{3} - \sqrt{4Hf(K)t} - 1.$$

The above inequality implies that $f(K) \leq 2\sum_{k=1}^K X_k + 4Ht/3 + 2 + 4Ht$, due to the simple inequality: if $x \leq a\sqrt{x} + b$, $x \leq a^2 + 2b$. Then setting $t = \log(m/\delta)$ completes the proof. □

# H   BASELINE ALGORITHMS

For completeness, we give the definition of linear MDPs as follows.

**Definition 4** (Linear MDPs (Yang & Wang, 2019; Jin et al., 2020)). *An MDP has a linear structure if for any $(s, a, s', h)$, we have:*

$$r_h(s, a) = \phi_h(s, a)^T \theta_h, \mathbb{P}_h(s'|s, a) = \phi_h(s, a)^T \mu_h(s'),$$

*where $\phi : \mathcal{S} \times \mathcal{A} \to \mathbb{R}^{d_{lin}}$ is a known feature map, $\theta_h \in \mathbb{R}^{d_{lin}}$ is an unknown vector, and $\mu_h : \mathcal{S} \to \mathbb{R}^{d_{lin}}$ are unknown signed measures.*

We also give the details of the baseline algorithms: LinLCB in Algorithm 3, LinGreedy in Algorithm 4, Lin-VIPeR in Algorithm 5, NeuraLCB in Algorithm 6 and NeuralGreedy in Algorithm 7. For simplicity, we do not use data split in these algorithms presented here.

---

**Algorithm 3** LinLCB (Jin et al., 2021)

1: **Input**: Offline data $\mathcal{D} = \{(s_h^k, a_h^k, r_h^k)\}_{h \in [H]}^{k \in [K]}$, uncertainty multiplier $\beta$, regularization parameter $\lambda$.
2: Initialize $\tilde{V}_{H+1}(\cdot) \leftarrow 0$
3: **for** $h = H, \dots, 1$ **do**
4:     $\Lambda_h \leftarrow \sum_{k=1}^{K} \phi(s_h^k, a_h^k) \phi(s_h^k, a_h^k)^T + \lambda I$
5:     $\hat{\theta}_h \leftarrow \Sigma_h^{-1} \sum_{k=1}^{K} \phi_h(s_h^k, a_h^k) \cdot (r_h^k + \hat{V}_{h+1}(s_{h+1}^k))$
6:     $b_h(\cdot, \cdot) \leftarrow \beta \cdot \|\phi_h(\cdot, \cdot)\|_{\Sigma_h^{-1}}$.
7:     $\hat{Q}_h(\cdot, \cdot) \leftarrow \min\{\langle \phi_h(\cdot, \cdot), \hat{\theta}_h \rangle - b_h(\cdot, \cdot), H - h + 1\}^+$.
8:     $\hat{\pi}_h \leftarrow \arg\max_{\pi_h} \langle \hat{Q}_h, \pi_h \rangle$ and $\hat{V}_h^k \leftarrow \langle \hat{Q}_h^k, \pi_h^k \rangle$.
9: **end for**
10: **Output**: $\hat{\pi} = \{\hat{\pi}_h\}_{h \in [H]}$

---

**Algorithm 4** LinGreedy

1: **Input**: Offline data $\mathcal{D} = \{(s_h^k, a_h^k, r_h^k)\}_{h \in [H]}^{k \in [K]}$, perturbed variances $\{\sigma_h\}_{h \in [H]}$, number of bootstraps $M$, regularization parameter $\lambda$.
2: Initialize $\tilde{V}_{H+1}(\cdot) \leftarrow 0$
3: **for** $h = H, \dots, 1$ **do**
4:     $\hat{\theta}_h \leftarrow \Sigma_h^{-1} \sum_{k=1}^{K} \phi_h(s_h^k, a_h^k) \cdot (r_h^k + \hat{V}_{h+1}(s_{h+1}^k))$
5:     $\hat{Q}_h(\cdot, \cdot) \leftarrow \min\{\langle \phi_h(\cdot, \cdot), \hat{\theta}_h \rangle, H - h + 1\}^+$.
6:     $\hat{\pi}_h \leftarrow \arg\max_{\pi_h} \langle \hat{Q}_h, \pi_h \rangle$ and $\hat{V}_h^k \leftarrow \langle \hat{Q}_h^k, \pi_h^k \rangle$.
7: **end for**
8: **Output**: $\hat{\pi} = \{\hat{\pi}_h\}_{h \in [H]}$

---

---

**Algorithm 5** Lin-VIPeR

---

1: **Input**: Offline data $\mathcal{D} = \{(s_h^k, a_h^k, r_h^k)\}_{h \in [H]}^{k \in [K]}$, perturbed variances $\{\sigma_h\}_{h \in [H]}$, number of bootstraps $M$, regularization parameter $\lambda$.
2: Initialize $\tilde{V}_{H+1}(\cdot) \leftarrow 0$
3: **for** $h = H, \ldots, 1$ **do**
4: $\quad \Lambda_h \leftarrow \sum_{k=1}^K \phi(s_h^k, a_h^k)\phi(s_h^k, a_h^k)^T + \lambda I$
5: $\quad$ **for** $i = 1, \ldots, M$ **do**
6: $\qquad$ Sample $\{\xi_h^{\tau,i}\}_{\tau \in [K]} \sim \mathcal{N}(0, \sigma_h^2)$ and $\zeta_h^i = \{\zeta_h^{j,i}\}_{j \in [d]} \sim \mathcal{N}(0, \sigma_h^2 I_d)$
7: $\qquad$ Solve the perturbed regularized least-squares regression:

$$\tilde{\theta}_h^i \leftarrow \arg\max_{\theta \in \mathbb{R}^d} \sum_{k=1}^K \left( \langle \phi(s_h^k, a_h^k), \theta \rangle - (r_h^k + \tilde{V}_{h+1}(s_{h+1}^k) + \xi_h^{k,i}) \right)^2 + \lambda \|\theta + \zeta_h^i\|_2^2,$$

8: $\quad$ **end for**
9: $\quad$ Compute $\tilde{Q}_h(\cdot, \cdot) \leftarrow \min\{\min_{i \in [M]} \langle \phi(\cdot, \cdot), \tilde{\theta}_h^i \rangle, H - h + 1\}^+$
10: $\quad \tilde{\pi}_h \leftarrow \arg\max_{\pi_h} \langle \tilde{Q}_h, \pi_h \rangle$ and $\tilde{V}_h \leftarrow \langle \tilde{Q}_h, \tilde{\pi}_h \rangle$
11: **end for**
12: **Output**: $\tilde{\pi} = \{\tilde{\pi}_h\}_{h \in [H]}$

---

**Algorithm 6** NeuraLCB (a modification of (Nguyen-Tang et al., 2022a))

---

1: **Input**: Offline data $\mathcal{D} = \{(s_h^k, a_h^k, r_h^k)\}_{h \in [H]}^{k \in [K]}$, neural networks $\mathcal{F} = \{f(\cdot, \cdot; W) : W \in \mathcal{W}\} \subset \{\mathcal{X} \rightarrow \mathbb{R}\}$, uncertainty multiplier $\beta$, regularization parameter $\lambda$, step size $\eta$, number of gradient descent steps $J$
2: Initialize $\tilde{V}_{H+1}(\cdot) \leftarrow 0$ and initialize $f(\cdot, \cdot; W)$ with initial parameter $W_0$
3: **for** $h = H, \ldots, 1$ **do**
4: $\quad \hat{W}_h \leftarrow \text{GradientDescent}(\lambda, \eta, J, \{(s_h^k, a_h^k, r_h^k)\}_{k \in [K]}, 0, W_0)$ (Algorithm 2)
5: $\quad \Lambda_h \leftarrow \lambda I + \sum_{k=1}^K g(s_h^k, a_h^k; \hat{W}_h)g(x_h^k; \hat{W}_h)^T$
6: $\quad$ Compute $\hat{Q}_h(\cdot, \cdot) \leftarrow \min\{f(\cdot, \cdot; \hat{W}_h) - \beta\|g(\cdot, \cdot; \hat{W}_h)\|_{\Lambda_h^{-1}}, H - h + 1\}^+$
7: $\quad \hat{\pi}_h \leftarrow \arg\max_{\pi_h} \langle \hat{Q}_h, \pi_h \rangle$ and $\hat{V}_h \leftarrow \langle \hat{Q}_h, \hat{\pi}_h \rangle$
8: **end for**
9: **Output**: $\hat{\pi} = \{\hat{\pi}_h\}_{h \in [H]}$.

---

**Algorithm 7** NeuralGreedy

---

1: **Input**: Offline data $\mathcal{D} = \{(s_h^k, a_h^k, r_h^k)\}_{h \in [H]}^{k \in [K]}$, neural networks $\mathcal{F} = \{f(\cdot, \cdot; W) : W \in \mathcal{W}\} \subset \{\mathcal{X} \rightarrow \mathbb{R}\}$, uncertainty multiplier $\beta$, step size $\eta$, number of gradient descent steps $J$
2: Initialize $\tilde{V}_{H+1}(\cdot) \leftarrow 0$ and initialize $f(\cdot, \cdot; W)$ with initial parameter $W_0$
3: **for** $h = H, \ldots, 1$ **do**
4: $\quad \hat{W}_h \leftarrow \text{GradientDescent}(\lambda, \eta, J, \{(s_h^k, a_h^k, r_h^k)\}_{k \in [K]}, 0, W_0)$ (Algorithm 2)
5: $\quad$ Compute $\hat{Q}_h(\cdot, \cdot) \leftarrow \min\{f(\cdot, \cdot; \hat{W}_h), H - h + 1\}^+$
6: $\quad \hat{\pi}_h \leftarrow \arg\max_{\pi_h} \langle \hat{Q}_h, \pi_h \rangle$ and $\hat{V}_h \leftarrow \langle \hat{Q}_h, \hat{\pi}_h \rangle$
7: **end for**
8: **Output**: $\hat{\pi} = \{\hat{\pi}_h\}_{h \in [H]}$.

---

