# OpenReview forum: "VIPeR: Provably Efficient Algorithm for Offline RL with Neural Function Approximation"
_ICLR.cc/2023/Conference — ICLR 2023 notable top 25%_

### Official Review · Reviewer_EhDB · 2022-10-22

**Confidence:** 4
**Correctness:** 4
**Technical Novelty And Significance:** 3
**Empirical Novelty And Significance:** 3
**Recommendation:** 8

**Clarity, Quality, Novelty And Reproducibility:**

This paper is of high quality in general. The theoretical analysis and empirical results are solid.

The paper is clearly written and properly posited in the literature.

Though the idea of randomized rewards and pessimism have appeared in the literature, the idea of combining them and the theoretical / practical considerations therein are fresh.

**Details Of Ethics Concerns:**

I have no ethics concerns.

**Strength And Weaknesses:**

Strengths:

1. A novel algorithm to pessimism.

This work contributes to the literature of offline RL by proposing a solid and practical approach to constructing confidence lower bounds. This idea is clever and refreshing. It is both practically useful and theoretically interesting.

2. Solid theoretical results.

This paper provides solid theoretical analysis. It is quite invovled but I think it will be useful for future works in the literature.


Weaknesses:

1. Clarity of results.

A drawback (which I think can be resolved to some extent) is that although the overall idea is clean, the presentation is way too overwhelming. For example, it will help the readers if important steps in the algorithms can be explained / highlighed / commented, or given more wording in explaining their roles. Also, Theorem 1 poses a huge challenge to the whole flow as the conditions are too complicated. I will suggest replacing it with an informal theorem, and put all these conditions to the appendix.

2. Relation to the literature.

Although the authors generally did a good job in relating to the literature, more discussion on the novelty of this idea can further improve this work and posit this paper appropriately in the literature. For instance, 1) how does the construction of uncertainty quantification differ from that in the online setting (e.g., Jia et al. 2022)? Is there special tricks for dealing with distribution shift? 2) how does the analysis of 2-layer neural networks rely on existing ones, and which parts are specific to the new offline setting? etc.

Questions:

1. Since $\mathcal{Q}^*$ is used for fitting the networks, I am curious about the definition of $\mathcal{Q}^*$. Is it equivalent to the class of 2-layer neural networks with ReLU activation, or is it a superset of it?

2. What is the relationship between $\mathcal{H}_{ntk}$ and the class of 2-layer NN with ReLU activation? Please clarify this to help the readers.

3. The conditions in Theorem 1 are too lengthy. Is it possible to reduce it to a cleaner version (potentially with some harmless relaxation)?

4. It will help if there is a sketch or overview of the key theoretical techniques in Theorem 1. For example, why do the randomized rewards lead to a valid uncertainty quantifier.

Minor issues:

1. The last sentence in Section 4 is difficult to read. What does it mean by `` with a provable implicit uncertainty quantifier and $\tilde{\mathcal{O}}(1/\sqrt{K})$''?

2. Remark 1 is a bit difficult to read. What does it mean by ``data-dependent quantity that measures the number principle dimensions over which the.."?

3. What does $\sigma'$ mean in Definition 1? Please provide a formal definition.



**Summary Of The Paper:**

This paper proposes a novel approach to pessimism-based offline RL. Instead of the standard practice of explicitly constructing a lower confidence bound for value functions, this new approach uses perturbed rewards to implicitly quantify the training uncertainty and construct lower confidence bounds for the ground truth. Such an approach is argued to improve the practicality of pessimistic offline RL especially in situations where the value functions are approximated by large neural networks, whence the large scale model elucidates traditional theoretical analysis. The theoretical property of this approach is studied in overparametrized neural networks with gradient descent training. Empirical experiments are conducted to show the favorable performance of the proposed approach.

**Summary Of The Review:**

I find this to be a strong paper. The idea of using randomized rewards in pessimism-based offline RL is clever, and the theoretical analysis is nontrivial and solid. Empirical studies are conducted to support the arguments. However, I think the presentation of the results can be improved, and more discussion on the relation to the literature is need.

---

> ### Author Response · Authors · 2022-11-09
> **Response to Reviewer EhDB (Part 1)**
>
> Thank you for your positive comments and constructive feedback.
>
> ___
>
> > Clarity of the results, and your Q3 (Cleaner version of Theorem 1)
>
> **Response**: Thanks for the suggestion. We have revised this part of the paper to increase its readability.
>
> ___
>
> > Relation to the literature
>
> **Response**: In what follows, we provide a more detailed discussion when placing our technical contribution in the context of the related literature. Our technical result starts with the value difference lemma in Jin et al. (2021) to connect bounding the suboptimality of an offline algorithm to controlling the uncertainty quantification in the value estimates. Thus, our key technical contribution is to provably quantify the uncertainty of the perturbed value function estimates which were obtained via reward perturbation and gradient descent. This problem setting is largely different from the current analysis of overparameterized neural networks for supervised learning which does not require uncertainty quantification.
>
> Our work is not the first to consider uncertainty quantification with overparameterized neural networks, since it has been studied in Zhou et al. (2020); Nguyen-Tang et al. (2022); Jie et al. (2022). However, there are significant technical differences between our work and these works. The works of Zhou et al. (2020); Nguyen-Tang et al. (2022) consider contextual bandits with overparameterized neural networks trained by (S)GD and quantify the uncertainty of the value function with explicit empirical covariance matrices. We consider general MDP and use reward perturbation to implicitly obtain uncertainty, thus requiring different proof techniques.
>
> Jie et al. (2022) are more related to our work since they consider reward perturbation with overparameterized neural networks (but they consider contextual bandits). However, our reward perturbation strategy is largely different from that of Jie et al. (2022). Specifically, Jie et al. (2022) perturb each reward only once while we perturb each reward multiple times, where the number of perturbing times is crucial in our work and needs to be controlled carefully. We show in Theorem 1  that our reward perturbation strategy is effective in enforcing sufficient pessimism for offline learning in general MDP and the empirical results in Figure 2, Figure 3, Figure 5, and Table 2 are strongly consistent with our theoretical suggestion. Thus, our technical proofs are largely different from those of Jie et al. (2022).
>
> Finally, the idea of perturbing rewards multiple times in our algorithm is inspired by Ishfaq et al. (2021). However, Ishfaq et al. (2021) consider reward perturbation for obtaining optimism in online RL. While perturbing rewards are intuitive to obtain optimism for online RL, for offline RL, under distributional shift, it can be paradoxically difficult to properly obtain pessimism with randomization and ensemble (Ghasemipour et al., 2022b), especially with neural function approximation. We show affirmatively in our work that simply taking the minimum of the randomized value functions after perturbing rewards multiple times is sufficient to obtain provable pessimism for offline RL. In addition, Ishfaq et al. (2021) do not consider neural network function approximation and optimization. Controlling the uncertainty of randomization (via reward perturbation) under neural networks with extra optimization errors induced by gradient descent sets our technical proof significantly apart from that of Ishfaq et al. (2021).
>
> Besides all these differences, in this work, we propose an intricately-designed data splitting technique that avoids the uniform convergence argument and could be of independent interest for studying sample-efficient RL with complex function approximation.
>
> We have added this discussion as Section C.1 in our revision.
>
> ___
> > Q1, Q2: The relationship between $\mathcal{Q}^*$, $\mathcal{H}_{ntk}$ and neural net
>
> **Response**: $\mathcal{Q}^*$ is a dense subset of the RKHS $\mathcal{H}_{ntk}$ which is induced by the NTK kernel (Definition 1) corresponding to the infinite width neural network.
>
> This RKHS  $\mathcal{H}_{ntk}$ is in turn dense in the space of continuous functions since the NTK kernel for ReLU net is a universal kernel. We have added this remark to our revision.

---

> > ### Author Response · Authors · 2022-11-09
> > **Response to Reviewer EhDB (Part 2)**
> >
> > > Q4: Sketch proof
> >
> > The key steps for proving Theorem 1 and Theorem 2 are highlighted in Subsection C.2 and Subsection C.3, respectively. Here, we discuss an overview of our proof strategy. The key technical challenge in our proof is to quantify the uncertainty of the perturbed value function estimates. To deal with this, we carefully control both the near-linearity of neural networks in the NTK regime and the estimation error induced by reward perturbing. A key result that we use to control the linear approximation to the value function estimates is Lemma D.3. The technical challenge in establishing Lemma D.3 is how to carefully control and propagate the optimization error incurred by gradient descent. The complete proof of Lemma D.3 is provided in Section E.3.
> >
> >  The implicit uncertainty quantifier induced by the reward perturbation is established in Lemma D.1 and Lemma D.2, where we carefully design a series of intricate auxiliary loss functions and establish the anti-concentration of the perturbed value function estimates. This requires a careful design of the variance of the noises injected into the rewards.
> >
> > To deal with removing a potentially large covering number when we quantify the implicit uncertainty, we propose our data splitting technique which is validated in the proof of Lemma D.1. in Section E.1.  Moreover, establishing Lemma D.1 in the overparameterization regime induces an additional challenge since a standard analysis would result in a vacuous bound that scales with the overparameterization. We avoid this issue by carefully incorporating the use of the effective dimension in Lemma D.1.
> >
> > We have added this discussion as Section C.1 in our revision.
> >
> > ___
> > > Minor issues
> >
> > We have revised our paper to include your suggestion. Here $\sigma’(x) = 1[{x \geq 0}]$.

---

### Official Review · Reviewer_Svop · 2022-10-24

**Confidence:** 4
**Clarity, Quality, Novelty And Reproducibility:** The writing of this paper is clear an…
**Correctness:** 4
**Technical Novelty And Significance:** 3
**Empirical Novelty And Significance:** 3
**Recommendation:** 8

**Strength And Weaknesses:**

Strength: This paper combines the randomized value function idea and the pessimism principle. In addition, this paper proposes a novel data splitting technique that helps remove the dependence on the potentially large log covering number in the learning bound. The authors empirically corroborate the statistical and computational efficiency of our proposed algorithm in a wide set of synthetic and real-world datasets.

Weakness: 1. Your assumption 5.1 requires for any $H+1$-bounded function, Bellman update maps it to $Q^*$. What will happen if your function has $\epsilon$-misspecified error, i.e. $\inf_{V}\sup_{||V'||_\infty\leq H+1}\|\|B_h V-V'\|\|_\infty \leq \epsilon$, how will the $\epsilon$ model error affect your results? You don't need to derive the result for this case but explain in a few sentences is fine.

2. While Appendix B.1 already provide a nice comparison with Xu&Liang,2022, they consider a different setting where only trajectory reward is observed. If the per-step award is aware, how would their result be like and how would it compare to PERVI? I am asking this since in your setting per-step reward is aware, so I am wondering which of the two methods would be better under this case.

Minor: The recent paper https://arxiv.org/pdf/2210.00750.pdf also considers the pessimism offline RL with parametric function class. How is your result compared to theirs since both results contain the similar measurement $\sum_{h=1}^H ||g(x;W_0)||_{\Lambda_h^{-1}}$ in the suboptimality bound?


**Summary Of The Paper:**

This paper proposes the PEturbed-Reward Value Iteration (PERVI) which combines the randomized value function idea with the pessimism principle. PERVI only needs $O(1)$ time complexity for action selection while LCB-based algorithms require at least $\Omega(K^2)$, where $K$ is the total number of trajectories in the offline data. It proposes a novel data splitting technique that helps remove the potentially large log covering number in the learning bound. PERVI yields a provable uncertainty quantifier with overparameterized neural networks and achieves an $\tilde{O}(\frac{\kappa H^{5/2}\tilde{d}}{\sqrt{K}})$ sub-optimality. statistical and computational efficiency of PERVI is validated with an empirical evaluation in a wide set of synthetic and real-world datasets.

**Summary Of The Review:**

This paper overall has a high quality and provides complete study of neural network approximation for offline RL. If the authors can address my concern in above, I am happy to keep my score.

---

> ### Author Response · Authors · 2022-11-09
> **Response to Reviewer Svop**
>
> Thank you for your positive comments and constructive feedback.
>
> ___
> > Model error $\epsilon$
>
> **Response**: We can include $\epsilon$ into our error propagation analysis which should incur a $\mathcal{O}(\epsilon)$ term as an additional approximation error in our final bound.
>
> ___
>
> > Clarifying comparison with Xu&Liang,2022
>
> **Response**: The per-step award unawareness in Xu&Liang,2022 incurs an extra uncertainty quantifier parameter $\beta_2$ (in their Theorem 1). $\beta_2$ emerges due to the fact that Xu&Liang,2022 need to handle the uncertainty of per-step reward estimation. Consequently, their bound scales with $\max ( \beta_1, \beta_2 )$, as compared to only $\beta_1$ if they consider per-step award awareness. Corollary 1 in Xu&Liang,2022 points out that $\beta_1$ and $\beta_2$ scale with the same order; so their bound would be the same form if they consider per-step award awareness. That suggests that our comparison with Xu&Liang,2022 is valid even when Xu&Liang,2022 consider per-step award awareness.
>
> ___
> > The recent paper [https://arxiv.org/pdf/2210.00750.pdf](https://arxiv.org/pdf/2210.00750.pdf)
>
> **Response**: Thanks for pointing us to this interesting paper. The referenced paper studies provably efficient offline RL with a general parametric function approximation that unifies the guarantees of offline RL in linear and generalized linear MDPs, and beyond with potential applications to other classes of functions in practice. We remark that the result in the referenced paper is orthogonal/complementary to our paper since they consider the parametric class with third-time differentiability which cannot apply to neural networks with non-smooth activation such as ReLU. In addition, different from our work, they do not consider reward perturbation in their algorithmic design or optimization errors in their analysis. We have discussed this referenced paper in our revision in Section B.1

---

### Official Review · Reviewer_69m1 · 2022-10-24

**Confidence:** 3
**Clarity, Quality, Novelty And Reproducibility:** Overall I think the paper scores high…
**Correctness:** 3
**Technical Novelty And Significance:** 3
**Empirical Novelty And Significance:** 2
**Recommendation:** 8

**Strength And Weaknesses:**

There are several things to like about this paper:
- The problem of RL and decision making with offline data is an important one for the community.
- The PERVI algorithm passes a "sanity check" intuitively... by this I mean that it's not just an algorithm for a proof... but you have a sense this is something close to something someone would actually want to use.
- The quality of the writing and presentation in the paper overall is very high.
- The paper has a progression of intuition, to hard theoretical guarantees in simple settings, to empirical success in more complex settings. I like the way of analysing problems with overparameterized NTK!
- Discussion of related work and the key intuitions for the approach appear to be pretty comprehensive for a short paper, although I am likely missing important pieces.

There are some places where the paper probably could be further strengthened:
- In some sense, many of the results and analyses are sort of incremental. The application of randomized value functions plus pessimism has existed before, but this is a slightly new twist on that as opposed to a "game changing" new perspective.
- Some of the ways the Theorems are presented are really messy... you need to read through lines and lines of bizarre constants/terms to even get to the result!
- Something must be missing (at a high level) from these theorems, since they don't really expose clearly a dependence on the *quality* of the offline data... an algorithm should be able to learn very differently when the demonstration data is very good, versus when it is very bad... and a good algorithm should be able to kind of work that out and leverage it.
- Why do you use the term "SubOpt" instead of regret?
- Does PERVI (pervy?) raise some of the issues that NeurIPS (nips) had to deal with?

**Summary Of The Paper:**

This paper looks at the problem of reinforcement learning from offline data.
They authors introduce PERVI, which uses "randomized value functions" to generate an approximate posterior distribution over value functions, and then acts pessimistically with respect to those estimates for safety.
The authors support their new algorithm through an analysis in tabular MDPs, as well as more empirical evaluation with neural network function approximation.

**Summary Of The Review:**

This paper presents PERVI, a combination of randomized value functions with pessimism for offline reinforcement learning.
The paper takes mostly-existing concepts but combines them in a more elegant and coherent framework than previous papers I have seen.
The resultant algorithm is sane and sensible, and the progression of support from intuition to analysis to empirical results is comprehensive.

In terms of "the grand challenge" of offline RL I think this algorithm leaves potentially a lot of value on the table.
However, I do think this is a valuable piece of the literature and will be useful to the conference.

---

> ### Author Response · Authors · 2022-11-09
> **Response to Reviewer 69m1**
>
> Thank you for your positive comments and constructive feedback.
> ___
> > presentation of Theorems
>
> **Response**: We have revised our paper to improve its readability per your suggestion.
> ___
>
>
> > Something must be missing (at a high level) from these theorems, since they don't really expose clearly a dependence on the quality of the offline data
>
> **Response**: We remark that both Theorem 1 and Theorem 2 express the dependence on the quality of the offline data. In Theorem 1, the dependency is via the empirical regularized covariance matrix $\Lambda_h$ which is computed from the offline data. In Theorem 2, the dependency is more explicit as it depends on the optimal-policy concentrability coefficient $\kappa$ (Assumption 5.2) which measures the distribution shift from the offline data.
> ___
> > SubOpt vs regret
>
> **Response**: We follow the convention of offline RL literature to use “SubOpt” (sup-optimality) and would reserve “Regret” for the online RL setting.
>
> ___
> > Does PERVI (pervy?) raise some of the issues that NeurIPS (nips) had to deal with?
>
> **Response**: Thanks for your suggestion. We have changed our algorithm name to VIPeR (Value Iteration with Perturbed Rewards).

---

### Official Review · Reviewer_NEMf · 2022-10-24

**Confidence:** 2
**Clarity, Quality, Novelty And Reproducibility:** The paper is clearly written with suf…
**Correctness:** 4
**Technical Novelty And Significance:** 3
**Empirical Novelty And Significance:** 3
**Recommendation:** 6

**Details Of Ethics Concerns:**

NA.

**Strength And Weaknesses:**

Strength:

The proposed neural network-based offline RL algorithm is new. In appendix B3 the authors compare with existing literature on the sample complexity results.

Weakness:

I did not spot any major errors in the paper. However, I have several minor technical questions:

Q1. In line 10 of Algorithm 1, shouldn't we take argmax over action space?

Q2. A related question is, how is the argmax implemented in code since we are considering the large space-action space?


**Summary Of The Paper:**

The paper studies offline RL problems with perturbed rewards. In particular, the Q function will be parametrized as the minimum of $M$ neural networks, trained on $M$ perturbed datasets. The benefit of the proposed algorithm, compared to the UCB-based method is reducing the time complexity of action selection. On a technical side, the authors propose a data splitting analysis technique to improve dependence on log covering the number in the sample complexity result.

**Summary Of The Review:**

The idea of learning Q functions from the perturbed datasets and thus implicitly implementing the pessimism principle is neat. Theoretical analysis and preliminary experiments illustrate the potential usefulness of the proposed algorithm. I suggest a marginally accept. However, I'm not familiar with the main related papers cited in this work, so I choose a confidence score of 2.

---

> ### Author Response · Authors · 2022-11-09
> **Response to Reviewer NEMf**
>
> Thank you for your comments.
> ___
> > your Q1, Q2
>
> **Response**: For finite actions, given $\tilde{Q}_h$ (recall that $\tilde{Q}_h$ is the state-action value function estimate computed at line 9 of Algorithm 1), we simply take the maximum of $\tilde{Q}_h(s, \cdot)$ over the action space to represent $\tilde{\pi}_h(s)$. Please also see Section 3.1 for the definition of the state-value function computed from the state-action value function.
>
> In the continuous action problem, we can use the soft actor-critic framework to extract the policy  $\tilde{\pi}_h$ from the given critic  $\tilde{Q}_h$. Please see the newly added Eq. (2) in our revision. In Section A.3 (newly added) in our revision, we have discussed this soft actor-critic implementation in detail (and evaluated our algorithm variant in the D4RL benchmark).
>
> Please let us know if you have any further questions that you need to consider for changing your initial score.

---

### Author Response · Authors · 2022-11-09
**Thank you - We have provided response and revised our paper accordingly**

Dear Reviewers,

Thank you for your positive reviews and constructive feedback. We have responded to each individual review below. We have revised our paper accordingly based on your suggestions (we have also added the experimental result of our algorithm variant for the continuous action domain in the D4RL benchmark in Section A.3 of the revised paper). The newly added parts in the revised paper are highlighted in blue.




We are happy to take any further questions.


Best,
The authors.

---

### Decision · Program_Chairs · 2023-01-20

**Decision:**

Accept: notable-top-25%

**Justification For Why Not Higher Score:**

NA

**Justification For Why Not Lower Score:**

NA

**Metareview: Summary, Strengths And Weaknesses:**

This work considers an offline RL policy search which fuses randomized value function estimation with the pessimism principle to develop a unique algorithm for this setting. It exhibits strictly tighter sample complexity than prior results, and exhibits explicit dependence on the data quality of offline data. Reviewers consistently concluded the merits of the technical execution both theoretically and in experiments.

**Note From Pc:**

if the above contains the word "oral" or "spotlight" please see: "oral" presentation means -> notable-top-5% and "spotlight" means -> notable-top-25%. As stated in our emails, we are disassociating presentation type from AC recommendations